# Saturation genome editing of *BAP1* functionally classifies somatic and germline variants

Andrew J. Waters [1] ✉, Timothy Brendler-Spaeth[1,12], Danielle Smith[1,12], Victoria Offord[1], Hong Kee Tan[1], Yajie Zhao [2], Sofia Obolenski[1], Maartje Nielsen [3], Remco van Doorn [4], Jo-Ellen Murphy[5], Prashant Gupta [1], Charlie F. Rowlands [6], Helen Hanson[7,8], Erwan Delage [1], Mark Thomas[1], Elizabeth J. Radford [1,9], Sebastian S. Gerety [1], Clare Turnbull[6,10,11], John R. B. Perry [2], Matthew E. Hurles [1] & David J. Adams [1] ✉

Many variants that we inherit from our parents or acquire de novo or somatically are rare, limiting the precision with which we can associate them with disease. We performed exhaustive saturation genome editing (SGE) of *BAP1*, the disruption of which is linked to tumorigenesis and altered neurodevelopment. We experimentally characterized 18,108 unique variants, of which 6,196 were found to have abnormal functions, and then used these data to evaluate phenotypic associations in the UK Biobank. We also characterized variants in a large population-ascertained tumor collection, in cancer pedigrees and ClinVar, and explored the behavior of cancer-associated variants compared to that of variants linked to neurodevelopmental phenotypes. Our analyses demonstrated that disruptive germline *BAP1* variants were significantly associated with higher circulating levels of the mitogen IGF-1, suggesting a possible pathological mechanism and therapeutic target. Furthermore, we built a variant classifier with >98% sensitivity and specificity and quantify evidence strengths to aid precision variant interpretation.

In clinical practice, variants of uncertain significance (VUS) represent a major challenge to patient care. Germline loss-of-function variants in the BRCA1-associated protein 1 gene (*BAP1*) cause an autosomal dominant tumor predisposition syndrome, with most such variants generating frameshift or truncating alleles, yet >1,000 missense variants have been clinically observed to date. This includes 396 variants reported by multiple investigators, most of which are rare and functionally ambiguous, with >98% classified as VUS (variants with ≥1*

review status in ClinVar, 20 September 2023)[1]. Because screening guidelines for individuals who carry pathogenic germline variants in *BAP1* have recently been published, it is imperative to identify the at-risk population and further refine surveillance recommendations and risk-reduction strategies[2–4]. Although germline variants contribute to disease risk, identifying disruptive somatic *BAP1* variants in tumors may facilitate targeted oncological treatments. For example, recent evidence suggests that *BAP1*-deficient mesotheliomas are exquisitely

[1]Wellcome Sanger Institute, Hinxton, UK. [2]Metabolic Research Laboratory, Wellcome-MRC Institute of Metabolic Science, University of Cambridge School of Clinical Medicine, Cambridge, UK. [3]Department of Clinical Genetics, Leiden University Medical Center, Leiden, the Netherlands. [4]Department of Dermatology, Leiden University Medical Center, Leiden, the Netherlands. [5]Foundation Medicine Inc., Cambridge, MA, USA. [6]Division of Genetics and Epidemiology, The Institute of Cancer Research, London, UK. [7]Department of Clinical Genetics, Royal Devon University Healthcare NHS Foundation Trust, Exeter, UK. [8]Faculty of Health and Life Sciences, University of Exeter, Exeter, UK. [9]Department of Paediatrics, University of Cambridge, Cambridge, UK. [10]National Cancer Registration and Analysis Service, NHS England, London, UK. [11]Cancer Genetics Unit, The Royal Marsden NHS Foundation Trust, London, UK. [12]These authors contributed equally: Timothy Brendler-Spaeth, Danielle Smith. ✉e-mail: aw28@sanger.ac.uk; da1@sanger.ac.uk

sensitive to treatments including poly(ADP-ribose) polymerase inhibitors[5], zoledronic acid and tazemetostat[6]. Of note, of the somatic *BAP1* variants reported in pan-cancer studies on cBioPortal, functionally equivocal missense variants account for 43% (628/1,465), including 375 located in the highly conserved ubiquitin C-terminal hydrolase (UCH) domain[7].

*BAP1* encodes a ubiquitously expressed deubiquitinating enzyme that has important roles in a range of cellular processes, including contributions to transcriptional regulation, cell cycle and growth, response to DNA damage and chromatin dynamics[8]. Remarkably, rare de novo heterozygous missense variants in *BAP1* have recently been associated with Küry−Isidor syndrome (Online Mendelian Inheritance in Man, 619762), a neurodevelopmental disorder[9]. Intriguingly, neurocognitive phenotypes have not been reported in patients with cancer-associated loss-of-function *BAP1* variants, suggesting that neurodevelopment is altered by mechanisms other than loss of function, that other variants that influence *BAP1* function are in *cis* or *trans*, or that there is variable expressivity of *BAP1*-associated phenotypes. Similarly, it remains unclear why *BAP1* loss is associated with uveal melanoma, cutaneous melanoma, mesothelioma, cholangiocarcinoma, renal cancer and meningioma, which are proportionally uncommon malignancies[8].

In this study, we use saturation genome editing (SGE)[10] to profile 99% of all possible single-nucleotide variants in the *BAP1* coding sequence (6,501/6,570) with the aim of improving precision medicine. We also exhaustively profile exon-flanking intron and untranslated region (UTR) sequences, single-nucleotide and codon deletions, and short indels in ClinVar[1] and gnomAD[11]. We show that SGE data allow us to preemptively make accurate predictions regarding the pathogenicity/benignity of variants found in cancer kindreds and tumors and to identify previously unreported functional residues. We also conduct a phenome-wide association study on 63,590 carriers of *BAP1* variants and find an increased frequency of cancer in carriers of disruptive variants, as well as elevated levels of circulating insulin-like growth factor 1 (IGF-1), which is a tumor promoter and mitogen, revealing a potentially targetable mechanism contributing to *BAP1*-associated malignancies.

## Results

### Optimized SGE approach improves experiment quality
We developed a HAP1 DNA ligase 4 (*LIG4*)-knockout (KO) line with genomic integration of a clonally derived Cas9 (HAP1-A5) and confirmed *BAP1* essentiality in this line (Figs. 1 and 2a), high Cas9 activity (Fig. 2b and Extended Data Fig. 1a) and robust maintenance of haploidy (Extended Data Fig. 1b). We also optimized plasmids and transfection protocols, increasing transfection efficiency in HAP1 cells from <5% to >60% compared to other[12] SGE experiments (Extended Data Fig. 1c and Methods). To screen all coding exons of *BAP1*, we used five time points: day (D)4, D7, D10, D14 and D21. Our optimized SGE protocol led to increased editing by homology-directed repair (HDR), with ~1% unedited reads (Fig. 2c and Supplementary Table 1).

Of note, the canonical *BAP1* transcript (ENST00000460680.6) has 17 exons (Fig. 1a), and because oligonucleotide synthesis lengths are limited, 22 SGE target regions of ≤245 bp were designed to saturate all of the coding sequence, with 20- to 90-bp exon-flanking sequences also saturated (intron, 5′ UTR, 3′ UTR). For larger exons, partially overlapping regions were designed. All HDR template libraries were designed using VaLiAnT[13]. These libraries contained two different synonymous protospacer adjacent motif (PAM)/protospacer protection edits (PPEs) that were refractory to single guide RNA (sgRNA)−Cas9 cutting, preventing cleavage of incorporated tracts. Each SGE region was targeted in two separate experiments; HDR template library A contained a PPE for one sgRNA (A) and library B contained a different PPE for a different sgRNA (B) within the same target region. Transfections were performed in triplicate for both library A and library B for all 22 regions, with samples collected at the five time points mentioned above (Fig. 1b).

In total, data from 598 genomic DNA time point-replicate libraries progressed to data analysis (Fig. 1c), with an average variant coverage of 535× generated on the Illumina platform (Supplementary Table 2).

### *BAP1* essentiality permits mutational consequence separation
We used cell fitness as a biological readout of *BAP1* function, first rigorously reconfirming *BAP1* essentiality (Extended Data Fig. 2a−c) and SGE efficacy (Extended Data Fig. 2d) in HAP1 cells. To aid the selection of appropriate sgRNAs for experimentation, we performed a targeted CRISPR−Cas9 screen with 193 sgRNAs tiled across all 17 *BAP1* exons (Fig. 2a). sgRNAs for SGE were selected based principally on design parameters (as previously described[13]), with depletion kinetics also considered (Methods). We deployed these sgRNAs and variant libraries across all 22 *BAP1* target regions and confirmed editing (Extended Data Fig. 3). As expected for an essential gene amenable to SGE, scaled counts between D4 and D21 for stop-gained and frameshift variants were reduced, suggesting the depletion of cells with these variants, whereas synonymous and intron variant counts remained unchanged (Fig. 2d). By combining library A and library B, we calculated a single 'functional score' for each variant (Methods). This is the apparent growth rate across D4, D7, D10, D14 and D21 computed by log-linear regression in DESeq2 (ref. 14) and represents $\log_2$-transformed fold change (LFC) per unit time (Methods). Stop-gained, frameshift and splice donor/acceptor variants exhibited predominantly negative functional scores, whereas synonymous, intron and UTR variants had a unimodal distribution centered at 0 (Fig. 2e). Missense variants exhibited a continuum of functional scores with a negatively skewed unimodal distribution (Fig. 2e).

We next used functional scores and standard errors computed using DESeq2 to accurately define variant effects. For each variant tested, a *z*-score distribution of functional score divided by standard error was used to calculate *P* values using a two-tailed *z*-test (Methods). All unique variants were collated and the false discovery rate (FDR) was derived from the *P* value using the Benjamini−Hochberg (BH) procedure[15] to correct for multiple testing. The behavior of individual variants within this spectrum was intriguing, with, for example, specific synonymous alterations appearing disruptive and specific stop-gained and frameshift alleles, particularly those in the terminal exon, appearing nondisruptive. Codon deletions (in-frame, sequentially deleted codons) also exhibited a spectrum of scores with a bimodal distribution, which allowed us to refine key residues/domains within the BAP1 protein. Individual variant functional scores relative to the FDR threshold are shown in Fig. 2f. All mutational consequence categories, except UTR variants, had a significantly different median functional score from synonymous variants as measured by Dunn's nonparametric pairwise multiple-comparisons procedure (*q* < 0.0001; Supplementary Table 3).

### Functional analysis of gene architecture and conservation
Functional scores and FDR values were used to categorize variants into functional classes, following the integration and validation of data as described below. Variants with an FDR ≥0.01 were categorized as 'unchanged', those with an FDR <0.01 and a negative functional score were categorized as 'depleted' and those with an FDR <0.01 and a positive functional score were categorized as 'enriched'. Data for 18,108 unique variants were collected after filtering steps with variants categorized as follows: 11,912 unchanged, 5,665 depleted and 531 enriched (Supplementary Table 2). Unchanged variants centered tightly around a zero functional score (median = 0.00; range = 0.09) and enriched variants had modestly increased scores (median = 0.01; range = 0.03), while depleted variants exhibited a wider score distribution (median = −0.13; range = 0.27; Fig. 3a). As above, stop-gained variants were depleted consistently across all exons, except exon 17, suggesting an escape of nonsense-mediated decay. No enriched variants were observed for stop-gained alleles (Fig. 3b). Functional scores for missense variants were significantly different between exons as

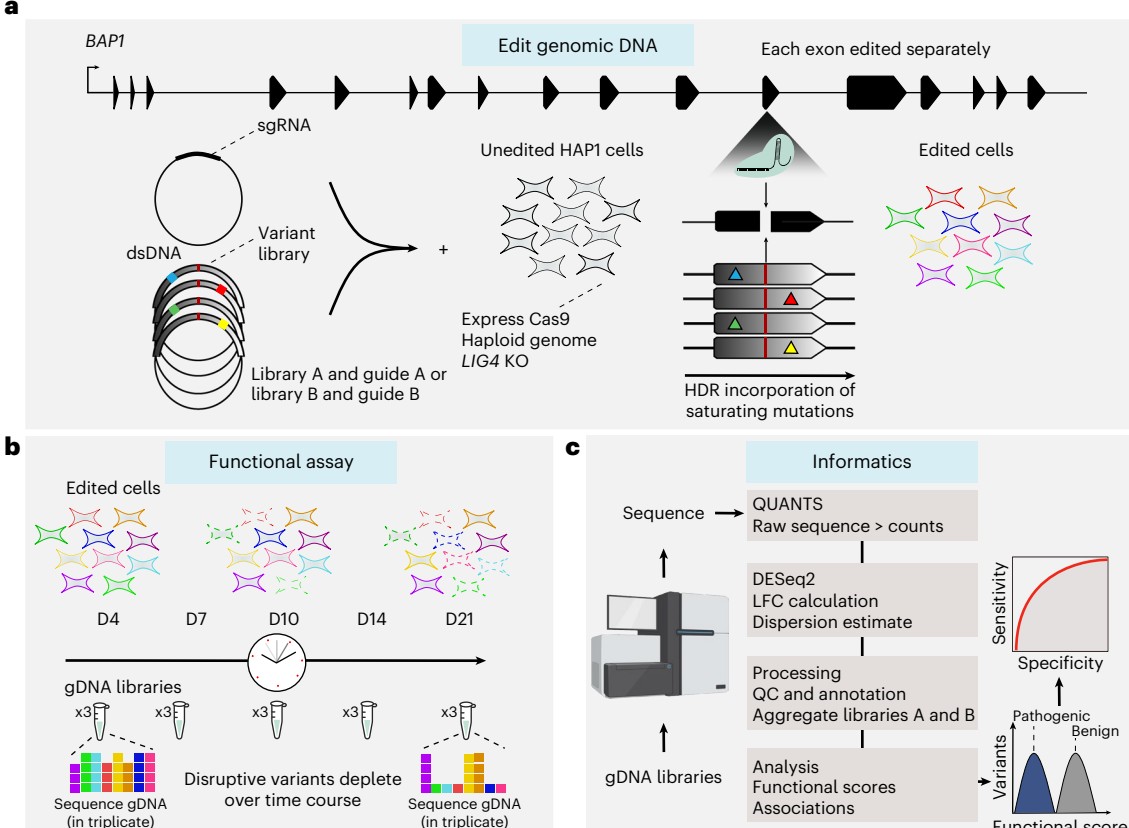

**Fig. 1 | Experimental design and workflow for SGE of the *BAP1* locus.**
**a**, Target regions of ≤245 bp were designed for all coding exons of the canonical *BAP1* transcript: ENST00000460680.6 (ref. 52). Target regions were processed in separate experiments to sequentially cover all regions. For each region, *LIG4*-KO, Cas9-expressing HAP1-A5 cells were transfected in triplicate with an sgRNA-expressing plasmid and a corresponding variant library; homologous recombination with this template library at the Cas9 lesion/cut site results in the introduction of variants into the genome, generating populations of edited cells. This allows for assessment of variant function, because only benign variants will rescue cell fitness following CRISPR–Cas9-mediated disruption of *BAP1*, an essential gene. Each region was edited separately using two independent template library–sgRNA pairs; each variant library (library A or

library B) contained saturating mutations (colored squares) and library-specific synonymous PPEs (dark red line) to prevent sgRNA–Cas9-mediated recutting of incorporated genomic tracts. dsDNA, double-stranded DNA. **b**, Cells were cultured over time with pellets collected at D4, D7, D10, D14 and D21. gDNA, genomic DNA. **c**, Sequencing was used to assess the population dynamics of genomic DNA libraries, generating counts for each variant using the QUANTS pipeline. DESeq2 was used to convert counts into an LFC of variant abundance over time. LFCs were then median scaled and a single functional score was computed through the aggregation of library A and library B data. Functional scores were categorized on the basis of a significance threshold and assessed for accuracy against variants with known pathogenic or benign classifications.

measured by Kruskal–Wallis rank sum test ($P < 0.0001$, $H = 1,093.3$). Interestingly, while missense variants were depleted in all exons, we noted that proportionally more of these variants were depleted in exons 1–9 and 15–17, and that fewer missense variants were depleted in exons 10–14. Exons 1–9 and 15–17 encode conserved UCH and protein interaction motifs, respectively. Indeed, we found a significant positive correlation between the depleted missense functional classification and conservation as measured by ortholog identity at each amino acid position in the protein (Spearman's rank: $r_s = 0.45$, $P < 0.0001$). This relationship was also observed for codon deletions (Spearman's rank: $r_s = 0.44$, $P < 0.0001$).

Because Evolutionary model of Variant Effect (EVE)[16] scores can be used as a measure of conservation for missense variants, we compared this metric to our SGE results and found that depleted and enriched variants were under more evolutionary constraint (8,525 of 8,822 total unique missense variant assessed; Fig. 3c). Variants under more evolutionary constraint are expected to be observed less frequently in population-ascertained cohorts of healthy controls from the gnomAD database, which was the case for both depleted and enriched variants compared to unchanged variants (chi-squared test; $\chi^2 = 49.1$, $P < 0.0001$; Fig. 3d). We also observed that the conserved N-terminal UCH domain

of BAP1 showed greater intolerance to missense changes and codon deletions compared to the more central regions of the protein (Fig. 3e), in keeping with its amino acid conservation. The conserved C-terminal protein interaction motifs also demonstrated intolerance to change. Of note, codon deletions precisely delineated critical domains with high accuracy and highlight uncharacterized regions (Fig. 3f,g).

Before making comparisons to clinical data, we examined the reproducibility of the functional scores and functional classifications for each variant by comparing LFCs from separate genome editing experiments. Overall, 81% of variants (14,624/18,108) were separately examined using library A and library B HDR templates, with close to linear LFC value correlations (Pearson's $R = 0.95$, $P < 0.0001$; Fig. 4a). When functional classifications were computed using library A or library B LFCs and FDRs, a 90% concordance of variant classification was observed (13,106/14,624; Fig. 4a). As LFCs and functional classifications were found to be highly correlated, to increase robustness, library A and library B LFCs for each variant were combined into a single 'combined LFC' and termed the abovementioned 'functional score' (Methods). As expected, variant LFCs within PPE codons differed between libraries (Extended Data Fig. 4a,b). Therefore, variants in PPE codons examined by only library A or library B were excluded from downstream analyses

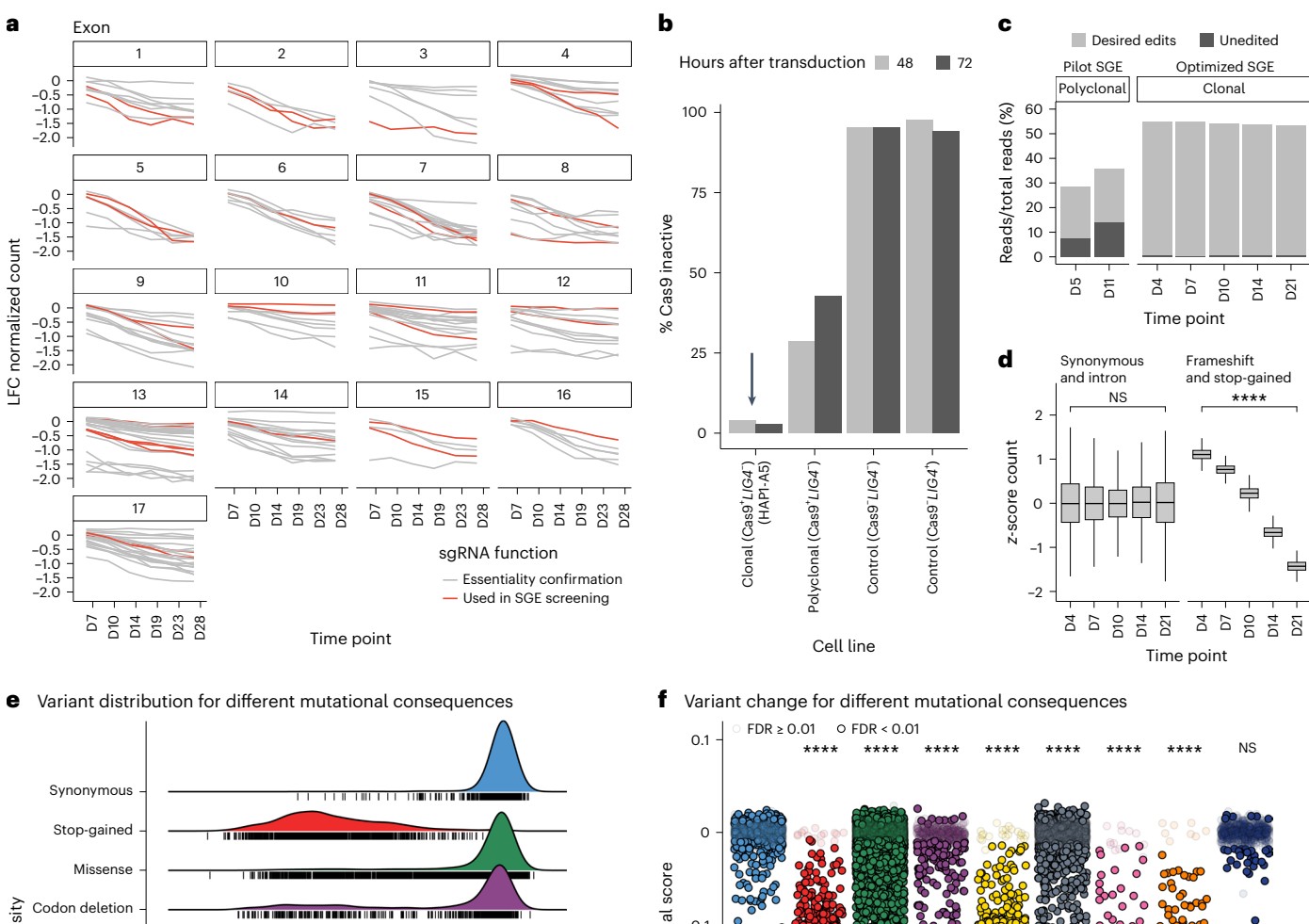

**Fig. 2 | Cell fitness/essentiality using optimized SGE reports the mutational consequences of editing of the *BAP1* locus. a**, A targeted CRISPR–Cas9 screen in HAP1-A5 cells confirmed *BAP1* essentiality and permitted selection of sgRNAs with favorable depletion kinetics for use in SGE (Methods). **b**, FACS analysis counts (green fluorescent protein (GFP)-positive cells) demonstrated that the HAP1-A5 clone has very high Cas9 activity (arrow), measured at 48 and 72 h after transduction with a GFP/blue fluorescent protein (BFP) activity construct (Methods: 'Ploidy and FACS analysis'), compared to the parental 'Polyclonal (Cas9+ *LIG4*−) line. A total of 10,000−20,000 cells were analyzed for each line. Cell count percentages derived from negative-control lines with no Cas9 showed expected low levels of Cas9 activity (see Extended Data Fig. 1a and Supplementary Fig. 1a for representative FACS data). **c**, Editing using pilot SGE conditions: a template library (496 variants) coupled with sgRNA-A targeting exon 5 of *BAP1* was transfected into the polyclonal (Cas9+ *LIG4*−) line and cells were sampled at D5 and D11 (time points previously[10] used in SGE). More than 10% of the counts were unedited (wild type), which decreased to <1% when the clonal (Cas9+ *LIG4*−) cell line (HAP1-A5) was edited using the same sgRNA and HDR homology arms with optimized SGE conditions, including a high-complexity template library

(1,040 variants) sampled over five time points. **d**, Count abundance for variants that resulted in synonymous changes or edited intronic regions did not change significantly over a 21-day SGE screen (two-sided Mann−Whitney−Wilcoxon test; D4 versus D21 counts, $P = 0.3$; NS, not significant), whereas variants resulting in stop-gained and frameshift consequences were significantly depleted (****$P < 2.2 \times 10^{-16}$; $n = 8,707$ synonymous and intronic variants; $n = 5,628$ frameshift and stop-gained variants; mean $z$-score counts of three biological replicates at each time point). Boxes show the interquartile range, the horizontal lines show the median $z$-score count and whiskers show the maximum and minimum values that are not outliers. **e**, Density plot showing functional scores colored by Ensembl Variant Effect Predictor (VEP)[53] mutational consequence. Black tick marks represent single variant values. **f**, Jitter plot showing VEP mutational consequence categories versus functional score. Data points that have FDR ≥ 0.01 are semitransparent and the median synonymous functional score differs significantly from that for all other categories except UTR (Kruskal−Wallis test: $P < 2.2 \times 10^{-16}$, $H = 6,692.2$; two-sided Dunn's BH FDR ****$q < 2.2 \times 10^{-16}$).

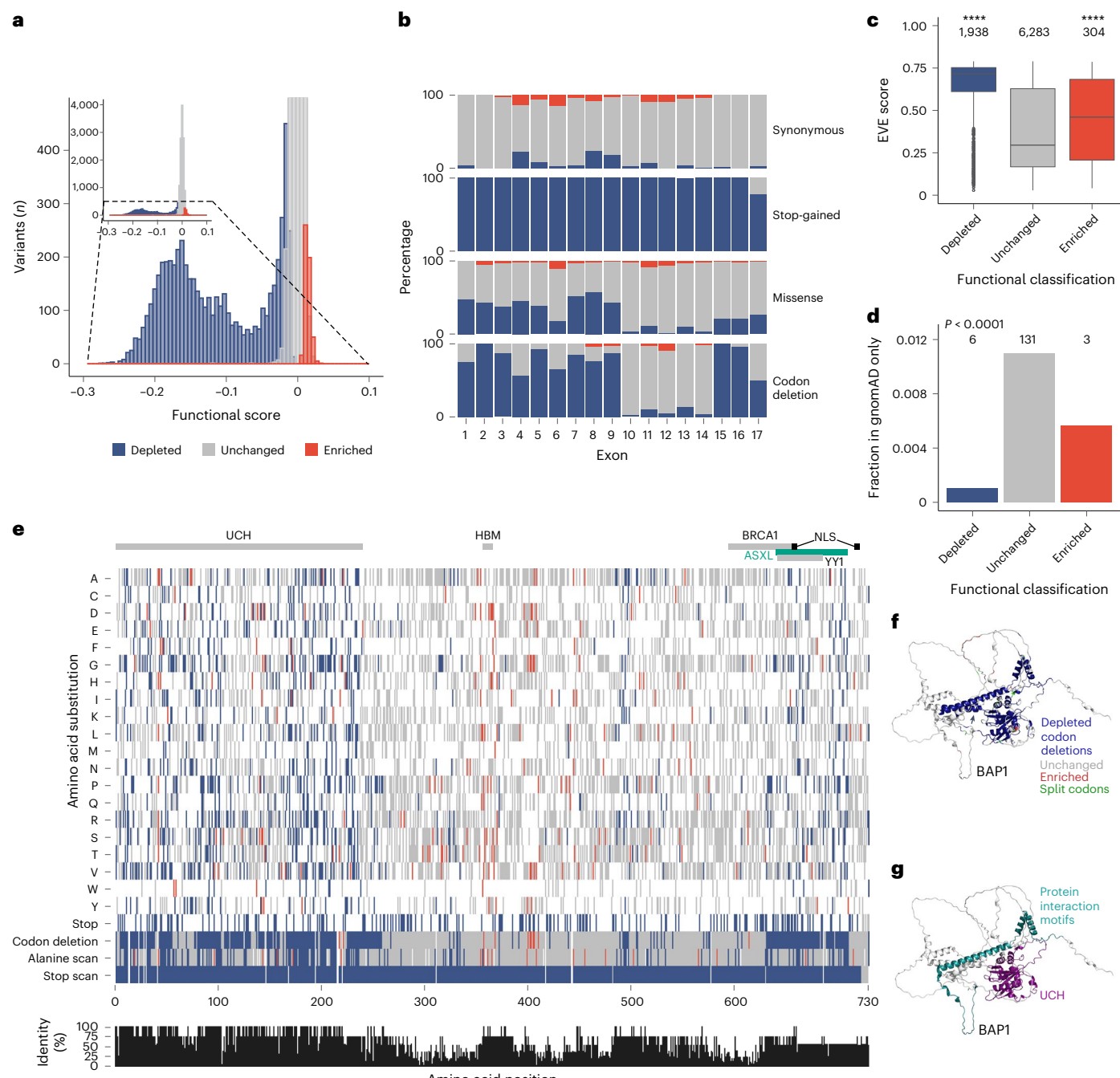

**Fig. 3 | Functional classification of *BAP1* variants. a**, Histogram showing all 18,108 unique variants assayed, grouped into 75 intervals and colored according to functional classification. Inset shows a magnified section of functional score intervals with ≤500 variants. **b**, Composition of functional classes by exon and mutational consequence (color key as in **a**). **c**, EVE scores for functional classes (*n* variants in class shown). Both depleted and enriched classes have significantly different median values from the unchanged class (Kurskal–Wallis, $P < 2.2 \times 10^{-16}$; two-sided Dunn's BH FDR, ****$q < 0.0001$; depleted $q < 2.2 \times 10^{-16}$ and enriched $q = 3.4 \times 10^{-5}$), demonstrating that depleted and enriched variants are less represented over evolution compared to unchanged variants and are therefore more likely to be disruptive. Boxes show the interquartile range, horizontal lines show the median EVE score, whiskers show maximum and minimum values that are not outliers, and outliers are shown as points. **d**, The bar chart shows the number of variants in each class that are in gnomAD and not ClinVar (*n* shown) divided by the number of variants in each class assayed. Fewer depleted and enriched variants than unchanged variants were observed in gnomAD (two-sided

chi-squared test: $\chi^2 = 49.1$, $P < 2.14 \times 10^{-11}$). **e**, Heat map showing amino acid-level substitutions ('A':'stop') created by nucleotide-level saturation across 730 codons (single nucleotide variants (SNVs) only), colored by functional classification (SNV missense changes with discordant functional classifications between alternative codons were excluded; *n* = 158). Of note, 'codon deletion', 'alanine scan' and 'stop scan' changes were designed to be incorporated at each of the 720 nonsplit codons (of 730 total codons). Bar chart shows the percentage identity calculated from Geneious alignment of the eight species shown in Fig. 6d. Key protein regions are shown (UCH, ubiquitin C-terminal hydrolase; HBM, HCF1 binding motif; BRCA1, BRCA1 binding domain; ASXL, additional sex combs like 1/2/3 interaction; YY1, Ying Yang 1 binding domain; NLS, nuclear localization signal). **f,g**, AlphaFold[54] BAP1 model with SGE-depleted codon deletions colored dark blue (**f**). Depleted codon deletions accurately delineate the UCH domain (purple) and protein interaction region (cyan), as highlighted in **g**. Depletion also occurs in uncharacterized regions, including the α-helix C terminal to the UCH domain, proximal to the protein interaction region (arrow, **f**).

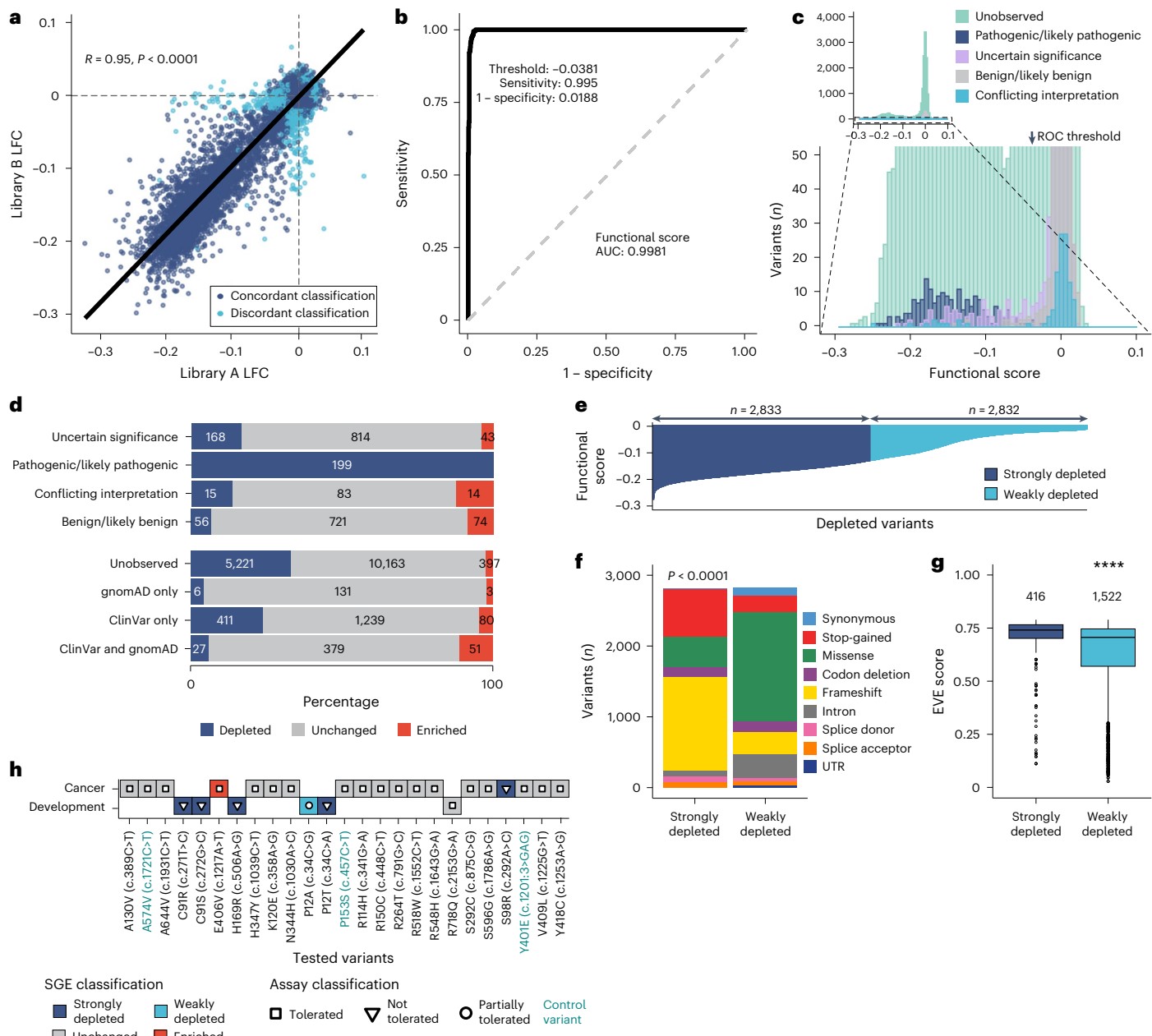

**Fig. 4 | SGE data are technically robust and provide highly accurate clinical classification. a**, Independent SGE libraries (A and B) were used to edit most target regions with 13,106 of 14,624 variants showing a concordant functional classification (dark blue) and 1,518 variants discordant between libraries (light blue). Of note, the degrees of LFC for each independent variant measurement were highly concordant based on Pearson's correlation coefficient ($R$) and two-tailed $t$-test $P < 2.2 \times 10^{-16}$. **b**, ROC curve for SGE functional score, with AUC value shown. Also shown is the ideal threshold for maximum diagnostic sensitivity and specificity (plotted as '1 − specificity'). Calculated using pROC (version 1.18.4)[55] in R. **c**, Top, a histogram showing the 18,108 unique variants grouped within 75 intervals of functional score, colored by ClinVar clinical significance. Bottom, a magnified region highlights that pathogenic/likely pathogenic (dark blue) variants are depleted. The arrow shows the $x$-axis position of the ideal threshold. **d**, Top, functional classification by ClinVar clinical significance (≥1*, 4 September 2023). Bottom, functional classification by observation in ClinVar and gnomAD ($n$ variants shown). **e**, Depleted variants ($n = 5,665$) categorized into strongly depleted (lower 50%, dark blue) and weakly depleted

(upper 50%, light blue) variants, either side of the median functional score (−0.1260642). **f**, More frameshift and stop-gained variants and fewer missense variants were strongly depleted compared to weakly depleted variants (two-sided chi-squared test, $\chi^2 = 10,759$, $P < 2.2 \times 10^{-16}$). **g**, Strongly and weakly depleted missense variants have significantly different EVE scores (two-sided Mann–Whitney–Wilcoxon test, ****$P < 2.2 \times 10^{-16}$). Boxes show the interquartile range, horizontal lines show the median EVE score, whiskers show maximum and minimum values that are not outliers, and outliers are shown as points. **h**, Concordance of SGE functional classification and orthogonal functional assays for VUS in patients with cancer and developmental disorders[9,25]. Color indicates SGE classification and shape corresponds to orthogonal assay classification. Control variants (from a case–control study[25]) are shown in green text. SGE variants that were strongly depleted (dark blue) and not tolerated in orthogonal assays (triangles) are completely concordant. P12A, which was partially tolerated in an orthogonal assay, was weakly depleted in SGE. All tolerated variants (white squares) in assays were unchanged in SGE (gray), except for E406V, which was enriched (red).

**Table 1 | Quantification of *BAP1* SGE assay performance in the classification of missense variants using the ACMG-AMP framework**

| Validation truth set | No. Path. | No. Ben. | Assay readout (pathogenics) | | Assay readout (benigns) | | LR$_{Path}$ | PS3 | LR$_{Ben}$ | BS3 |
|---|---|---|---|---|---|---|---|---|---|---|
| | | | Dep. | U/E | Dep. | U/E | | | | |
| ClinVar (≥2*) | 0 | 6 | 0 | 0 | 0 | 6 | – | – | 0 | NA |
| ClinVar (≥1*) | 7 | 6 | 7 | 0 | 0 | 6 | 6.0 | PS3_mod | 7.0 | BS3_mod |
| Systematic | 2,423 | 138 | 2,419 | 4 | 4 | 134 | 27.6 | PS3_str | 470.6 | BS3_vstr |

Assay performance was evaluated based on the relative numbers of depleted (Dep.) and unchanged/enriched (U/E) readouts observed for the truth sets of pathogenic (Path.) and benign (Ben.) variants. Truth sets were either constructed using all available ClinVar-classified missense variants with ≥2* review status or ≥1* review status or using a systematic approach in which the pathogenic truth set consisted of nonsense and frameshift variants and the benign truth set consisted of missense variants ascribed benignity based on current ACMG-AMP requirements (two evidence items toward benignity unless BA1 was met). ACMG, American College of Medical Genetics and Genomics; AMP, Association for Molecular Pathology; mod, moderate; str, strong; vstr, very strong; NA, not applicable.

($n$ = 140). As above, our functional score was calculated as the apparent growth rate over five time points, an analysis previously used in SGE[17]. This approach is appropriate for our data, as LFCs between later time points were linearly related (Extended Data Fig. 4c–f). The functional scores, functional classifications (depleted, unchanged and enriched) and downstream comparisons used throughout this study were derived from these combined LFC values.

Full nucleotide and protein-level variant effect maps are provided in Extended Data Figs. 5 and 6, respectively. The full dataset with annotations and scores is also available for download at https://github.com/team113sanger/Waters_BAP1_SGE and as Supplementary Data 1 and 2. Variant scores and classifications can also be searched on the BAP1 Viewer: https://bap1-viewer.shinyapps.io/bap1viewer/.

### Sensitive and specific classification of clinical variants

To further examine functional scores, we first identified variants with strong clinical/functional data in support of their classification, curating 851 benign ('true negative') and 199 pathogenic ('true positive') variants that had at least one star (≥1*) in ClinVar (downloaded 4 September 2023). We used the functional scores for these variants to generate a receiver operating characteristic (ROC) curve, with the area under the curve (AUC) computed (Fig. 4b). We found that our functional score was highly accurate at classifying these variants with a sensitivity of >99%, a specificity of >98%, a classification error rate close to 0 (<0.002%) and a precision-recall AUC of >0.999 (Supplementary Table 4). We also used our data to explore the relationship between functional score/classification and reported clinical classifications and found high concordance (Fig. 4c,d).

Of note, many clinically used in silico classifiers, including EVE[16], SIFT[18] and PolyPhen-2[19], use protein-level information to predict function, whereas SGE assesses function at the nucleotide level, capturing variant effects on splicing, RNA folding, codon usage and other non-protein-level processes. We observed that few synonymous variants were depleted in our screen (Figs. 2e,f and 3b). Importantly, synonymous variants that were classified as depleted had significantly higher SpliceAI scores than unchanged synonymous variants ($P$ < 0.0001, two-sided Mann–Whitney–Wilcoxon test; Extended Data Fig. 7a), suggesting functional relevance. In the absence of functional or in silico data, synonymous variants are routinely classified as VUS[20], suggesting that these variants could be misclassified without SGE. Importantly, we found that variants (missense, stop-gained and synonymous) created by SGE through different nucleotide-level changes had highly correlated LFCs, as expected (Pearson's $R$ = 0.91, $P$ < 0.0001; Extended Data Fig. 7b). Missense changes alone also showed a high correlation in LFCs between alternative codons (Pearson's $R$ = 0.89, $P$ < 0.0001). Of note, 8,822 unique nucleotide-level changes in our screen resulted in 4,619 unique missense changes at the protein level, of which 3,993 could be examined using alternative codon generation, with 16.7% (667/3,993) showing different

functional classifications. Thus, not all missense changes have equal effects when encoded by alternative codons, further highlighting the importance and richness of SGE functional assessment at the nucleotide level.

Because very few *BAP1* missense variants have been ascribed to be pathogenic or benign, a direct comparison of sensitivity and specificity using an AUC summary metric between in silico tools and SGE functional scores for missense variants alone is not possible. However, when we compared experimental data with in silico tools, we found that EVE, PolyPhen-2 and CADD[21] predicted SGE classifications of non-splice region missense variants with 77–79% accuracy (Extended Data Fig. 7c). With per-variant examination, it is notable that EVE, PolyPhen-2 and CADD classify proportionally more missense variants as pathogenic, probably damaging and likely pathogenic, respectively, suggesting that SGE may have a relatively higher specificity (Extended Data Fig. 7d–g).

### *BAP1* SGE assay evaluation against the ACMG evidence framework

Next, we sought to quantify the evidence strength at which predictions from our assay could be applied using the American College of Medical Genetics and Genomics (ACMG) framework for variant interpretation[20]. To this end, we generated further truth sets of high-confidence pathogenic and benign variants (Methods and Supplementary Table 4) against which to evaluate assay performance using the established framework from Brnich et al.[22]. We aimed to evaluate assay performance in predicting the impact of missense variants, which are challenging to classify.

We observed that 99.8% (2,419/2,423) of variants in our pathogenicity truth set exhibited the expected depletion in the assay output, whereas 97.1% (134/138) of variants in our benignity truth set were unchanged or enriched in the assay (Table 1). These observations equate to likelihood ratios toward pathogenicity of 27.6 and benignity of 470.6, which correspond to strong and very strong evidence strengths, respectively[22,23]. Notably, when using truth sets constructed using ClinVar-classified missense variants only (≥1* review status), there was full concordance with assay results; however, due to small sample numbers, these truth sets yielded likelihood ratios toward pathogenicity and benignity of 6.0 and 7.0, respectively, both equating to a moderate strength of evidence. Further limiting truth sets by restriction to ClinVar variants of ≥2* did not allow the generation of evidence strengths due to the absence of pathogenic variants.

### Assessment of *BAP1* variants in cancer and neurodevelopment

We were intrigued by the observation that some patients with germline *BAP1* variants have been reported as being predisposed to tumors, whereas others have a neurodevelopmental disorder. SGE allows us to test whether these variants have different functional outcomes.

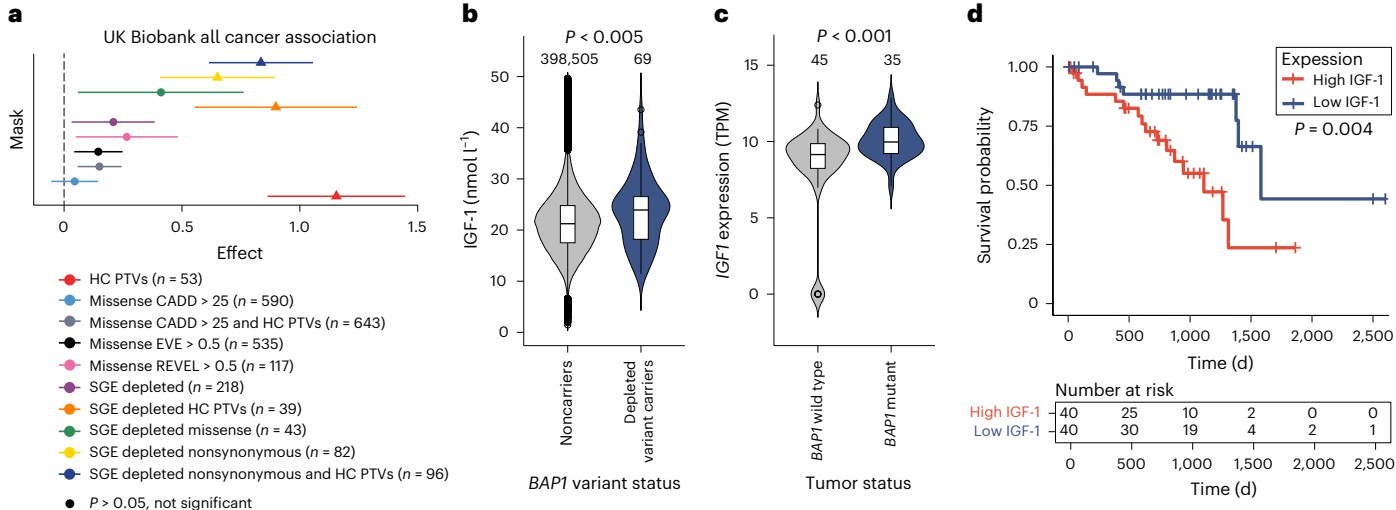

**Fig. 5 | SGE-depleted variants are associated with population-level cancer risk and increased IGF-1 levels. a**, PheWAS forest plot for all-site cancers using SGE-depleted variants and controls; regression model effect is shown by data points and ± effect standard error is shown by bars (Supplementary Table 7). Rare variant burden test masks (and CADD, EVE and REVEL[56] predictors) are shown by color for *BAP1* variants in UK Biobank (*n* carriers shown in key). Significance, according to the corrected *P* value determined by generalized linear modeling (Supplementary Method 14), is indicated by a triangle (significant) or a circle (not significant). SGE-depleted nonsynonymous variants (yellow) showed a significant effect and are therefore associated with increased cancer risk. SGE-depleted high-confidence (HC) protein-truncating variants (PTVs; orange) demonstrated a significant effect, as did HC PTVs (red). **b**, UK Biobank SGE-depleted nonsynonymous variant carriers (*n* = 69) had a significantly higher median blood concentration of IGF-1 compared to noncarriers (*n* = 398,505); *P* < 0.005 (*P* = 0.0033, two-sided Mann–Whitney–Wilcoxon test). Violin plots are colored by *BAP1* variant status, boxes show the interquartile range, horizontal lines show the median IGF-1 blood concentration (nmol l⁻¹), whiskers show

maximum and minimum values that are not outliers, and outliers are shown as points. **c**, *IGF1* mRNA expression levels in transcripts per million (TPM) obtained from TCGA for 80 uveal melanoma tumors[28]. *BAP1*-mutant tumors (*n* = 35) have higher *IGF1* expression than those with wild-type *BAP1* (*n* = 45). Colors, outliers and box description are as in **b**, except the horizontal line is the median *IGF1* expression in tumors. *P* < 0.001 (*P* = 0.00029, two-sided Mann–Whitney–Wilcoxon test). **d**, The 80 samples from patients with uveal melanoma were ranked by TCGA *IGF1* expression level, with tumors with the top 50% highest expression levels classified as having high expression and the bottom 50% classified as having low expression. Top, Kaplan–Meier estimates, with deceased status (overall survival) shown by vertical tick marks and the model for survival probability based on the overall survival time (in days) shown by lines colored to indicate *IGF1* expression level. The *P* value was calculated using the log-rank test and indicates a significant difference between the overall survival probability for tumors with high and low *IGF1* expression from patients in the cohort. Bottom, number at-risk table shows a higher number of patients alive at each time increment for patients whose tumor expressed low versus high levels of IGF-1.

To this end, we ranked the 5,665 depleted variants (we excluded enriched variants) by categorizing them on either side of the median, defining them as strongly depleted (*n* = 2,833) or weakly depleted (*n* = 2,832) (Fig. 4e). We observed that the proportions of mutational consequences seen in strongly and weakly depleted categories were significantly different from one another (chi-squared test; $\chi^2$ = 10,759, *P* < 0.0001), with more missense and fewer stop-gained and frameshift mutations weakly depleted (Fig. 4f). We also observed that weakly depleted missense variants were less conserved (*P* < 0.0001, two-sided Mann–Whitney–Wilcoxon test; Fig. 4g). Strongly depleted variants were also depleted at an earlier time point (D10) in the screen compared to most weakly depleted variants (Extended Data Fig. 7h). Taking these findings together, it appears that a subset of missense variants (*n* = 426; strongly depleted) behave similarly to stop-gained/frameshift variants and a larger number of missense variants (*n* = 1,548; weakly depleted) have a less extreme LFC and slower change in variant abundance.

Sixteen *BAP1* germline variants have been associated with developmental disorders[9,24]. In our screen, we assayed 15 of these 16 variants and found that 13 of 15 were classified as depleted (Supplementary Note 1 and Extended Data Fig. 7i). Functional studies have previously been performed on variants associated with development[9] and cancer[25], with perfect concordance observed between these orthogonal assays and SGE to the degree that a putative hypomorphic allele can be distinguished (Supplementary Note 1 and Fig. 4h).

Next, we analyzed data from a comprehensive clinical analysis of families with *BAP1*-tumor predisposition syndrome (TPDS)[26]

(Supplementary Note 1 and Supplementary Table 5). Interestingly, we found that carriers of depleted variants had a significantly earlier age of onset than carriers of unchanged variants (*P* < 0.01, two-sided Mann–Whitney–Wilcoxon test; Extended Data Fig. 7j). However, we saw no differences between strongly and weakly depleted classifications for age of onset or cancer type (Supplementary Note 1 and Extended Data Fig. 7j,k). Moreover, while there was a difference in molecular consequences (Fig. 4f), conservation (Fig. 4g) and effect sizes, germline cancer-associated variants did not have different functional score effect sizes compared to development-associated variants (Extended Data Fig. 7i,k).

### *BAP1* disruption is associated with cancer and high IGF-1 levels in UK Biobank

Next, we used whole-exome sequencing data from 454,787 individuals in UK Biobank to explore the phenotypic consequences of *BAP1*-disruptive alleles[27]. We identified 57 SGE-depleted, 80 SGE-enriched and 754 SGE-unchanged variants in the exomes of 297, 1,960 and 61,333 carriers, respectively (Supplementary Table 6). We performed a phenome-wide association study (PheWAS) analysis (Supplementary Method 14), focusing on depleted variants only. To evaluate the association of these variants with overall cancer risk, we generated cancer-type phenotypic variables and rare variant burden test masks (variant sets) (Fig. 5a, Extended Data Fig. 8a and Supplementary Table 7). We found that SGE-depleted nonsynonymous variants were significantly associated with all-site cancer predisposition (*P* = 7.85 ×10⁻⁰³; *n* = 82) with this variant set/mask composed of missense and high-confidence

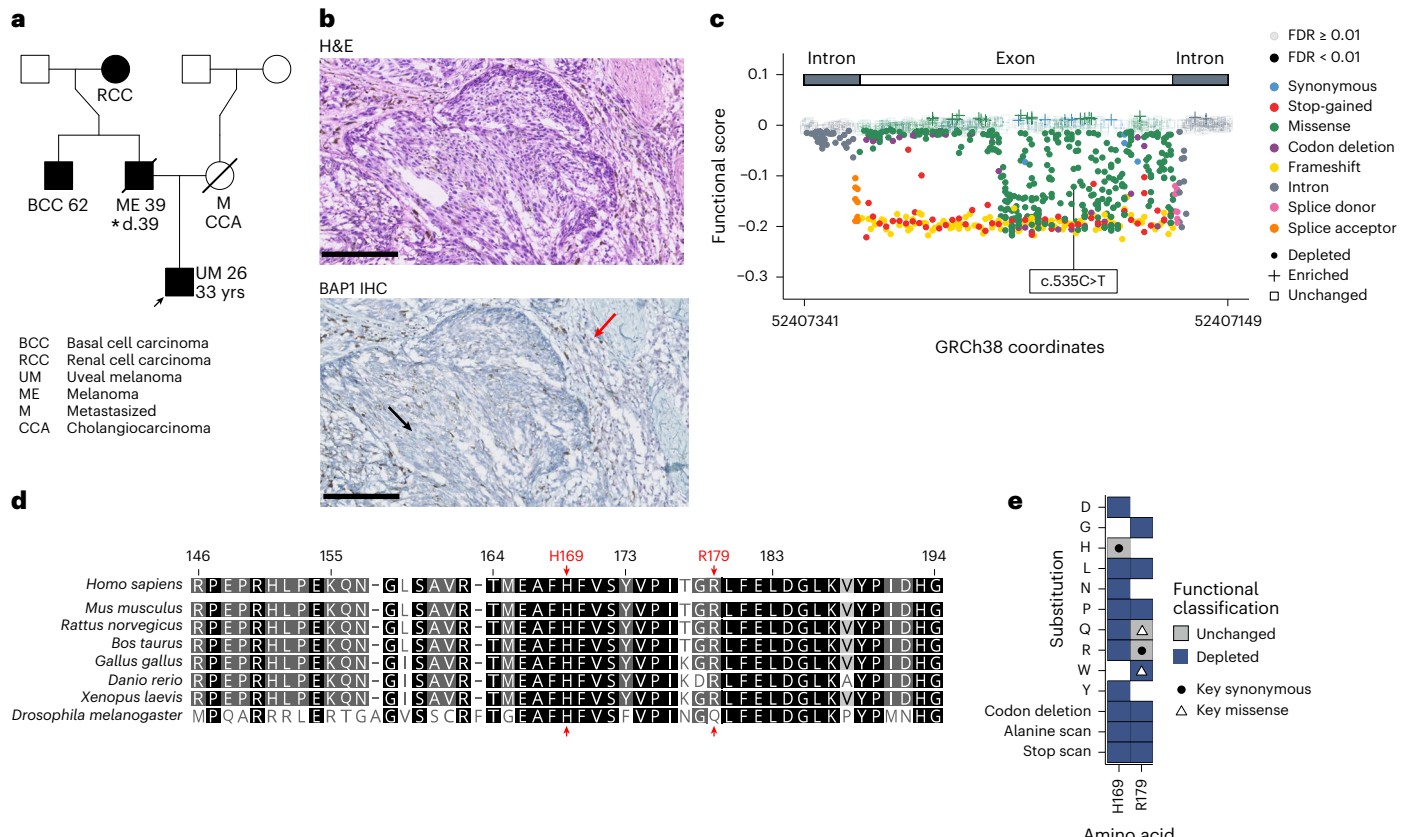

**Fig. 6 | Integration of the *BAP1* SGE functional score with a clinical example. a**, Pedigree with a proband carrying a c.535C>T variant (HGVSc, ENST00000460680.6:c.535C>T; HGVSp, ENSP00000417132.1:p.Arg179Trp; R179W) in exon 7 of *BAP1*. The proband was a 33-year-old male presenting with uveal melanoma (UM) at 26 years (arrow) whose father, uncle and grandmother presented with melanoma (ME), basal cell carcinoma (BCC) and renal cell carcinoma (RCC), respectively. The proband's mother was not known to be a carrier and died of metastatic (M) cancer, possibly cholangiocarcinoma (CCA). The pedigree follows established nomenclature: black, clinically confirmed disease (malignant tumor); square, male; circle, female; diagonal line, deceased; d., age at death; number, age at disease presentation. An asterisk indicates the patient for whom samples are shown in **b**. **b**, Pathology of the primary cutaneous melanoma in the patient from **a**. Top, micrograph showing hematoxylin and eosin (H&E) staining; Bottom, micrograph showing BAP1 immunohistochemical staining; staining is absent in tumor tissue (black arrow) but is present (purple cells) in immune infiltrate (red arrow). Scale bars, 100 μm. Micrographs are representative of three histological sections. **c**, Functional scores across exon 7. Exonic/intronic ranges within the target region are shown, with points colored

by VEP consequence. Transparency based on FDR. Shape denotes functional classification. The variant in **a** is labeled. **d**, Multiple-sequence alignment of exon 7 created by global alignment of BAP1 orthologs from eight species (gap open/extension penalty = 12/3); numbers are protein positions of human BAP1 (ENSP00000417132.1) and residues are colored by identity (black, 100%; dark gray, 80–100%; light gray, 60–80%; white, <60%). R179 (and the highly conserved H169 proton donor) is highlighted by a red arrow. Note that the glutamine residue in *Drosophila* aligns at human position R179, the only missense variant at this position tolerated in SGE. **e**, Heat map (see Extended Data Fig. 6 for the full heat map) of amino acid substitutions for two key positions, H169 and R179, colored by functional classification. White space results from SNV saturation not producing all amino acid substitutions. c.535C>T produces R179W, which is depleted. R179R, a synonymous change, is unchanged, other missense changes (R179P/L/G/A/*) and R179 codon deletion are depleted and only R179Q is tolerated. H169 in the catalytic core is intolerant to all observed changes, except for a synonymous change. Black circle, key synonymous changes; white triangle, key missense changes.

protein-truncating variants (PTVs), which were classified as depleted by SGE.

Beyond cancer, we also examined the association between UK Biobank *BAP1* variants and quantitative traits (Supplementary Table 8). As a result, we identified that circulating IGF-1 levels were significantly increased in carriers of SGE-depleted nonsynonymous *BAP1* variants compared to noncarriers (Fig. 5b; $P < 0.005$, two-sided Mann–Whitney–Wilcoxon test). Importantly, IGF-1 levels in carriers with and without a cancer diagnosis did not differ, indicating that significantly increased IGF-1 levels are specific to individuals with SGE-depleted nonsynonymous *BAP1* variants rather than a cancer diagnosis, and suggests a possible mechanism of BAP1-mediated pathogenicity (Supplementary Note 2).

To further investigate the association of SGE-depleted alleles with increased IGF-1 levels, we obtained The Cancer Genome Atlas (TCGA) RNA-seq data[28] for uveal melanomas and found a strong association

of loss-of-function *BAP1* alleles with *IGF1* mRNA expression ($P < 0.001$, two-sided Mann–Whitney–Wilcoxon test) and poor prognosis by Kaplan–Meier estimate ($P = 0.004$) (Fig. 5c,d).

## SGE use in kindred resolution and molecular tumor boards

As a further exemplar of the value of our *BAP1* SGE data, we identified a family whose proband presented at the age of 26 years with uveal melanoma. A review of the family history revealed other *BAP1*-associated tumors segregating over three generations (Fig. 6a,b), with sequencing revealing a germline c.535C>T (R179W) variant in the *BAP1* gene. c.535C>T was depleted in our SGE experiment, with a functional score of −0.122 and an FDR of <0.01 (Fig. 6c). This variant had been classified in the clinic as a VUS, but together with our SGE data it has been reclassified as likely pathogenic (ACMG, class IV), a result that will contribute to the clinical management of this kindred. Of note, R179W falls in a highly conserved region of BAP1, which includes the proton donor

residue at H169. At codon R179, the only SGE-tolerated substitution is R179Q, with glutamine being the conserved residue in the *Drosophila melanogaster* BAP1 ortholog Calypso (Fig. 6d,e).

Finally, to further explore the use of SGE data and identify novel *BAP1* variants, we queried tumor sequence data for a cohort of 394,756 patient samples in the Foundation Medicine database[29] and found 12,172 (3.1%) unique *BAP1*-altered specimens harboring 13,283 *BAP1* alterations, including all possible changes at codon R146 (Extended Data Fig. 9a–d). Because these variants were derived from tumor-only sequencing, germline DNA was obtained[30] and sequenced from a patient with breast/cholangiocarcinoma whose sister was diagnosed with renal carcinoma (Extended Data Fig. 9a–d). Both patients were confirmed to carry a germline R146K (c.437G>A) variant identified as disruptive by SGE (Extended Data Fig. 9a–e), providing another example of how SGE data can help improve diagnostic precision.

## Discussion

*BAP1* encodes a tumor suppressor involved in a variety of cellular processes, including DNA damage response (controlling BRCA1 through association with BARD1 (ref. [31])), transcription (by acting in complex with HCF1 (refs. [32,33]) and YY1 (ref. [34])), cell cycle regulation[35] and apoptosis[36], and functions as a deubiquitinase. *BAP1* is somatically mutated in many tumors, with germline variants predisposing to cancer[37–39] and a developmental disorder[9]. However, predicting the pathogenicity of missense variants in the context of either condition is extremely challenging.

Using cell fitness as a phenotypic readout, we assayed 18,108 variants across five time points in triplicate, showing clear separation of function for *BAP1*, with 99% (906/920) of stop-gained variants significantly depleted compared to 10% (572/5,714) of synonymous, intronic and UTR variants. Synonymous variants are generally held to be nonpathogenic/nondisruptive, which we confirmed for *BAP1*. However, a synonymous variant (c.936T>G) observed in renal cancers[40] and extensively characterized[41] as loss of function due to exon skipping was found to be depleted in our SGE screen. Furthermore, we observed that depleted synonymous variants were more likely to be associated with cryptic splicing[42] than unchanged variants, demonstrating the high sensitivity of our data and the value of functional assessment in the endogenous genomic/nucleotide context. Critically, for 8,822 missense variants, including many clinically observed VUS, we ascribe function. Other studies have functionally assessed subsets of *BAP1* variants, and we found that functional classification between these assays and SGE is highly concordant, to the extent that a putative hypomorph can be jointly distinguished[9,25]. This is encouraging because *BAP1* functional assays orthogonal to SGE are low throughput and have semiquantitative readouts. SGE provides data at a near-exhaustive scale that are sensitive, specific and quantitative.

It is interesting that depleted variants are associated with both cancer and developmental disorders, with key codons including C91 and H169 (nucleophile and proton donor, respectively) altered in both conditions[9,43]. Intriguingly, most *BAP1* variants associated with developmental disorders are missense, with the majority known to be de novo, and therefore more likely to be causative of disease (as neither parent has the variant or a developmental disorder). Of note, eight of nine reported[9] missense changes in Küry–Isidor syndrome are classed as depleted by SGE, with one variant (c.2153G>A) in terminal exon 17 having unchanged abundance (consistent with functional data and VUS classification by Küry et al.[9]). Interestingly, heterozygous *BAP1* frameshift and missense variants reported in a meta-analysis of over 31,000 developmental disorder families[24], where variant carriers had phenotypes overlapping with Küry–Isidor syndrome, are also depleted by SGE, suggesting that *BAP1* haploinsufficiency contributes to pathogenesis.

To exemplify the value of our *BAP1* SGE data, we obtained phenotypes from the UK Biobank[27] for all *BAP1* variant carriers and noncarrier controls, revealing that nonsynonymous *BAP1* variants classified as disruptive/depleted by SGE were significantly associated with a cancer diagnosis and, independent of cancer, significantly higher levels of circulating IGF-1. In addition, we found elevated *IGF1* mRNA expression in *BAP1*-mutant uveal melanomas, where higher expression levels are associated with a poorer prognosis. This suggests that increased IGF-1 expression downstream of *BAP1* loss has both non-cell-autonomous and cell-autonomous effects.

IGF-1 is a circulating hormone and growth factor that functions in cellular proliferation and apoptosis[44]. High circulating IGF-1 levels are associated with an increased risk of colorectal, breast and prostate cancers, with limited evidence for other cancer types[45]. IGF-1 is also a neurotrophic peptide and a major regulator of fetal growth and development, performing critical roles in the central nervous system[46]. In the brain, IGF-1 is extensively expressed during development, with expression limited to specific areas and very low levels once the brain is formed[47]. The postnatal therapeutic use of IGF-1 has been trialed for several neurodevelopmental disorders, including autism spectrum disorder (ASD), Rett syndrome and fragile X syndrome[48]. Increased circulating IGF-1 levels have been found in children with ASD, a disorder seen in individuals with Küry–Isidor syndrome[49]. Furthermore, *IGF1* mRNA overexpression in mice results in abnormal brain development, including increased myelination[50], a phenotype also observed in a mouse ASD model[51]. Therefore, it will be of value to explore the candidacy of IGF-1 as a prophylactic target in both cancer and Küry–Isidor syndrome.

In conclusion, we show that the exhaustive functional assessment of loci with SGE has the potential to aid patient diagnosis, our biological understanding of disease mechanisms and our fundamental understanding of gene/protein function.

## Online content

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

## Methods

All research conducted in this study complies with all relevant ethical regulations documented in the 'Good Research Practice Guidelines' (Version 4, February 2021) issued by the Wellcome Sanger Institute. Consent for all human participants was obtained either explicitly for those specific to this study or through the terms of enrollment in UK Biobank, Foundation Medicine, *BAP1* TPDS and MSK-IMPACT studies. Participants were not compensated.

### HAP1-A5 cell model generation

A HAP1 *LIG4*⁻ cell line (HZGHC000759c005) with a 10-bp deletion in *LIG4* and its wild-type control were obtained from Horizon Discovery. HAP1 *LIG4*⁻ cells were transduced with pKLV2-EF1aBsdCas9-W (Addgene, 67978) lentivirus and selected for Cas9 expression with blasticidin at 10 μg ml⁻¹ (InvivoGen). The polyclonal Cas9-positive cells were then banked in liquid nitrogen (as reported previously[17]). To assess Cas9 activity, cells were transduced with a blue fluorescent protein (BFP)/green fluorescent protein (GFP) activity construct encoded by pKLV2-U6gRNA5(gGFP)-PGKBFP2AGFP-W (Addgene, 67980). A control construct, pKLV2-U6gRNA5(Empty)-PGKBFP2AGFP-W (Addgene, 67979), was also used with fluorescence-activated cell sorting (FACS) analysis performed on 10,000–20,000 cells for each condition with 405-nm and 488-nm channels for BFP⁺ and GFP⁺, respectively[57] (see Extended Data Fig. 1a and Supplementary Fig. 1a for representative gating). The polyclonal line was sorted for haploid cells using Hoechst 33342 at 10 μg ml⁻¹ and propidium iodide at 1 μg ml⁻¹ with incubation to allow gating of the haploid and viable cell fraction, respectively. A total of 0.5 × 10⁶ cells were sorted and returned to culture at 37 °C. The haploid polyclonal Cas9⁺ *LIG4*⁻ cells were expanded and banked at 5 × 10⁶ cells per vial.

To derive the monoclonal lines, haploid polyclonal Cas9⁺ *LIG4*⁻ cells from the bank were thawed, cultured and subcloned. Several clones were karyotyped by mFISH with 30 metaphase spreads examined per line. The monoclonal line, clone A5, showed optimal Cas9 activity, a haploid karyotype and no critical karyotypic abnormalities, with few chromosomal breaks or deletions. The parental line, KBM-7, comes from a chronic myelogenous leukemia line from a male patient and contained the following constitutional chromosomal rearrangements: 23,X,der(9)t(9;22),der(19)t(19;15;19),der(22)t(9;22). The 10-bp deletion in *LIG4* was confirmed in banks and clones by Sanger sequencing (Genewiz) (Extended Data Fig. 1d). The *LIG4* locus was amplified and the amplicon was sequenced using the primers GTAGTGACATTATGCAACTCAGCAG and TAGAGATGGAAAAGATGC-CCTCAAA. All HAP1 cell lines were cultured at 37 °C with 5% $CO_2$, in IMDM (with 25 mM HEPES and L-glutamine, Gibco), 10% FBS (Gibco) and 1% penicillin-streptomycin (Gibco), without supplementing 10 μg ml⁻¹ blasticidin unless specified in the text. HAP1-A5 is available from Horizon Discovery (Catalog ID: HZGHC-LIG4-Cas9).

### Ploidy and FACS analysis

Ploidy of cell line stocks was assessed during cell culture experiments (see Extended Data Fig. 1b for results). Early passage (P3, 9-day culture) after-thaw cells were used for the control line bank and wild-type HAP1 line assessment. On D3 and D19 (final passage), after-transfection cells were used to determine experimental effects on ploidy status, using reagents targeting exon 5 (sgRNA 5A and HDR template library 5A) (see Methods: 'Tissue culture, cell transfection and sampling' for transfection conditions). Metaphase-arrested cells were used to accurately assess ploidy in the cell populations. To perform metaphase arrest, 5 × 10⁶ to 8 × 10⁶ cells were seeded in T75 flasks. The next day, cells were treated with 0.2 nM nocodazole (Sigma) for 14 h at 37 °C. After incubation, the medium was removed and the cells were washed with 5 ml PBS (Sigma) and dissociated with 2 ml TrypLE Express (no phenol red, ThermoFisher) for 4 min at 37 °C, followed by the addition of 8 ml of medium. Samples were centrifuged at 250$g$ for 5 min and

resuspended in 8 ml PBS and then cell concentrations were determined using a Countess (ThermoFisher) with trypan blue staining (Gibco). Cells were diluted to 5 × 10⁶ to 9 × 10⁶ cells per 500 μl. Samples were added dropwise to 5 ml ice-cold 80% ethanol, while vortexing. Fixed samples were incubated on ice for 30 min and stored at 4 °C before FACS analysis. To prepare samples for FACS analysis, fixed cells were centrifuged at 845$g$ (3,000 r.p.m.) for 10 min and resuspended in 10 ml PBS for washing. Cells were then centrifuged at 376$g$ (2,000 r.p.m.) and resuspended in 1 ml of 0.1% Triton X-100 in PBS, counted on a Countess as above, and diluted to 1 × 10⁶ cells per ml, followed by staining with 10 μg ml⁻¹ 4′,6-diamidino-2-phenylindole (DAPI) (Sigma) for 30 min at room temperature. Samples were analyzed on an LSRFortessa (BD Biosciences) FACS machine with low flow rate settings, gating for singlet cells with DAPI signal assessed using the 405-nm channel. Analysis was performed on at least 1 × 10⁴ cells (selected in SSC-A versus FSC-A gate; see Supplementary Fig. 1b for representative gating).

To assess the transfection efficiency of HAP1-A5 cells, the plasmid pMin (5,275 bp; Supplementary Data 3) and a GFP-expressing plasmid pMax-GFP (Lonza, 3,486 bp) were transfected as described in Methods: 'Tissue culture, cell transfection and sampling'. A total of 7.5 μg of pMin and 15 μg of pMax-GFP were used for these transfections. Cells were dissociated 3 days after transfection. Live cells were incubated with 10 μg ml⁻¹ DAPI for 30 min at room temperature before FACS analysis of 50,000 cells. GFP-positive cells were measured using 405-nm (DAPI) and 488-nm (GFP) channels (see Extended Data Fig. 1c for results and Supplementary Fig. 1c for representative gating).

### Essentiality phenotyping

See Supplementary Methods 1–4.

### HDR library generation

See Supplementary Methods 5–7.

### sgRNA selection and cloning

All sgRNAs with a 20-nucleotide spacer across the *BAP1* gene were obtained through the CRISPR function within Geneious, with off-targets scored against the GRCh38 genome. sgRNAs for SGE were chosen based on a set of criteria, as previously reported[13]. The criteria included the selection of sgRNAs where synonymous changes in the PAM or protospacer were possible (to enable the inclusion of PPEs) and the sgRNA target site position was distal to splice junctions. In addition, sgRNAs were required to have no predicted off-targets in coding sequence (CDS), and >2 mismatches in any non-CDS off-target. In addition, sgRNA A and sgRNA B for the same target region were chosen to be nonoverlapping where possible and PPEs were selected to avoid codons where ClinVar or gnomAD variants have been reported, where possible. sgRNA selection for SGE was also evaluated through depletion dynamics in the targeted CRISPR screen of *BAP1* (Supplementary Method 3), with those demonstrating gradual depletion over time (~25% reduction in cell fitness between each of the first three time points) preferentially selected. We hypothesize that such sgRNAs exhibit cleavage events associated with locus-specific death, whereas general genotoxicity might be expected to result in immediate, strong depletion. sgRNA target sequence oligonucleotides were appended with 5′-CACC-3′ on the sense (CACCG if the target site did not start with a G, to allow optimal transcription from the U6 promoter used in the expression construct) and 5′-AAAC-3′ on the antisense oligonucleotide (with 3′-C appended if CACCG was used on the sense oligonucleotide). A volume of 1 μl of each oligonucleotide at 100 μM was phosphorylated and annealed with 0.5 μl polynucleotide kinase (PNK) and 1 μl of 10× T4 ligation buffer (NEB) in 10 μl with water and incubated at 37 °C for 30 min, followed by ramp down from 95 °C to 25 °C at 5 °C per min. Annealed oligonucleotides were diluted 1:200. Then, 20 μg of maxi-prepped pMin-U6-ccdb-hPGK-puro (see Supplementary Data 3 for the GenBank map) was digested with BbsI (NEB) in a 100-μl

reaction with 10 µl of 10× CutSmart Buffer (NEB), for 3 h at 37 °C. Gel purification of the 3,653-bp band was performed on a 0.8% agarose-TAE gel (Qiagen QIAquick Gel Extraction kit), with the sample divided between two wells. Gel purification (Qiagen QIAquick Gel Extraction kit) was followed by MinElute purification using the standard protocol (Qiagen MinElute kit), and concentration and purity were assessed by NanoDrop. A total of 1 µl of a 1:200 dilution of annealed sgRNA oligonucleotides was ligated into 50 ng of gel-purified pMin backbone (Qiagen QIAquick Gel Extraction kit), with 5 µl of 2× Quick Ligase Buffer (NEB) and 1 µl Quick Ligase (NEB), with incubation at room temperature for 10 min. The reactions were diluted 1:4 with water, and 2 µl was transformed into 50 µl of TOP10 competent cells (Invitrogen). Colonies were picked and cultured in ampicillin-supplemented (100 µg ml⁻¹) Luria–Bertani broth. Glycerol stocks were made and clones with the correct sequence were confirmed through Sanger sequencing (Eurofins) using guide_seq_f/r primers (Supplementary Table 9). Correct sgRNA clones were cultured in 125 ml Luria–Bertani broth with ampicillin inoculated with 5 µl glycerol stock and processed by maxiprep (Qiagen), as described in Supplementary Method 7, to produce transfection-quality plasmid DNA.

## Tissue culture, cell transfection and sampling
Vials of ~5 × 10⁶ HAP1-A5 cells were thawed and seeded into T75 tissue culture flasks (Corning) 9 days before transfection, in 15 ml of medium with blasticidin at 10 µg ml⁻¹ (InvivoGen), to select for cells with an integrated Cas9 construct. The cells were passaged at a 1:10 ratio 6 days before transfection and then expanded 3 days before transfection into multiple T150 flasks, in 35 ml blasticidin-containing medium (as above). Cell seeding stock with 15 ml of medium (without blasticidin) containing 8 × 10⁶ cells was prepared 1 day before transfection. A 15-ml suspension was seeded into each T75 flask required for transfection. On the day of transfection (day 0), the medium was changed 1 h before transfection. Xfect (Takara) was used to transiently transfect cells with an sgRNA and corresponding HDR library. A bottle of Xfect buffer was thawed from −20 °C at 4 °C overnight and then maintained at room temperature. Then, 7.5 µg of sgRNA and 15 µg of HDR plasmid library were added to an Eppendorf tube, and room-temperature Xfect buffer was added to a total of 750 µl. Then, 13.5 µl of freshly thawed Xfect polymer (0.6 µl per 1 µg plasmid DNA) was added and the tube was vortexed and incubated at room temperature for 10 min. Replicate transfection mixtures were pooled together and vortexed, and 750 µl was added dropwise into the medium of each replicate T75 flask. Flasks were incubated for 4 h at 37 °C, and then the medium was aspirated and replaced and the flasks were incubated overnight. On day 1 and day 2 after transfection, the medium was replaced with fresh medium containing blasticidin (10 µg ml⁻¹) and puromycin (3 µg ml⁻¹, InvivoGen) to select for the integrated Cas9 construct and the transfected sgRNA plasmid, respectively. On day 3 after transfection, dissociated cells were split at 50% into two T75 flasks in 15 ml medium (with blasticidin) and incubated overnight. On day 4 after transfection, one T75 flask per transfection from the day 3 split was collected: cells were washed once with 5 ml PBS (Sigma) and dissociated with 1.5 ml TrypLE Express (no phenol red, ThermoFisher). Flasks were incubated at 37 °C for 4 min and then 1.5 ml of medium was added (no antibiotics). The cell suspension was transferred to a Falcon tube, and the flask was washed with 5 ml of medium. The viable cell concentration of the suspension was determined using Countess (ThermoFisher) with trypan blue staining (Gibco). The suspension was then centrifuged at 300g for 3 min, washed with 1 ml PBS, recentrifuged and resuspended in PBS to give 6 × 10⁶ cells per ml, which was then aliquoted at 1 ml per Eppendorf tube, and the tubes were centrifuged at 300g for 3 min and stored at −80 °C until gDNA extraction. On days 5, 12, 17 and 19 after transfection, 5 × 10⁶ cells were passaged into new flasks. Sample collection, as described above for day 4, was performed on days 7, 10, 14 and 21 after transfection.

## Genomic DNA extraction and sequencing
See Supplementary Methods 8–12.

## Informatics to convert raw sequencing data to variant counts
CRAM files were processed using the QUANTS pipeline version 1.2.1.0 (https://github.com/cancerit/QUANTS/releases/tag/1.2.1.0) to generate sequencing quality control metrics and exact match sequence mapping to the designed VaLiAnT library sequence outputs.

QUANTS (https://github.com/cancerit/QUANTS) is a Nextflow[58] pipeline built using the nf-core framework. Nextflow version 22.04.3 was used to run QUANTS version 1.2.10 with Singularity[59], which provides the underlying software dependencies. As input, QUANTS takes raw sequencing data in CRAM format, an HDR template library and a sample mapping file that links the sequencing data file to a user-defined sample name. Within QUANTS, read trimming was performed using cutadapt[60] version 3.2 with Python 3.8.6 (--cores 4 -a [adapterR1]... [adapterR2]). Quality control plots and statistics were generated for both raw and trimmed sequencing data using FastQC version 0.11.9 and SeqKit[61] (stats) version 0.15.0 and collated using MultiQC[62] version 1.10.1. pyCROQUET (https://github.com/cancerit/pycroquet) 1.5.0 was used to calculate the frequency of each unique trimmed read sequence (library-independent counts). The library-independent counts were parsed using a bespoke script (https://github.com/cancerit/QUANTS/tree/1.2.1.0/modules/local/R/post_pycroquet_quantification) to determine the frequency of each HDR template. Reads not mapping to designed libraries were also quantified. Sequencing reads mapping to 'ref_seq', that is, wild-type GRCh38 reference, and 'pam_seq', that is, wild-type sequences with no variant but PAM/protospacer protection edits alone, were used to calculate editing efficiency. The total mapped and unmapped counts for each sample are available in Supplementary Table 1, and a summary is available in Supplementary Table 2.

## Calculation of variant abundance and functional scores
Sequencing data, analysis code and functional score data are available at https://github.com/team113sanger/Waters_BAP1_SGE.

Analysis after count generation was performed using R code (waters_bap1_sge_analysis.R) in R studio. Counts mapping to the desired HDR libraries were merged with VaLiAnT metadata outputs. VEP annotations were retrieved (and merged with variant count files) using VaLiAnT VCF outputs, which consider the mutational consequences of variants in the presence of PPEs. Variants with <10 counts across all time point replicates were removed. D4 counts were compared against plasmid library counts across target regions, with sgRNA–library pairs with strong positional effects removed from the analysis. DESeq2 (ref. 14) was used to calculate LFCs, dispersion estimates and statistical measures between D4 and D7, D10, D14 and D21 for variant counts. The generalized linear model of DESeq2 includes the requisite exponential function (the log-link between the central parameter of the negative binomial distribution and the linear regression term) to compute the log-linear growth rate. Therefore, an apparent growth rate was computed in DESeq2, with time across time points represented as a continuous variable, such that days 4, 7, 10, 14 and 21 were assigned the values 0, 3, 6, 10 and 17, respectively, and the 'continuous LFC' produced represents the change in variant abundance per unit time.

Pearson's correlation between replicates was assessed, with any replicate time points demonstrating poor correlation removed from the analysis. To normalize the LFC of variant change against the LFCs of variants that are not expected to change in the screen, the default DESeq2 scaling factor was replaced with a normalization factor calculated based on synonymous and intronic variants for each time point replicate. The calculated LFCs were median scaled by subtracting the median LFC of synonymous and intronic variants for each target region. Target regions were evaluated in separate screens using library A and library B HDR plasmid repair templates, and median-scaled LFCs were combined to produce a single value using an inverse variance of the

mean-weighted average[63]. Where target regions overlapped (in larger exons: 11.1, 11.2, 12.1, 12.2, 13.1, 13.2, 13.3, 17.1 and 17.2), median-scaled and weighted LFCs were again weighted based on variance to produce a final, single LFC for multiply observed variant LFCs, which is the 'combined LFC' or 'functional score'.

Standard errors for library A and library B LFCs were combined by inverse variance weighting to produce a single translated standard error ('combined standard error'). This was used to generate z-scores for each variant (z-score = combined LFC/combined standard error). P values for each variant were calculated from the z-score distribution using a two-tailed z-test, with subsequent BH FDR correction across all unique variants to produce FDR values. Variants with an FDR <0.01 and a negative functional score were classified as 'depleted' and those with an FDR <0.01 and a positive functional score were classified as 'enriched'. Variants with an FDR ≥0.01 were classified as 'unchanged'.

To avoid confusion, all 'functional classifications' were made using the 'functional score' and its corresponding FDR. In summary, the continuous LFCs produced from DESeq2, which are LFCs per unit time over all time points (separately calculated for libraries A and B), were adjusted by median scaling and then the library values were combined by weighted average to produce the functional score. The functional score FDR for each variant was calculated by computing a z-score and then performing a z-test, which was then adjusted for multiple testing using the BH procedure. The functional score and the FDR were then used to produce functional classifications.

The code and detailed description and justification for the functional score calculation process can be found at https://github.com/team113sanger/Waters_BAP1_SGE.

LFCs for target regions 2, 3, 8 and 13.1 were computed from a single library (A for 2, 3 and 13.1; B for 8). This is because SGE using sgRNA 2B and HDR plasmid library 2B resulted in poor editing (<12% of reads mapped to the library; Supplementary Table 1). Target region 13.1 B was excluded because of a strong positional effect during editing (Extended Data Fig. 3b), with a high number (>30%) of wild-type reads in D14 and D21 samples (Supplementary Table 1). SGE data using sgRNAs and HDR plasmid libraries at target regions 3B and 8A were not obtained because of profound cell death at the transfection stage.

**Pathogenic and benign truth sets for ACMG evaluation**
See Supplementary Method 13.

**UK Biobank PheWAS analysis**
See Supplementary Method 14.

**Statistics and reproducibility**
Tests used throughout the study are stated in figure legends and the main text, and assumptions for specific tests were met in all cases with the Shapiro–Wilk test for normality applied where appropriate. Calculations to obtain the FDR using the BH method to correct for multiple testing were performed where appropriate. Where comparisons between FDR values are made, the exact FDR value (technically the q value) is reported because these are scaled values that have meaning relative to other values in the same test. Where the FDR value (or P value) was extremely low, the value was reported as $q < 2.2 \times 10^{-16}$ (or $P < 2.2 \times 10^{-16}$) because any value below this is not computationally meaningful.

All editing experiments were performed in triplicate. Sample sizes were determined by the number of variants detected in each sample/maximum number of carriers with necessary clinical data; therefore, no statistical method was used to predetermine sample size. The following were excluded from analyses because of a likely low signal-to-noise ratio: variants with fewer than ten counts detected across all replicate time points, replicates with low editing efficiency/strong positional effects in editing (Methods: 'Calculation of variant abundance and functional scores') and variants in PPE codons (Results: 'Functional

analysis of gene architecture and conservation'). The experiments were not randomized. The investigators were not blinded to allocation during the experiments and the outcome assessment. Data collection and analyses were not performed with blinding to the conditions of the experiments.

**Histological analysis of the primary cutaneous melanoma**
Immunohistochemistry was performed on paraffin sections using an automated immunohistochemistry staining system (Ventana BenchMark ULTRA, Ventana Medical Systems) with an alkaline phosphatase red detection kit by Erasmus MC Pathology Research and Trial Service.

The sections were deparaffinized and heated by heat-induced epitope retrieval for 64 min at 97 °C to retrieve antigens. Subsequently, the sections were incubated for 32 min at 37 °C with anti-BAP1 antibody (SC-28383; 1:50 dilution; Santa Cruz Biotechnology) as the primary antibody. Target amplification was performed, followed by incubation with hematoxylin II counterstain for 8 min. Blue-coloring reagent (Ventana Medical Systems) was used as an additional counterstain.

Non-neoplastic cells located next to the tumor tissue in the sections served as internal positive controls for BAP1 expression because BAP1 is expressed in most normal tissues. The BAP1 staining was inspected using a score from 0 (complete BAP1 loss) to 2 (no BAP1 loss), and localization of the immunohistochemical signal was classified as either nuclear or cytoplasmatic. The melanoma was clinically scored to have complete BAP1 loss, with weak positive staining in the non-neoplastic inflammatory infiltrate in the upper right of the micrograph section.

**SpliceAI score analysis**
We adapted the original SpliceAI code (https://github.com/Illumina/SpliceAI) to compute the scores for multinucleotide variation. We also made necessary changes in the code to extract scores for reference and alternative nucleotides along with acceptor and donor gains and losses. All SpliceAI scores for different variants were generated with a window size of 500 and the mask value set to 0 and using the GRCh38 reference genome and its annotation. We have shared the customized SpliceAI code at https://github.com/team113sanger/Waters_BAP1_SGE.

**Reporting summary**
Further information on research design is available in the Nature Portfolio Reporting Summary linked to this article.

## Data availability
Functional scores and classification data are available at https://github.com/team113sanger/Waters_BAP1_SGE.

Functional scores and classifications for all unique variants are provided in Supplementary Data 1. Additional annotations are available in Supplementary Data 2.

FASTA and CRAM files generated in this study for HDR plasmid libraries and edited genomic DNA libraries are available through the European Nucleotide Archive (ENA) under accession PRJEB64778.

Raw counts generated through the QUANTS pipeline and VaLiAnT and VEP annotation files are available through BioStudies under accession S-BSST1222.

Mapped counts and experimental and bioinformatics methods are accessible through MaveDB under accession urn:mavedb:00000662. Source data are provided with this paper.

## Code availability
The analysis code is available at https://github.com/team113sanger/Waters_BAP1_SGE.

A digital object identifier accession for code is available at https://doi.org/10.5281/zenodo.10489733 (ref. 64).

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

## Acknowledgements

This work was funded by the Wellcome Trust (220540/Z/20/A, awarded to D.J.A.) and Cancer Research UK (EDDPGM-Nov22/100004, awarded to D.J.A. and C.T.). The funders had no role in study design, data collection and analysis, decision to publish or preparation of the manuscript. We would like to thank G. Findlay (Francis Crick Institute), L. Parts and B. Lehner (Wellcome Sanger Institute, WSI) and members of the Atlas of Variant Effects Alliance (https://www.varianteffect.org/) for useful discussions. We also thank N. van der Stoep (Department of Clinical Genetics, LUMC) and R. Verdijk (LUMC and Erasmus MC) for their family pedigree analysis and pathology/histology slides, respectively. We are grateful to D. Gitterman (WSI), J. Urbanova, O. Dovey, M. Byrne (formerly WSI, now bit.bio) and G. Turner (formerly WSI, now Public Health Scotland) for their work on the generation of the HAP1-A5 cell line. We thank S. Walpole (University of Queensland) for providing published germline variant data and W. Huber (EMBL) for discussions on the use of the DESeq2 package.

## Author contributions

A.J.W. designed experiments, performed experiments, co-wrote code, analyzed the data and co-wrote the manuscript. D.J.A. supervised the study and co-wrote the manuscript. T.B.-S., D.S. and S.O. performed experiments. V.O. processed the SGE data using QUANTS. Y.Z. and J.R.B.P. performed UK Biobank analyses. M.N. and R.v.D. provided and analyzed patient pedigree data. J.-E.M. provided and analyzed foundation medicine data. M.T. analyzed CRISPR–Cas9 essentiality screen data. H.K.T. and E.J.R. wrote R and Python code and coedited the manuscript. S.S.G. and M.E.H. supervised H.K.T. and E.J.R. E.D. processed data for repository access, P.G. performed analyses to obtain SpliceAI values, C.F.R., H.H. and C.T. analyzed data and quantified evidence for the ACMG variant classification framework.

## Competing interests

The authors declare no competing interests.

## Additional information

**Extended data** is available for this paper at https://doi.org/10.1038/s41588-024-01799-3.

**Correspondence and requests for materials** should be addressed to Andrew J. Waters or David J. Adams.

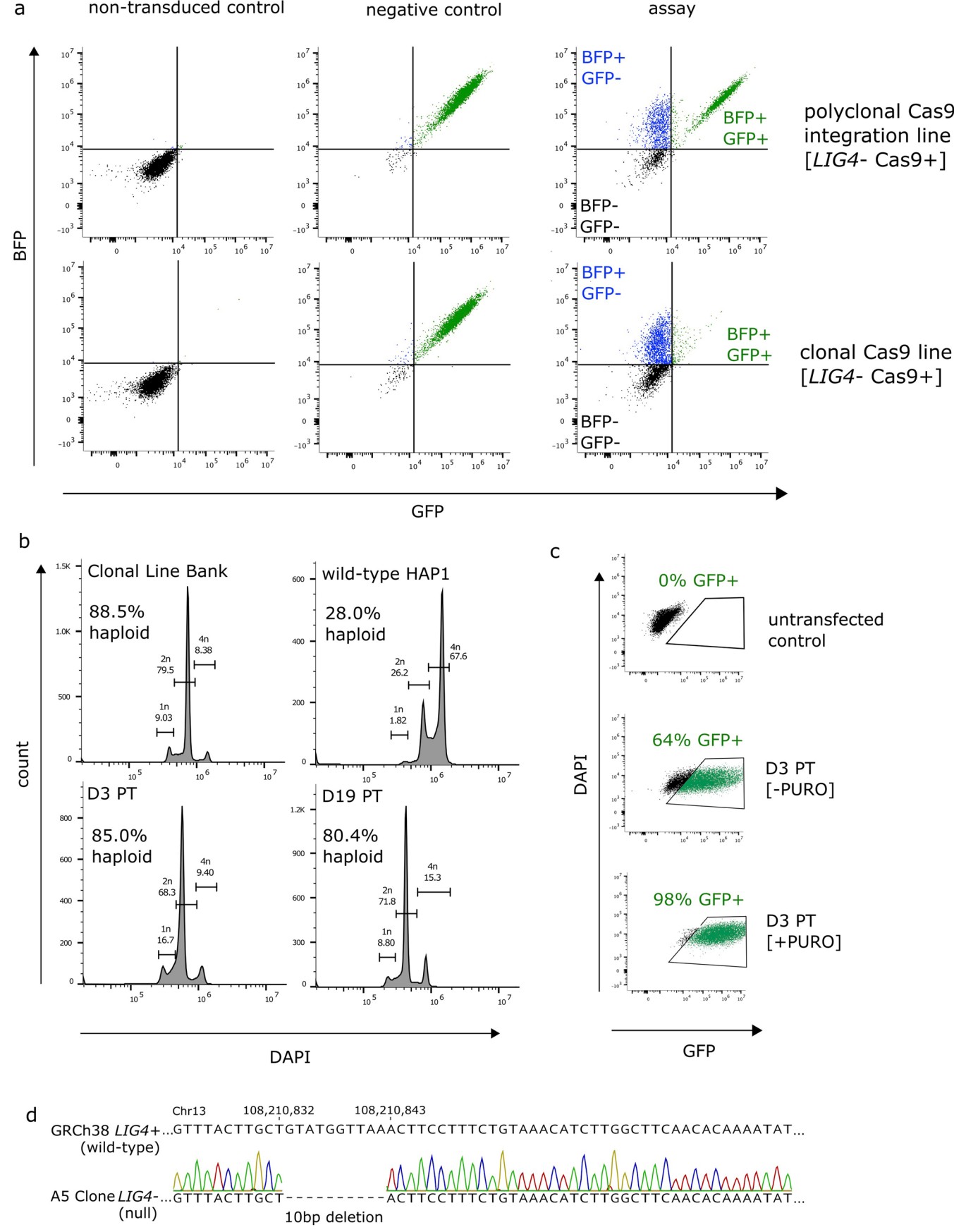

**Extended Data Fig. 1 | See next page for caption.**

**Extended Data Fig. 1 | Clonal HAP1-A5 line and experimental protocol optimization improves editing rate. a**. FACS data comparing the activity of Cas9 in a polyclonal Cas9+ *LIG4-* HAP1 cell line (top row) with a monoclonal line (HAP1-A5) derived from the same polyclonal line (bottom row) at 72hrs post-transduction. BFP+ GFP+ cells (green) gated in top right quadrant are cells in which Cas9 has failed to inactivate GFP by editing. The Cas9-inactive fraction (green) is significantly reduced, and the Cas9-active fraction (blue) is significantly increased in the clonal line. The negative control contains no sgRNA targeting to GFP coding sequence. Non-fluorescent cells in black. 10,000-20,000 cells were assessed per sample. See Supplementary Fig. 1a for representative gating. **b**. Nocodazole-based metaphase arrest and DAPI staining was performed on cells edited at exon 5 of *BAP1* at D3 post-transfection (PT) and D19 (final passage) PT. Unsorted and untransfected wild-type HAP1 cells were included as a control, as were untransfected clonal line cells. A slight increase in ploidy is seen as a result of transfection and culturing. 1n are un-arrested haploid cells, 2n and 4n are metaphase-arrested haploid and diploid cells, respectively. 10,000-20,000 cells were assessed per line. See Supplementary Fig. 1b for representative gating. **c**. X-fect (Takara) based transfection of 6 million clonal HAP1-A5 cells using the pMax-GFP (Lonza) 3486bp construct, which is of a similar to size to a typical HDR library, and exon 5 sgRNA-A plasmid. Pre-selection sees a transfection rate of 64% and post-selection with puromycin (+PURO) showing 98% GFP+ cells (green) at Day 3 (D3) post-transfection (PT). GFP- cells in black. 50,000 cells assessed per sample. See Supplementary Fig. 1c for representative gating. **d**. Sanger sequencing of HAP1-A5 clone at *LIG4* locus confirms expected frameshifting, 10 base-pair deletion in CDS, creating a null allele (Horizon Bioscience).

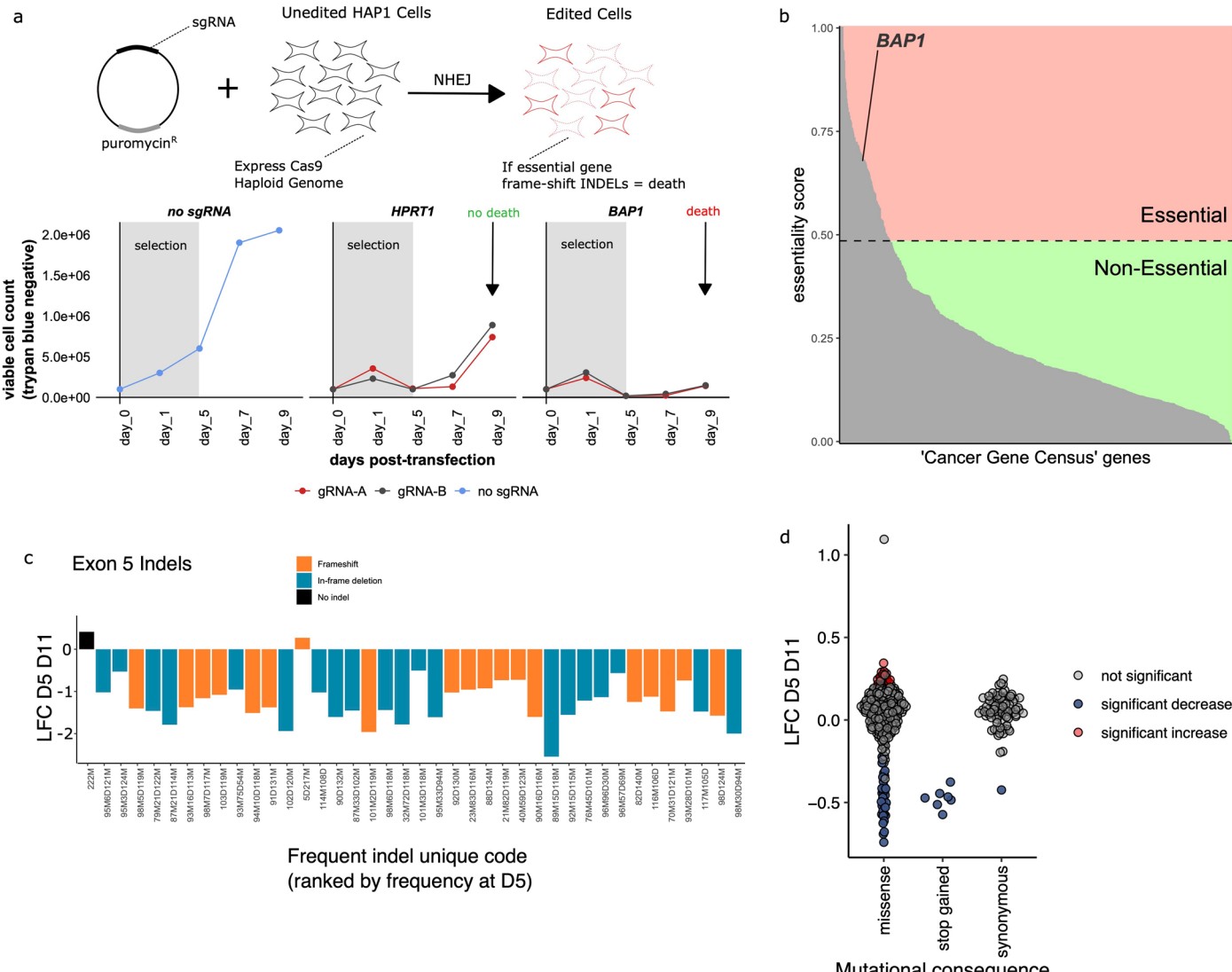

**Extended Data Fig. 2 | BAP1 is essential in HAP1 cells and amenable to SGE. a.** Experimental scheme and viable cell counts in polyclonal Cas9+ *LIG4*- HAP1 cells after targeting loci with sgRNAs, based on assumed non-homologous end joining (NHEJ). Cells targeted with *BAP1* sgRNAs do not strongly increase in number between day 7 and day 9 post-transfection whereas sgRNAs to a known HAP1 non-essential gene, *HPRT1* do. A plasmid not expressing a sgRNA which does not cleave the genome, shows log growth between day 5 and day 7, plateauing between day 7 and day 9 due to confluency. This demonstrates that cutting the genome at a non-essential locus has some genotoxic effects. **b.** Hart et al.[65], HAP1-derived Bayes factor essentiality data was scaled across all genes included in the Cancer Gene Census[66] (as of 2020). The dashed line marks the scaled value for the threshold Bayes factor (>6) above which there is a ~90% probability of essentiality. *BAP1* position relative to other genes is shown. **c.** Cells edited in a similar manner to that shown in 'a' were sampled at day 5 and day 11 and indels counted and ranked by frequency, frameshift and in-frame deletions deplete over time suggesting essentiality of the *BAP1* locus. **d.** A pilot SGE experiment using a minimal library at exon 5 shows screening by SGE at the *BAP1* locus works as expected, with stop-gained variants depleting, synonymous variants generally not changing and missense variants showing a spectrum of variant effect.

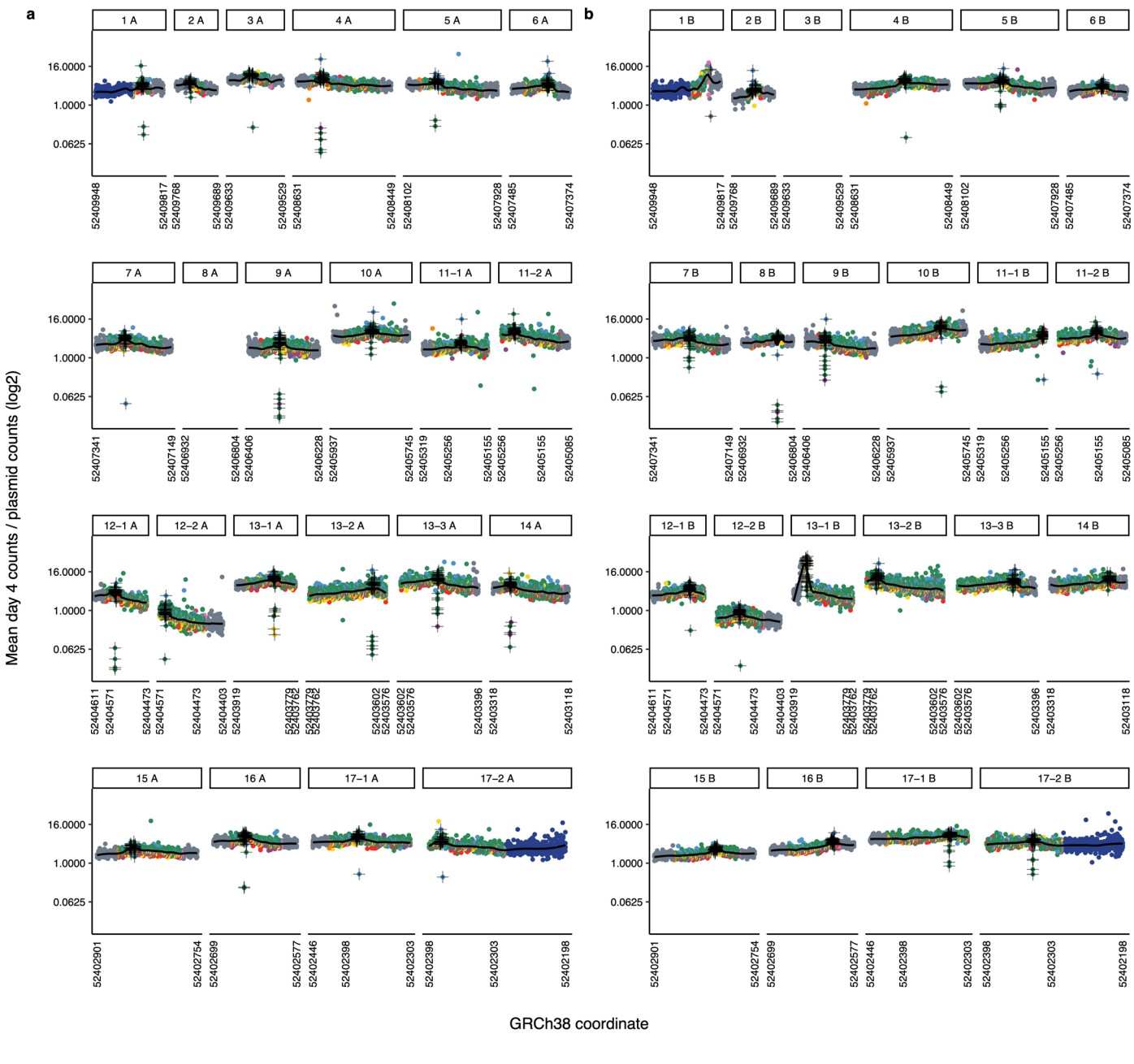

**Extended Data Fig. 3 | Positional effect modelling of editing across all SGE regions reveals minimal bias. a**. Counts for Day 4 Library A genomic editing events divided by plasmid library A counts for each variant as an indication of the relative rate of variant incorporation across editing regions. Variants that are edited at codons with PAM/protospacer protection edits (PPEs) are highlighted with a black cross. As expected, variants which revert the PPEs to the wild-type nucleotide sequence cannot prevent re-cutting and show depletion. Points are coloured by VEP mutational consequence. There is no extreme bias in the representation of stop-gained and frameshift variants at Day 4, indicating that compromising variants have yet to deplete by the baseline. A slight increase in incorporation is seen at most PPE positions, this is likely due to enhanced protection from re-editing relative to the rest of the edited regions. The rate of incorporation is slightly reduced at increasing distance from the Cas9 cleavage site (which is in close proximity to PPE labelled codons). Exon 8 A is missing as this guide resulted in profound cell death at the transfection stage. **b**. Counts for Day 4 Library B genomic editing events divided by plasmid library B counts. An extreme positional effect is observed for region 13-1 B, this data was excluded from analyses. Exon 3 B is missing as this guide resulted in profound cell death at the transfection stage.

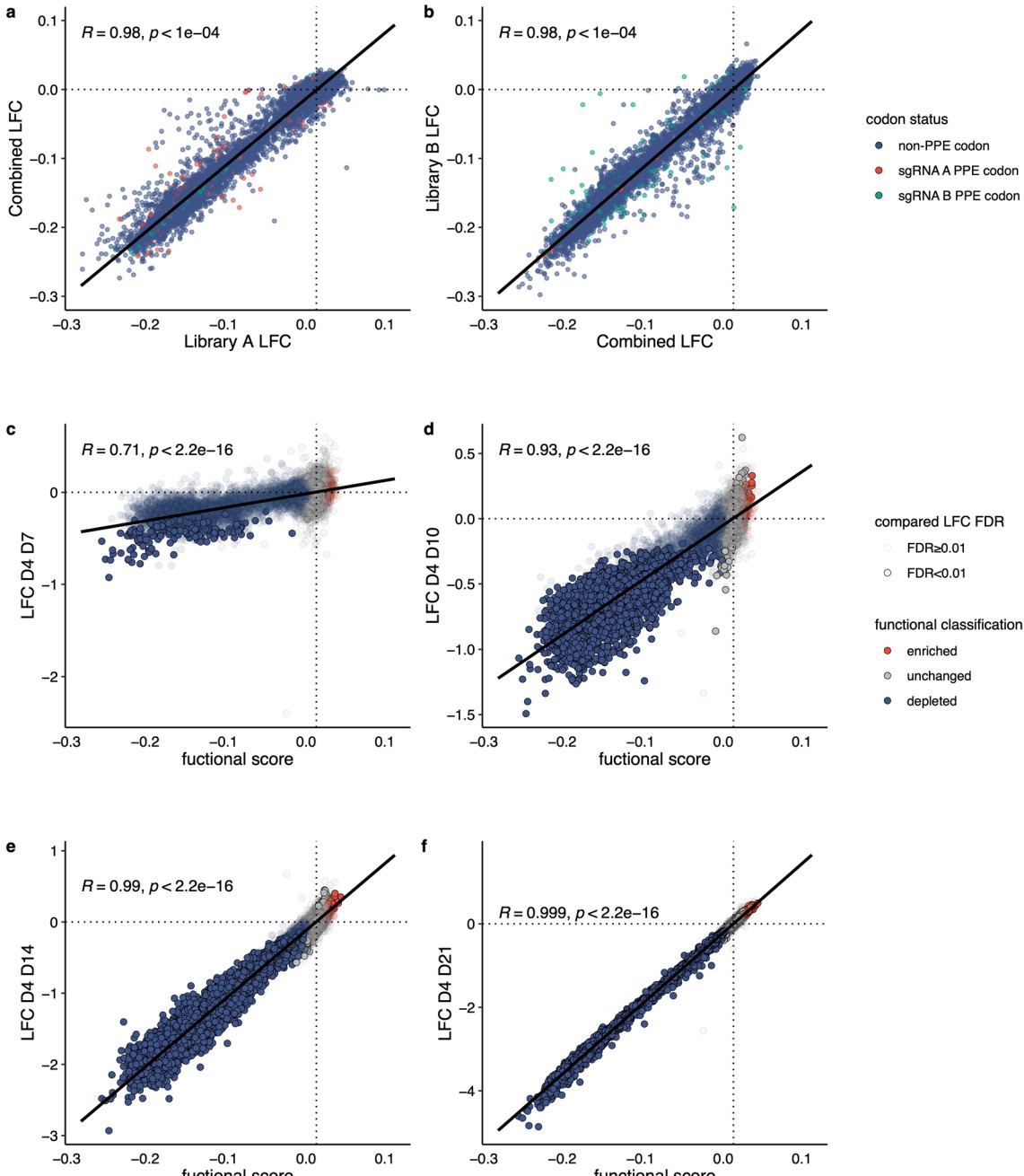

**Extended Data Fig. 4 | Comparison of LFCs between separate libraries or timepoints reveals kinetics of variant change. a.** and **b.** show the combined LFC calculated using an inverse-weighted mean of Library A and Library B LFCs plotted against either Library A (**a.**) or Library B (**b.**) LFC. LFCs are highly concordant as expected, based on Pearson's correlation coefficient (*R*) and two-tailed t-test *p*<2.2e-16. Variants that fall within codons that also contain fixed PAM/protospacer protection edits (PPEs, coloured red for Library A and green for Library B) are more likely to be outliers to the correlation, these codon variants were not weighted in Combined LFC calculations with the variant LFC derived from the library where there is not a PPE. **c.** LFC between D4 and D7 plotted against functional score (combined LFC) and coloured by functional classification, significant changes (FDR<0.01) for the LFC plotted on the y-axis

are shown as non-transparent. **d.** As 'c' but comparing LFC between D4 and D10, a subset of variants marked in transparent shade can be seen which do not deplete significantly by D10, those that are coloured transparent dark blue, do become significantly depleted by D21. Those that are not transparent and dark blue are significantly depleted by D10, suggesting different variant depletion kinetics. **e.** As in 'c' and 'd' but comparing LFC between D4 and D14. There is a linear relationship and high correlation between the penultimate timepoint and functional score, suggesting little difference in the kinetics of depletion. **f.** Functional score compared with LFC D4 D21 shows an extremely high correlation, suggesting very little/no difference between these two related metrics of variant change. Pearson's correlation coefficient is shown (*R*) and two-tailed t-test *p*<2.2e-16 for all plots.

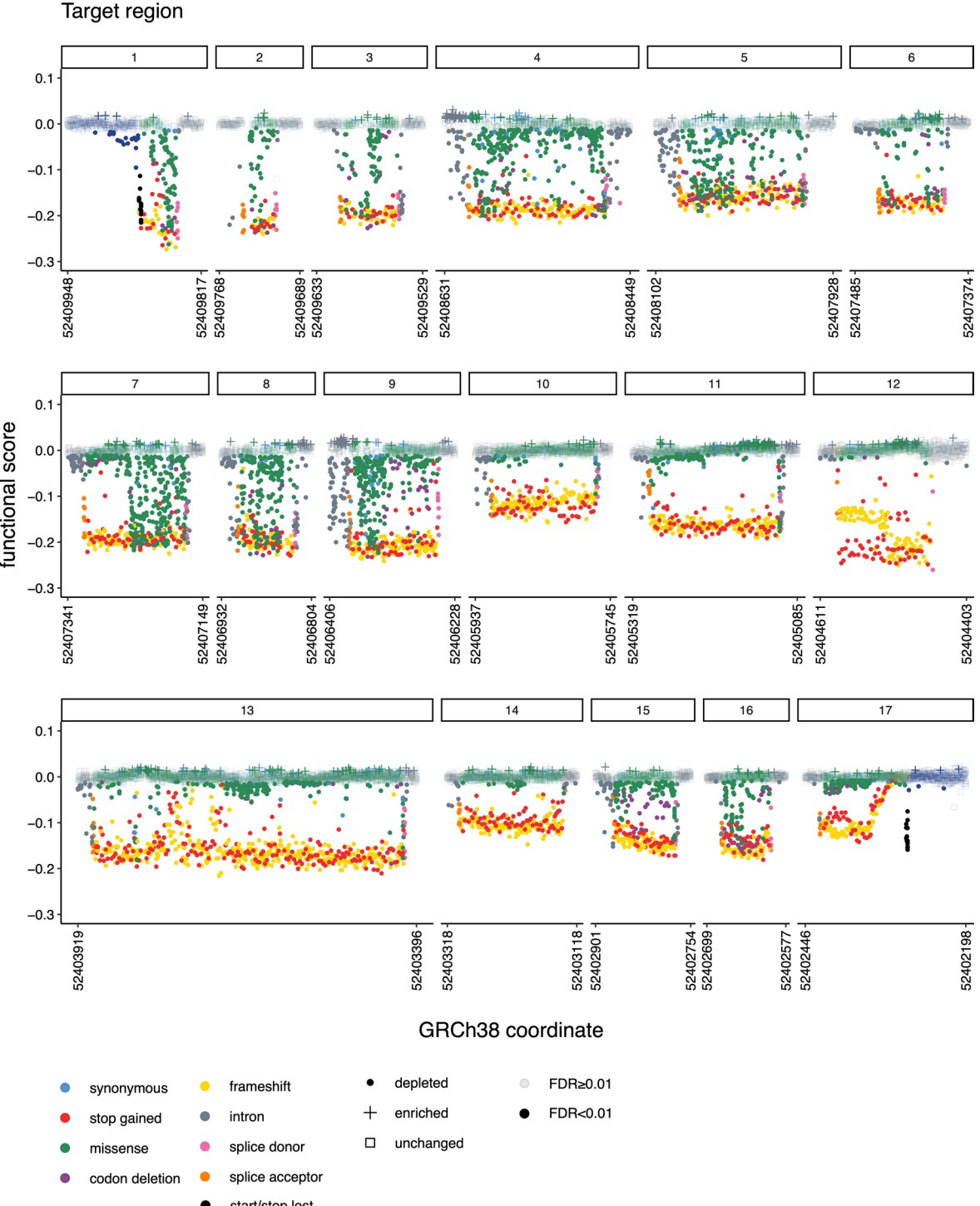

**Extended Data Fig. 5 | Nucleotide-level map of *BAP1* variant effect.** Functional scores for all 18,108 variants assessed, separated by exon region with GRCh38 genomic coordinates of regions targeted shown in the direction of transcription (*BAP1* is transcribed from the negative strand). Frameshift, splice donor/acceptor and stop-gained variants deplete throughout the length of the gene, distinct regions of missense intolerance can be seen. Data points are coloured by VEP mutational consequence, functional classifications are distinguished by shape and significance by transparency.

Target region

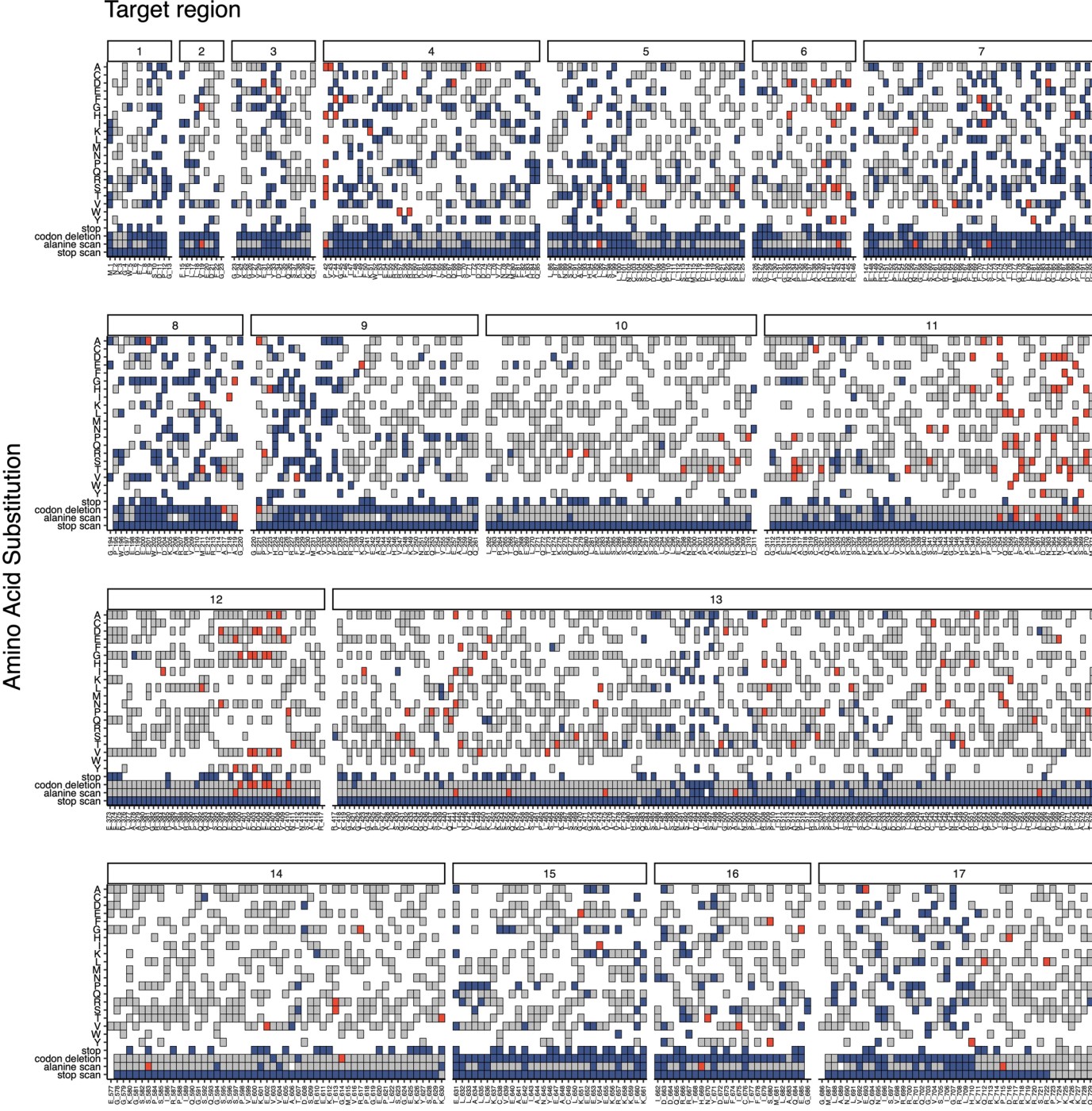

functional classification

- 🟥 enriched
- ⬜ unchanged
- 🟦 depleted

**Extended Data Fig. 6 | See next page for caption.**

**Extended Data Fig. 6 | Amino acid-level map of *BAP1* variant effect.** Heat map to show functional classification for protein-level changes. This includes missense changes created by SNVs, alanine scan, stop scan and codon deletions. Synonymous substitutions are also shown. Distinct regions of mutational intolerance can be seen. Stop-gained variants that do not significantly deplete can be seen in 7 terminal codons of exon 17. Non-depleting stop-gained variants generated through the 'stop scan' function (NNN>TGA) are also seen at K630 in exon 14 and S482 in exon 13, however SNVs leading to stop-gained mutations (NNN>TAG) do deplete at these loci. White space is due to either low-count filtering at the QC stage, because SNV level saturation does not produce all amino acid level substitutions, or because redundant SNVs for the same missense change resulting in discordant functional classifications were removed in this plot (n=158).

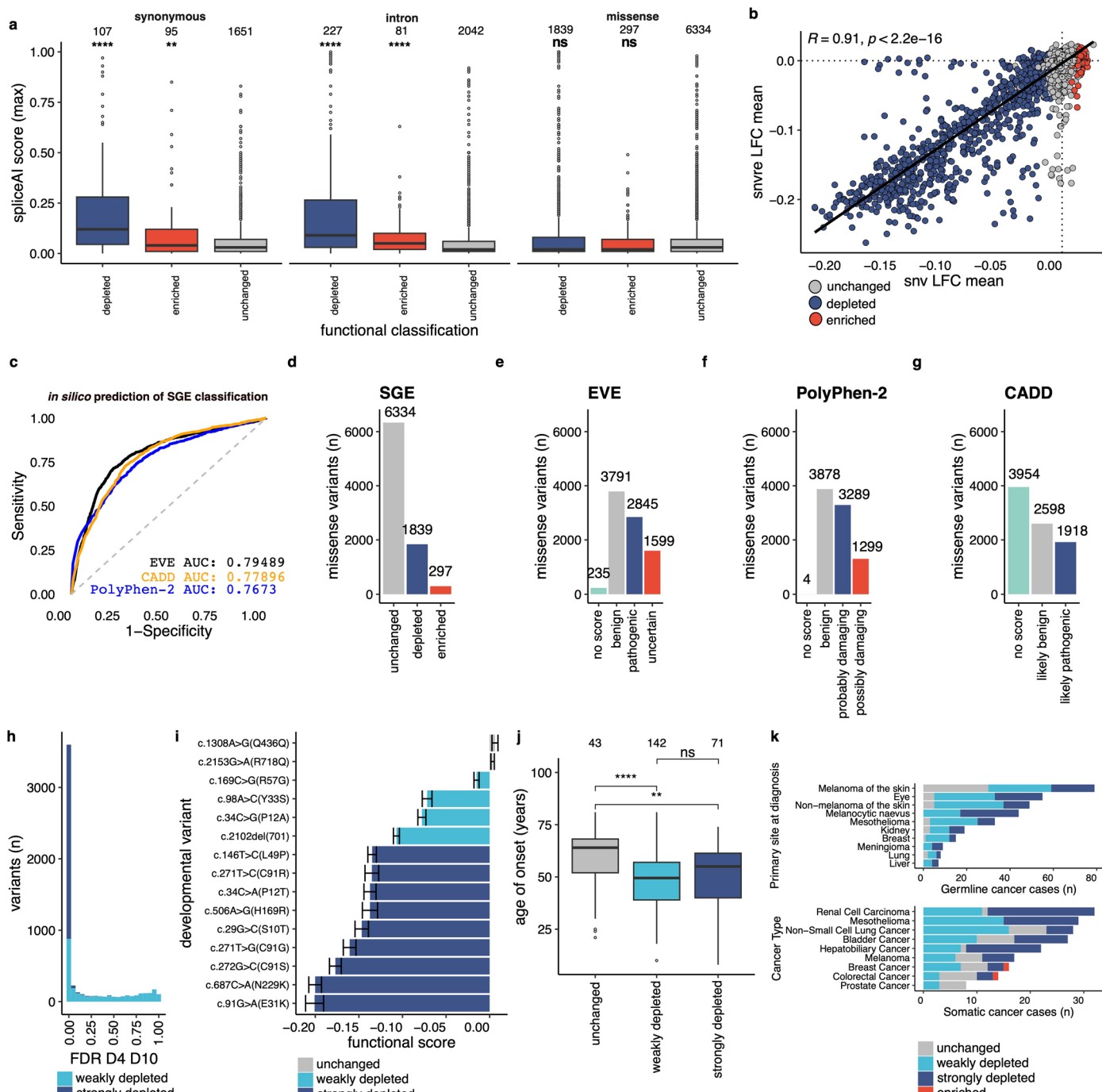

**Extended Data Fig. 7 | See next page for caption.**

**Extended Data Fig. 7 | Internal and external metric correlations. a**. SpliceAI values are significantly higher for depleted and enriched synonymous/intron variants vs unchanged synonymous/intron variants (two-sided Mann-Whitney-Wilcoxon Test, ****$p$<0.0001, **$p$<0.01, depleted/enriched synonymous vs unchanged synonymous $p$<2.2e-16 and $p$=0.009, respectively; depleted/enriched intron vs unchanged intron $p$<2.2e-16 and $p$=2.9e-05, respectively). Non-splice region missense variants are not significantly different between depleted/enriched missense vs unchanged missense $p$=0.15 and $p$=0.11, respectively (two-sided Mann-Whitney-Wilcoxon Test). Box shows interquartile range, horizontal line median maximum spliceAI score, whiskers show maximum and minimum values, outliers as points. **b**. Average functional score for 4,619 'control' variants (3,993 missense, 188 stop-gained and 438 synonymous) generated using redundant codons (snvre LFC mean) created in VaLiAnT[13], compared to average functional score for the same variant generated by a SNV, coloured by SNV classification. Pearson's Correlation Coefficient $R$ and two-sided t-test $p$ value shown. **c**. SGE classifications used as standards to compare *in silico* predictors: 8,470 non-splice region missense, 6,334 unchanged and 1,839 depleted variants (297 enriched were excluded). EVE, CADD and PolyPhen-2 reported SGE

classifications with 79.5%, 77.9% and 76.7% accuracy, respectively. **d-g**. Bar charts show variants by classification for 8,470 missense variants. **h**. Strongly depleted variants show earlier depletion than most weakly depleted variants, observed by LFC D4 D10 FDR. **i**. Known[9,24] *BAP1* developmental variants show strong and weak depletion (c.1308A>G and c.2153G>A are unchanged). Functional score (bar) and DESeq2-calculated standard error (+/-error bars) from 3 biological replicates. **j**. Age of cancer onset for 256 carriers of *BAP1* germline variants reported in a clinical analysis of 181 carrier families[26]. Strongly or weakly depleted variant carriers show no difference in age of onset. Carriers of variants in either depleted category have an earlier age of onset compared to unchanged variant carriers (two-sided Dunn's BH FDR, ****$q$=5.91e-05, **$q$=0.0074, ns $q$=0.17). Box shows interquartile range, horizontal line median age of onset, whiskers show maximum and minimum values, outliers as points. **k**. Top, germline cancer variants[26] by primary diagnosis site (where tumor site had >5 associated variants), coloured by functional classification. Bottom, MSK-IMPACT[43] somatic variants by cancer type (where cancer type had >5 associated variants). Strongly and weakly depleted classifications are distributed throughout cancer sites/types.

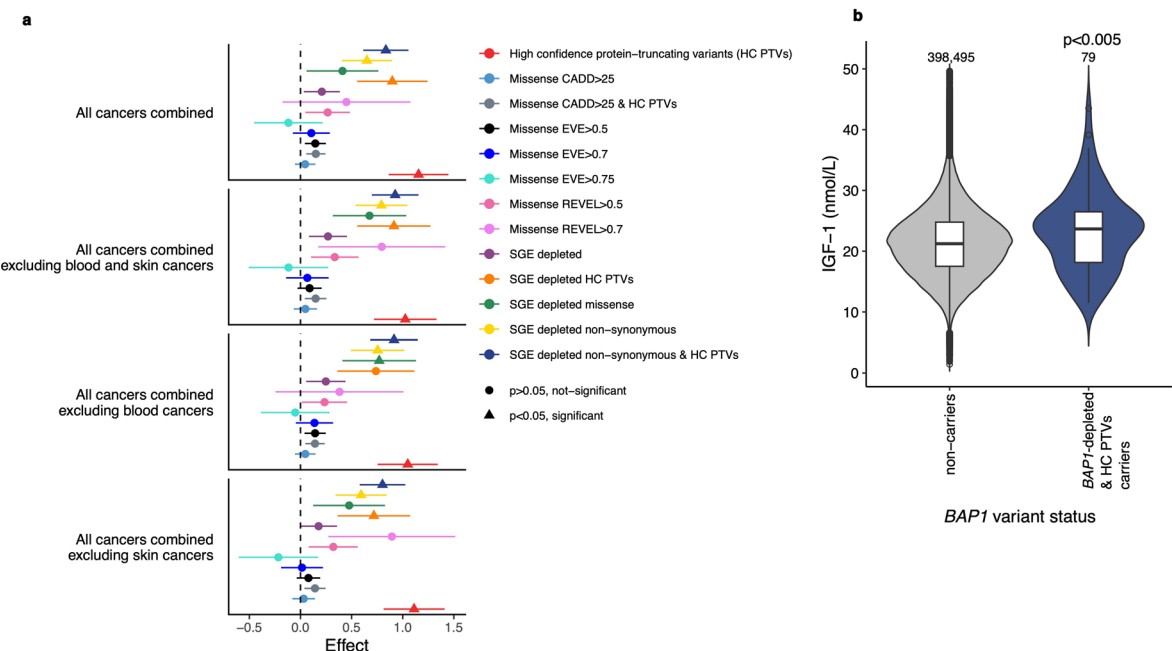

**Extended Data Fig. 8 | UKBB rare variant burden masks across cancer types and IGF-1 levels are increased in UKBB *BAP1* HC PTVs. a**. Rare variant burden test masks shown by colour for *BAP1* variants in UKBB across cancer phenotype masks. The significance was calculated according to corrected *p*-value determined by generalized linear modelling (see Supplementary Method 14), is signified by triangle (significant) or circle (not significant). In order to make comparisons, we also created masks separate from SGE-depletion, including; all *BAP1* HC PTVs in UKBB (red), *BAP1* missense variants with CADD scores > 25 (light blue) and *BAP1* HC PTVs plus missense variants with CADD scores > 25 (grey), missense with EVE score>0.75/>0.7/>0.5 (very high (cyan), high (royal blue), and moderate pathogenicity (black) bins, respectively), and REVEL score>0.7/>0.5 (high (light pink) and moderate (dark pink) pathogenicity thresholds, respectively). Significance and effect differ between cancer types. For all cancers, SGE depleted non-synonymous variants (yellow) show a significant effect and are therefore associated with an increased cancer risk. No *in silico* tools assessed

allow for a significant association with cancer to be achieved, most notably in 'All cancers combined excluding blood', where SGE depleted missense (green) are significantly associated with cancer, allowing for direct comparison with missense only prediction. The number of carriers for each rare variant burden test mask in each cancer phenotype mask can be seen in Supplementary Table 7. Error bars define the +/- standard error of the regression model effect. **b**. UKBB *BAP1* HC PTV variant carriers combined with SGE-depleted non-synonymous variant carriers (total n=79 out of 96 carriers, shown in Supplementary Table 7, have IGF-1 values) have a significantly higher median blood concentration of Insulin-like Growth Factor 1 (IGF-1) compared to non-carriers (n=398,495), *p*=0.004 (two-sided Mann-Whitney-Wilcoxon Test). Violin plot coloured by *BAP1* variant status for clarity. Box shows interquartile range, horizontal line the median IGF-1 blood concentration (nmol/L), whiskers show maximum and minimum values that are not outliers, outliers as circles.

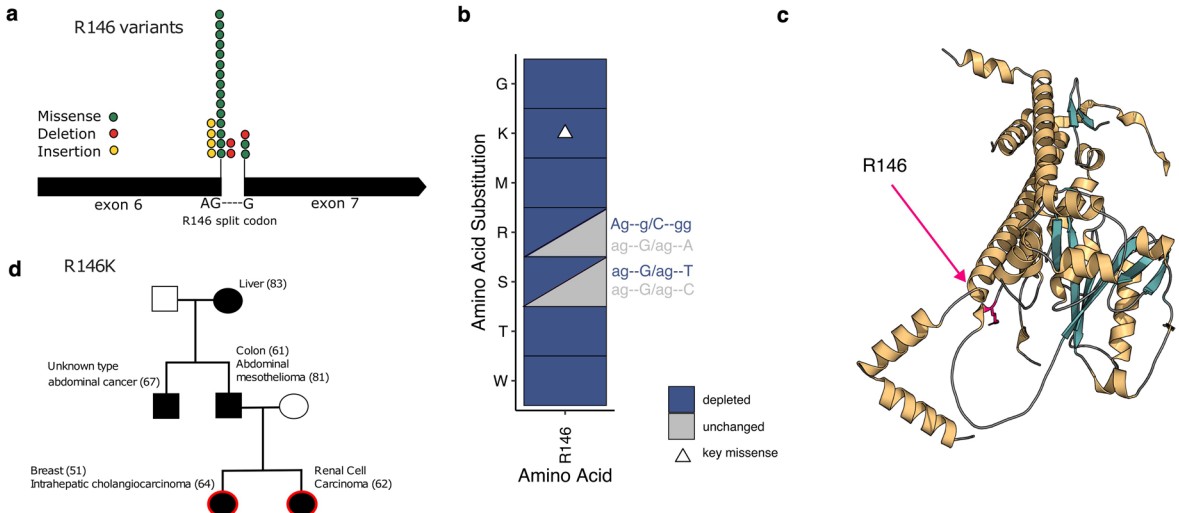

**Extended Data Fig. 9 | SGE resolves pathogenicity of variants in a recurrently mutated *BAP1* codon identified through large-scale next generation sequencing of tumors.** Analysis of R146 variants in the Foundation Medicine cohort. We searched the Foundation Medicine database to identify novel *BAP1* variants. This analysis revealed multiple variants in the R146 codon of *BAP1* including missense and frameshift events. **a**. Variants are shown against a *BAP1* gene structure and are split between different nucleotides within the codon spanning exons 6 and 7. **b**. All missense variants found at position R146 in the Foundation Medicine database are significantly depleted in SGE as seen by heatmap, the white triangle highlights R146K, of interest in 'd'. Two codons were measured redundantly at the nucleotide level, and have different classifications (triplet codes in blue/grey), this includes synonymous changes, indicating disruption to splicing over the split-codon. **c**. All altered residues at 146 fall into a side chain proximal to the catalytic core (R146 residue highlighted in pink). **d**. A *BAP1* R146K (c.437G>A) variant, observed in the Foundation Medicine database is a confirmed germline variant. A patient presenting with cholangiocarcinoma at 64 and their sister (a renal cell carcinoma (RCC) patient at 62) were found to carry the variant (red circles). Other first and second-degree relatives were reported to present with RCC, mesothelioma, melanoma, liver cancer, colon cancer, and a cancer of unknown primary **e**. Summary of patient demographics and variant details for Foundation Medicine *BAP1* accessions.

# Reporting Summary

## Statistics

For all statistical analyses, confirm that the following items are present in the figure legend, table legend, main text, or Methods section.

| n/a | Confirmed | |
|---|---|---|
| ☐ | ☒ | The exact sample size ($n$) for each experimental group/condition, given as a discrete number and unit of measurement |
| ☐ | ☒ | A statement on whether measurements were taken from distinct samples or whether the same sample was measured repeatedly |
| ☐ | ☒ | The statistical test(s) used AND whether they are one- or two-sided<br>*Only common tests should be described solely by name; describe more complex techniques in the Methods section.* |
| ☐ | ☒ | A description of all covariates tested |
| ☐ | ☒ | A description of any assumptions or corrections, such as tests of normality and adjustment for multiple comparisons |
| ☐ | ☒ | A full description of the statistical parameters including central tendency (e.g. means) or other basic estimates (e.g. regression coefficient) AND variation (e.g. standard deviation) or associated estimates of uncertainty (e.g. confidence intervals) |
| ☐ | ☒ | For null hypothesis testing, the test statistic (e.g. $F$, $t$, $r$) with confidence intervals, effect sizes, degrees of freedom and $P$ value noted<br>*Give P values as exact values whenever suitable.* |
| ☒ | ☐ | For Bayesian analysis, information on the choice of priors and Markov chain Monte Carlo settings |
| ☒ | ☐ | For hierarchical and complex designs, identification of the appropriate level for tests and full reporting of outcomes |
| ☐ | ☒ | Estimates of effect sizes (e.g. Cohen's $d$, Pearson's $r$), indicating how they were calculated |

*Our web collection on statistics for biologists contains articles on many of the points above.*

## Software and code

Policy information about availability of computer code

| Data collection | No software was used for data collection. Proprietary Illumina base-calling software/firmware on Miseq and HiSeq platforms was employed. |
|---|---|
| Data analysis | Please see 'Code Availability' section in the manuscript for GitHub and DOI for custom code used in the study.<br><br>Design and Analysis Software:<br><br>VaLiAnT version 1.0.0<br>FlowJo version 10.10<br>Geneious version 2023.0.4<br>R version 4.1.3 (2022-03-10) in RStudio Version 1.4.1106<br>QUANTS pipeline version 1.2.1.0<br>Nextflow version 22.04.3<br>Cutadapt version 3.2 with Python 3.8.6<br>FastQC version 0.11.9<br>SeqKit60 (stats) version 0.15.0<br>MultiQC61 version 1.10.1<br>pyCROQUET version 1.5.0 |

For manuscripts utilizing custom algorithms or software that are central to the research but not yet described in published literature, software must be made available to editors and reviewers. We strongly encourage code deposition in a community repository (e.g. GitHub). See our web collection on softwareandcode.

R Software versions:

FSA_0.9.4
pROC_1.18.4
ggpubr_0.6.0
DESeq2_1.34.0
DEGreport_1.30.3
ggridges_0.5.4
stringr_1.5.0
dplyr_1.1.2
tidyr_1.3.0
ggplot2_3.4.2
tidyverse_2.0.0

For manuscripts utilizing custom algorithms or software that are central to the research but not yet described in published literature, software must be made available to editors and reviewers. We strongly encourage code deposition in a community repository (e.g. GitHub). See the Nature Portfolio guidelines for submitting code & software for further information.

## Data

Policy information about availability of data

All manuscripts must include a data availability statement. This statement should provide the following information, where applicable:
- Accession codes, unique identifiers, or web links for publicly available datasets
- A description of any restrictions on data availability
- For clinical datasets or third party data, please ensure that the statement adheres to our policy

BAP1 variant functional scores and classifications are freely available for all non-profit uses and are available here:
https://github.com/team113sanger/Waters_BAP1_SGE and as Supplementary Data 1 and Supplementary Data 2.

FASTA and CRAM files generated in this study for HDR plasmid libraries and edited genomic DNA libraries are available through the European Nucleotide Archive (ENA) accession: 'PRJEB64778'

Raw counts generated through the QUANTS pipeline, and VaLiAnT and VEP annotation files,
are available for all non-profit uses through the BioStudies accession: 'S-BSST1222'

Mapped counts, experimental and bioinformatics methods are accessible through the MaveDB accession: 'urn:mavedb:00000662'.

GRCh38 used for all co-ordinates
gnomAD version 3
ClinVar downloaded 04/09/2023 (https://ftp.ncbi.nlm.nih.gov/pub/clinvar/vcf_GRCh38/archive_2.0/2023/) file= 'clinvar_20230903.vcf.gz'
ClinVar downloaded 20/09/2023 (https://ftp.ncbi.nlm.nih.gov/pub/clinvar/vcf_GRCh38/archive_2.0/2023/) file= 'clinvar_20230917.vcf.gz'

## Research involving human participants, their data, or biological material

Policy information about studies with human participants or human data. See also policy information about sex, gender (identity/presentation), and sexual orientation and race, ethnicity and racism.

| Reporting on sex and gender | na |
|---|---|
| Reporting on race, ethnicity, or other socially relevant groupings | na |
| Population characteristics | na |
| Recruitment | One patient is included in the study in a pedigree analysis. Permission was sought for inclusion, findings have been explained in person through genetic counseling and the patient has agreed to sign necessary declarations to allow for publication. |
| Ethics oversight | Department of Clinical Genetics, Leiden University Medical Center, Leiden, The Netherlands |

Note that full information on the approval of the study protocol must also be provided in the manuscript.

# Field-specific reporting

Please select the one below that is the best fit for your research. If you are not sure, read the appropriate sections before making your selection.

☒ Life sciences   ☐ Behavioural & social sciences   ☐ Ecological, evolutionary & environmental sciences

For a reference copy of the document with all sections, see nature.com/documents/nr-reporting-summary-flat.pdf

# Life sciences study design

All studies must disclose on these points even when the disclosure is negative.

| | |
|---|---|
| Sample size | No explicit sample size was decided upon a priori, rather we sought to assess all ~18,000 unique oligonucleotide species created through VaLiAnT, to comprehensively saturate all coding sequence and near-exon non-coding sequence of BAP1.<br><br>In order to maintain library complexity at the genome editing stage, 8 million cells were seeded 1 day before transfection. 6 million cells were sampled for timepoint-replicates gDNA extractions. 5 million cells were passaged at each timepoint to maintain culture through the screen.<br><br>Some variants were edited into the genome in multiple instances through over-lapping target regions and HDR repair libraries - subsequent to editing and during analysis steps, separate editing events were combined into a a single metric value through weighted mean calculations. |
| Data exclusions | Excluded time-point replicates and reasons for exclusion can be see in Supplementary Table 1 'analysis_status' column. Exclusion was either due to strong positional effect during editing or library indexing error.<br><br>Variants with fewer than 10 counts (generated through the QUANTS pipeline from CRAM file analysis) were excluded during analysis steps. |
| Replication | All transfections were performed in triplicate. Separate triplicates were maintained in culturing and sampling. When transfections failed, all three replicates failed, when transfections were successful all three replicates were successful. Successful transfection was determined by high cell survival post transfection and puromycin selection compared to non-transfected controls which were always included.<br><br>Data from 40/44 total HDR libraries were processed (library A and library B) at the analysis stage, with high reproducibility seen between Library A and Library B experiments (Fig.4a). |
| Randomization | Variants were edited into genomic loci in multiplex. ~1000 unique variants were integrated at each transfection, with variants related by proximity. gDNA libraries A and B for the same target regions were grouped into the same HiSeq sequencing run. All time point replicates for a target region were grouped together in the same HiSeq sequencing run. All tiled target regions (multiple target regions for larger exons) were grouped such that all gDNA libraries were grouped by exon in the same HiSeq sequencing run. Allocation of target regions to sequencing runs after these groupings were made was dictated by the order in which target regions were selected to be experimented upon (ie. when gDNA libraries were ready to be sequenced), which was essentially random. Groupings were as follows:<br><br>1 A+B, 3 A, 7 A+B, 9 A+B, 15 A+B<br>5 A+B, 10 A+B, 14 A+B<br>11.1 A+B, 11.2 A+B, 12.1 A+B, 12.2 A+B<br>2 A+B, 4 A+B, 17.1 A+B, 17.2 A+B<br>13.1 A+B, 13.2 A+B, 13.3 A+B<br>6 A+B, 8 B, 16 A+B<br><br>All plasmid HDR libraries were sequenced on the same MiSeq sequencing run. |
| Blinding | Functional scores and classification calculations were performed en masse, independently of known pathogenicty status. Assumptions were made about the likely minimal functional effect of synonymous and intronic variants in normalization processes, during which variants were systemically identified for inclusion using VEP. Blinding was not relevant to analyses or experiments in that all functional effect classifications and conclusions (including those for synonymous and intronic variants) emanated a posteriori after empirical data collection and analysis. |

# Reporting for specific materials, systems and methods

We require information from authors about some types of materials, experimental systems and methods used in many studies. Here, indicate whether each material, system or method listed is relevant to your study. If you are not sure if a list item applies to your research, read the appropriate section before selecting a response.

## Materials & experimental systems

| n/a | Involved in the study |
|---|---|
| ☐ | ☒ Antibodies |
| ☐ | ☒ Eukaryotic cell lines |
| ☒ | ☐ Palaeontology and archaeology |
| ☒ | ☐ Animals and other organisms |
| ☒ | ☐ Clinical data |
| ☒ | ☐ Dual use research of concern |
| ☒ | ☐ Plants |

## Methods

| n/a | Involved in the study |
|---|---|
| ☒ | ☐ ChIP-seq |
| ☐ | ☒ Flow cytometry |
| ☒ | ☐ MRI-based neuroimaging |

## Antibodies

| | |
|---|---|
| Antibodies used | SC-28383, Santa Cruz Biotechnology, Dallas, Texas, USA |
| Validation | The histological sections of human primary melanoma samples allow for internal validation as tumour cells (expected BAP1 negative) |

| Validation | and immune infiltrate (expected BAP1 positive) can be distinguished with 1:50 dilutions of primary antibody, seen in at least three replicates. Data sheet from Santa Cruz can be found here: https://datasheets.scbt.com/sc-28383.pdf |

# Eukaryotic cell lines

Policy information about cell lines and Sex and Gender in Research

| Cell line source(s) | A HAP1 LIG4- cell line (HZGHC000759c005) with a 10bp deletion in LIG4, and its wild-type control were obtained from Horizon Discovery. This line was transduced with Cas9 lentivirus to create a polyclonal line from which single cell clones were derived to the create the HAP1 A5 cell line. |
| Authentication | Authenticated by karyotype: mFISH using 30 metaphase spreads. Sanger sequencing over LIG4 lesion to confirm LIG4-. Cas9 activity analysis and metaphase-arrest ploidy assessment. |
| Mycoplasma contamination | Cells were tested for Mycoplasma by commercial PCR and confirmed to be negative. |
| Commonly misidentified lines (See ICLAC register) | No commonly misidentified cell lines were used in the study. |

# Flow Cytometry

## Plots

Confirm that:

☒ The axis labels state the marker and fluorochrome used (e.g. CD4-FITC).

☒ The axis scales are clearly visible. Include numbers along axes only for bottom left plot of group (a 'group' is an analysis of identical markers).

☒ All plots are contour plots with outliers or pseudocolor plots.

☒ A numerical value for number of cells or percentage (with statistics) is provided.

## Methodology

| Sample preparation | To assess the Cas9 activity: cells were transduced with a BFP/GFP activity construct 'pKLV2-U6gRNA5(gGFP)-PGKBFP2AGFP-W' (Addgene, 67980), a control construct was also used 'pKLV2-U6gRNA5(Empty)-PGKBFP2AGFP-W' (Addgene, 67979) with FACS analysis performed on 10,000-20,000 cells for each condition with 405nm and 488nm channels for BFP+ and GFP+, respectively.

To assess ploidy: Metaphase-arrested cells were used to accurately assess ploidy in the cell population. Day 3 and day 21 post-transfection cells were used (see Online Methods: 'Tissue culture, cell transfection and sampling' for transfection conditions). 5-8 Million cells were treated with 0.2nM of nocodazole (Sigma) for 14hrs at 37°C. Cells were dissociated and ethanol fixed.

To assess transfection efficiency: an empty vector of the sgRNA expression plasmid 'pMin-U6-ccdb-hPGK-puro' (5275bp) and a GFP-expressing plasmid 'pMax-GFP' (Lonza, 3486bp), were transfected into HAP1-A5 cells as described in Methods: 'Tissue culture, cell transfection and sampling'. 7.5μg of pMin and 15μg of pMax-GFP were used for the transfection. Cells were dissociated at 3 days post-transfection. The live cells were incubated with 10μg/mL of DAPI for 30min at room temperature before proceeding to FACS analysis. |
| Instrument | LSRFortessaTM (BD Biosciences) FACS machine with low flow rate settings (ploidy and transfection)

CytoFLEX for Cas9 Activity |
| Software | FlowJo version 10.10 |
| Cell population abundance | For ploidy and Cas9 activity: Analysis was performed on at least 1 x10^4 cells (selected in SSC-A vs FSC-A gate). For transfection efficiency analysis was performed on at least 5 x10^4 cells (selected in SSC-A vs FSC-A gate). |
| Gating strategy | For Cas9 activity: gating for singlet cells with BFP (405nM) and GFP (488nM). Supplementary Fig. 1a. For ploidy: gating for singlet cells with DAPI signal at 405nM channel. Supplementary Fig.1b For transfection efficiency: GFP positive cells were determined by gating with 405nM (DAPI) and 488nM (GFP) channels. Supplementary Fig.1c. |

☒ Tick this box to confirm that a figure exemplifying the gating strategy is provided in the Supplementary Information.

