## [Peer Review File · Nature Genetics]

Peer Review Information

Manuscript Title: Saturation genome editing of BAP1 functionally classifies somatic and germline variants

Corresponding author name(s): Dr David Adams, Dr. Andrew Waters

Reviewer Comments & Decisions:

Decision Letter, initial version:

1st Aug 2023

Dear Dr Adams,

hope this email finds you well and I apologize for the delayed response.

Your Article, "Comprehensive saturation genome editing of BAP1 to functionally classify somatic and germline variants" has now been seen by 2 referees. You will see from their comments copied below that while they find your work of considerable potential interest, they have raised quite substantial concerns that must be addressed. In light of these comments, we cannot accept the manuscript for publication, but would be very interested in considering a revised version that addresses these serious concerns.

We hope you will find the referees' comments useful as you decide how to proceed. If you wish to submit a substantially revised manuscript, please bear in mind that we will be reluctant to approach the referees again in the absence of major revisions.

To guide the scope of the revisions, the editors discuss the referee reports in detail within the team, including with the chief editor, with a view to identifying key priorities that should be addressed in revision and sometimes overruling referee requests that are deemed beyond the scope of the current study. In this case, we would like you to address Reviewers' comments in full. We would like to highlight that we agree with Reviewer#3 about the BAP-1 VUS re-classification, it would be important to take in consideration their suggestions in this regard. Please do not hesitate to get in touch if you would like to discuss these issues further.

If you choose to revise your manuscript taking into account all reviewer and editor comments, please highlight all changes in the manuscript text file. At this stage we will need you to upload a copy of the manuscript in MS Word .docx or similar editable format.

*2) If you have not done so already please begin to revise your manuscript so that it conforms to our Article format instructions, available [here](http://www.nature.com/ng/authors/article_types/index.html). Refer also to any guidelines provided in this letter.

[redacted]

If you wish to submit a suitably revised manuscript we would hope to receive it within 6 months. If you cannot send it within this time, please let us know. We will be happy to consider your revision so long as nothing similar has been accepted for publication at Nature Genetics or published elsewhere. Should your manuscript be substantially delayed without notifying us in advance and your article is eventually published, the received date would be that of the revised, not the original, version.

Nature Genetics is committed to improving transparency in authorship. As part of our efforts in this direction, we are now requesting that all authors identified as 'corresponding author' on published papers create and link their Open Researcher and Contributor Identifier (ORCID) with their account on the Manuscript Tracking System (MTS), prior to acceptance. ORCID helps the scientific community achieve unambiguous attribution of all scholarly contributions. You can create and link your ORCID from the home page of the MTS by clicking on 'Modify my Springer Nature account'. For more information please visit please visit

<http://www.springernature.com/orcid>>www.springernature.com/orcid.

Thank you for the opportunity to review your work.

Sincerely,
Chiara

Chiara Anania, PhD
Associate Editor
Nature Genetics
<https://orcid.org/0000-0003-1549-4157>

Referee expertise:

Referee #1:

Referee #2:cancer predisposition

Referee #3:mutational scanning, genomics

Reviewers' Comments:

Reviewer #1:
None

Reviewer #2:
Remarks to the Author:

A: The key result of the paper is a very extensive work which experimentally evaluates almost pre-emptive all single nucleotide variants in the BAP1 gene. In addition, a novel correlation to the IGF-1 concentration has been identified in pathogenic variant carriers

B: SGE has been used regarding other genes, but additional methods has been used in the design of this paper, and it is original with experimental evaluation of the BAP1 gene. It is a very important paper, which will be extensively used by researchers and contributes to further understanding of the function of the BAP1 gene, and it is also original with the correlation to IGF-1. The paper will give a substantial contribution very much needed in a clinical setting, as the majority of particular missense BAP1 variants are classified as a VUS currently.

C: The experimental work is very comprehensive and previous papers using SGE in other genes, have been shown to have falsely classified some variants due to additional genetic alterations incorporated in the cells, which then have generated false results for these specific variants. I believe this has been tried to be addressed in this paper as a library A and B, with three replications of the results have been generated and outliers removed.

D: The relevant statistic analysis have been used and the setup is appropriate also regarding uncertainties.

E: All in all, it is a very robust paper. It has been validated to published clinical data, and thereby shows robust and reliable results

F:

I would also like the authors to address

1. Previous papers regarding SGE in other genes, have been shown to have falsely classified some variants due to additional genetic alterations incorporated in the cells, which then have generated false results for these specific variants. I believe this has been tried to be addressed in this paper as a library A and B, with three replications of the results have been generated and outliers removed. However, when discrepancies have been identified not related to PPE, I would like the authors to add more information about the variants
2. In addition, the authors have in general compared the data to Clinvar pathogenic variants, and the previously published pathogenic variants regarding the 181 families. But I would like the authors to give more information about the variants that are altered with this classification, both regarding the current supporting clinical data, and if the SGE data has had discrepancies between library A and B, or outliers have been removed from the data. This also refers to the depleted variants currently only observed in Gnomad, where are of the observed patients, and which version of Gnomad (non-cancer) has been used.
3. Would you add information regarding splice prediction to all synonymous depleted variants
4. Line 17: Most inherited variants are not rare, it is only because you have previously filtered for rare variants. Could you please rephrase
5. Line 34: I assume these variants are from clinvar. Please state if the are from clinvar or other places and state time for the clinvar search
6. Line 38: please add "surveillance recommendations and risk-reduction strategies", as very few strategies will prevent cancer.
7. Line 42. Please state if this is only mesotheliomas or if other specific cancer types also have been shown to benefit from these treatments. If not alter to mesotheliomas
8. Line 60: Please specify the types of variants examined or state that not all variants have been examined, ie insertions, larges indels etc.
9. Line 83 "HDR repair" is a pleonasm (homology directed repair repair). Please remove repair. This phrase has been used multiple places in the manuscript.
10. Line 152: Please add if this is a scientific significant change. Comparing to the frequency in the background population is very strong. In addition, please add a comparions between LOF variants frequency in the background population and constrained missense variants (are they less constrained or as the LOF variants)
11. Line 171: please specify the number of the specific variants (LOF, missense etc)
12. Line 190: Please add any relevant published or available clinical data regarding these patients/families, which potential can corroborate the change in classification.
13. Line 217: Please add the average age of cancer onset in the different cohorts with STD depletion and non-depletion, to see if there is a younger onset of cancer in the depletion cohort.
14. Line 226: could you add information if there are differences in IGF_1 levels in depleted variants carriers with cancer compared to without cancer
15. Figure 3c lower panel, add significance level
16. Extended data figure 9: Alter the pedigree to standard. It looks like husband and wife where it presumably are siblings.

G: yes

H: Very clear and in a clear context. Remarks to abstact is in the above suggestions.

Reviewer #3:

Remarks to the Author:

In this manuscript, the authors present a large variant effect data set, gathered using saturation genome editing, for the cancer-associated gene Bap 1. The main attractions of the work are that the data is of beautiful quality; that functional data for Bap 1 is needed because of the burden of germline and somatic variants as well as the relationship of the gene to multiple diseases; and that the analysis includes data from various human cohorts like UKB that yield some surprising results. In particular the association of Bap1 function with IGF-1 levels in people and tumor cells is fascinating and really showcases the power of combining comprehensive functional data and large human cohorts. However, the work has some serious flaws. Most importantly, I feel that the analysis of the saturation genome editing data that was performed was far from optimal and, in places, didn't make sense. Additionally, much of the analyses following the SGE itself felt cursory and, in places, were factually incorrect.

Major comments:

-The authors generated data from a cell growth assay, measuring variant frequency at numerous time points. But, instead of doing a straightforward growth rate analysis they execute a number of confusing comparisons of specific time points to compute log-fold changes, then they compute something kind of like a growth rate and, finally, resort to inappropriate dimensionality reduction methods. The conclusions they reach aren't wrong, they're just not as good as they could be and quite confusing to follow. The work would be much better if the authors picked one metric for scoring each variant (in my opinion an apparent growth rate computed by log-linear regression, with an appropriate error model) and deviated from it only when necessary (e.g. to explore "growth rate changes," see comments below).

-The authors make statistical assumptions that are unsupported by the data, most especially assuming that various quantities are normally distributed when they are obviously not. There is no reason to do this and it causes problems for the interpretations that follow.

-The manuscript contains some serious errors. The best example of this is the explanation of how Bap1 SGE data could be used to reinterpret germline VUS. As written, the manuscript is wrong, and would mislead readers into thinking the authors did something they did not actually do.

-The work does not deliver on many of the promises made in the abstract. The best example of this is the IGF-1 finding, which is billed in the abstract as "Our analyses demonstrated that disruptive germline BAP1 variants are significantly associated with higher circulating levels of the mitogen IGF-1, suggesting a possible pathological mechanism and therapeutic target." This is true - they did find an association and it's very cool. But, they don't explore either mechanism or the possibility of a therapeutic. This finding at least needs quite a bit more exposition. The authors never explain what IGF-1 is or why it might relate to cancer or make a therapeutic target. Ideally, they would do more follow up experimentation and analysis but at least a solid discussion backing up the assertion in the abstract is needed.

-The manuscript contains a lot of undefined jargon which will cause problems for generalist readers.

Detailed comments

Line 72 - The authors screen many sgRNAs and pick some. Since they are presenting improvements for the SGE method they should explain why they picked the sgRNAs they did. It's not clear from

looking at Fig 2a - some chosen sgRNAs deplete very quickly (e.g. have a low frequency even at D7) whereas others don't. In other cases, what seem like clearly superior choices are ignored (e.g. in exon 3 the chosen sgRNA has less depletion from D7->28 than other not chosen sgRNAs). I know they point to reference 12, but they should briefly explain here, too.

Line 104/Figure 2e - The authors compute a z-score which is inappropriate for data that is, clearly, not normally distributed. It seems to me that a linear rescaling of the data would achieve much the same result without putting the data on a scale (z-score) that has a meaning (e.g. each unit is 1 SD) for normally distributed data but doesn't have that meaning here because the data are not normally distributed.

Line 104 - The authors use a z-test to identify significantly depleted variants. The core assumption of the z-test is that the data are normally distributed. Do we know that the replicate scores are actually normal? Would not a t-test (at least) be more appropriate since n is small and the assumption that score distributions are normal seems fraught? Maybe even better would be to use a nonparametric (e.g. Wilcoxon rank-sum test, etc) test.

Line 104 - Related to the above comment, it is difficult to tell exactly what tests were performed, here and in the methods. Did the authors really compute a test for every variant at each time point relative to D4? If so, why? It seems to me that they are creating a big multiple testing problem that, after correction, will result in many important variant depletions being nonsignificant. Why do they need to use an all-by-all statistical design (e.g. all variants x all timepoints)? Wouldn't just one timepoint suffice?

Line 104 - The authors collected beautiful time series data. Why not use the time series data to compute an apparent growth rate for each variant and then test for differences in growth rate? Many MAVE-type experiments have done exactly this, and it would ease the interpretation and statistical issues. The authors themselves state "scores between later timepoints were found to be linearly related" suggesting this approach would work well.

Line 148 - The authors assert that in silico predictors "were found to be highly sensitive, but not specific compared" to SGE. However, the cited extended figure does not contain any quantitative assertions about specificity and sensitivity of either SGE or the predictors (instead it just has plots where, because of overplotted points, it is impossible to appreciate sensitivity and specificity). The authors should make a quantitative comparison (e.g. by computing sensitivity and specificity) and summarize it in the main text.

Line 150 - The authors assert that there are important differences between the frequency of functional classes of variants in gnomad. They should do a statistical test to support this assertion (e.g. chi-squared, etc). More importantly, figure 3d is shown as a bar chart but that hides the fact that the assertions are based on very small numbers (e.g. just two enriched variants are in gnomad). Perhaps each bar could be labeled with n=X?

Line 167 - Editorial comment: It's great that the authors did a good experiment, but the type of inter-replicate correlations they see are pretty standard for this type of experiment these days. I'm not sure that it warrants three main figure panels.

Line 171 - What is a "PPE containing codon?" Nature Genetics is a generalist journal and this kind of

jargon should be avoided. Moreover, I don't know what a PPE codon is and why it is reasonable to remove them. The authors should explain.

Line 171 - The authors used ClinVar data but don't (as far as I could discover) explain how it was filtered. ClinVar is notoriously noisy, and the authors should remove variants from submitters from $<1^*$ at a minimum. They should also explain, in the methods, when ClinVar was accessed for their analysis.

Line 175 - Ah, so the authors did calculate a growth rate (or something very like it)! The methods are confusing, but I think their "continuous LFC" is basically a growth rate computed from all the time points. Unsurprisingly, it performs better than any single LFC.

Line 177 - The authors say that a LFC is "more intuitive" than the continuous/growth rate score which performed worse. But, it's only more intuitive because the authors explain the continuous score in a confusing way - I think that growth rates are extremely intuitive for a cell growth assay!

Line 182 - The VUS distribution appears not to match the all missense distribution, with more VUS appearing non-damaging. Is that true by a statistical test? If so, it might be interesting to mention.

Line 186 - The authors assert "Therefore 'depleted' variants can be classified as 'Pathogenic/Likely pathogenic' and 'unchanged' variants as 'Benign/Likely benign'," implying in Figure 4f that all missense VUS can be reclassified as pathogenic or benign on the basis of the data presented in the manuscript. This is egregiously wrong. VUS cannot be reclassified only on the basis of functional data alone. Instead, each must be reclassified using all available data (e.g. patient phenotype, pedigree, variant frequency, etc, etc). The authors own citation 19 are the (now old) rules for this procedure, which they did not follow. Citation 20 does actually do VUS reclassification using functional data and following the proper, updated rules. The authors should either do this analysis correctly or remove figure 4f and accompanying text. The data they collected is high quality and, if they were to use the appropriate rules would probably lead to the reclassification of many VUS. But, they didn't do that here and shouldn't imply that they did. The same comment applies to splice variants. A few sentences later the authors use "might be reclassified" phrasing, but this is totally inadequate especially when the figure gives the impression they did something they didn't actually do

Line 202 - The authors "restricted our analyses to primarily European genetic ancestry (due to power)". This is an insufficiently detailed explanation of the choice to exclude individuals from other ancestries. Why could those individuals not be included (e.g. is there some cancer- or BAP1-specific biology or genetics that means all individuals couldn't be analyzed)? Did the authors actually check other ancestries to verify that there were insufficient numbers for the analysis? They should include a table of the UKB dataset by ancestry and explain in more detail why pooled analyses and analyses of other ancestries is not possible.

Line 208 - "Editorial comment: we generated cancer-type phenotypic variables and rare variant burden test 209 masks" is another example of jargon that many readers will not follow. The authors should explain these types of procedures in simple terms that the generalist geneticist can understand.

Line 213 - The authors used CADD here, but EVE earlier. Why the switch? They should at least explain. I worry that CADD was cherry picked because it happened to not be very good in this

analysis, whereas other predictors did better. The authors should therefore ideally conduct a more robust analysis using a small panel of predictors (e.g. EVE, REVEL, etc).

Line 220 - The authors state that SGE-depleted missense variant carriers do not have a significantly increased cancer burden and note that "This is likely due to low power and effect size of some variants." OK. It might be nice to actually show a spectrum of SGE scores for these 40 variants, with points colored by cancer/not cancer, so the reader could appreciate whether there was a gradient with effect size. The truncating variants could be included too, since they were scored in SGE.

Line 222 - See comment above, but I didn't follow what "roughly double the effect" meant here - is this in the PheWAS or the SGE?

Line 227 - The authors identify a significant association with IGF-1 but don't say whether the p-value was corrected for multiple testing across all variables. They should explain.

Line 250 - The authors describe a case report, stating that the "variant had been classified as a VUS with no functional data available, SGE functional classification will now likely impact clinical management of this kindred." Much more information is required here. On the basis of what data was the variant classified as VUS? Was the variant actually reclassified by a clinician? If so, on the basis of what data (other than the SGE data). See comment above.

Line 259 - What is a "short variant"?

Line 270 - I have major reservations about this analysis. Must the authors really use UMAP to conduct a growth rate analysis? Looking at Figures 6a, b and c make it pretty clear that the UMAP is just "discovering" growth rate. In fact the authors state this "As change across time is the principal dimension regressed from changes between multiple timepoints using UMAP, the 'continuous LFC' metric—which is an orthogonally-calculated change across time—was used to compare LFCs between clusters (Fig.6b)." However, the "continuous LFC" metric, which is just a complicated and not-as-good-as-regression method for calculating a growth rate, is in no way orthogonal to the UMAP, because the data fed into both analyses is identical. If the authors want to use a UMAP, they need to defend why it is superior to simply presenting "continuous LFC" or, better, a growth rate derived from the variant frequency data.

Line 300 - The authors make "rate of change" arguments. I don't think they are actually trying to claim that the rate of depletion of a variant changes over time - I'm guessing most variants have log-linear depletion (as expected). But, if they are actually saying that the rate at which a variant depletes or enriches changes over time, then they need a more rigorous analysis to prove it (e.g. deviation from log linearity).

Line 325 - This entire section boils down to "cancer and developmental Bap1 variants have a similar spectrum of SGE variant effects." It's great that the authors SGE data line up with previous low-scale functional measurements. But, the take-home for this section and indeed the paper is that a simple growth assay could not distinguish the pleiotropic effects of variants in Bap1 (e.g. between cancer and developmental effects). That take home is later mentioned in the discussion, "It is interesting that variants with similar depletion kinetics are associated with both cancer and developmental disorders," but in other places the manuscript suggests otherwise.

Figure 2a - The legend is too pithy to really understand what this panel is showing. Presumably every plot is an exon and every line is a different sgRNA?

Figure 2b - Forgive me if I'm missing something, but it seems like a Cas9+, LIG4+ condition is required in this panel. Also, the legend should show replicates if they exist and also include number of cells sorted (e.g. >3,000 or whatever).

Figure 2e - Editorial comment: the tick marks are pretty useless because they are so dense and actually obscure the tail of the missense distribution, making it look like all missense are synonymous-like. I suggest doing something different.

Figure 3e - Editorial comment: most heatmaps of this type have a blue/red color scheme. Why break this convention?

Figure 5d - This panel would benefit from plain text annotation or a graphical legend as opposed to 3 letter abbreviations.

Author Rebuttal to Initial comments

Reviewer #2:

A: The key result of the paper is a very extensive work which experimentally evaluates almost pre-emptive all single nucleotide variants in the BAP1 gene. In addition, a novel correlation to the IGF-1 concentration has been identified in pathogenic variant carriers

We are most grateful to Reviewer 2 for their very thoughtful comments on our manuscript which we have addressed below. This series of experiments represent an enormous amount of work and the largest saturation genome editing experiment performed to date and profiles a gene whose mutation represents a major clinical challenge in cancer and also in neurodevelopmental genetics.

B: SGE has been used regarding other genes, but additional methods has been used in the design of this paper, and it is original with experimental evaluation of the BAP1 gene. It is a very important paper, which will be extensively used by researchers and contributes to further understanding of the function of the BAP1 gene, and it is also original with the correlation to IGF-1. The paper will give a substantial contribution very much needed in a clinical setting, as the majority of particular missense BAP1 variants are classified as a VUS currently.

We thank Reviewer 2 for making this note. Members of our team manage many families who carry VUS in *BAP1* and we are very pleased we can contribute to further understanding of these variants, which will improve patient care. We are also excited by the IGF-1 observation and the potential that this could lead to improved treatment and potentially prophylaxis in germline *BAP1* variant carriers.

C: The experimental work is very comprehensive and previous papers using SGE in other genes, have been shown to have falsely classified some variants due to additional genetic alterations incorporated in the cells, which then have generated false results for these specific variants. I believe this has been tried to be addressed in this paper as a library A and B, with three replications of the results have been generated and outliers removed.

We are very mindful that the data we have generated will be used together with clinical information (pathology results, disease type/presentation and penetrance) to make decisions about how patients and their families are managed. Thus, we have worked extremely hard to generate a dataset of the highest quality. This includes, as discussed in the manuscript, addressing the technical issue noted by Reviewer 2.

D: The relevant statistic analysis have been used and the setup is appropriate also regarding uncertainties.

We thank Reviewer 2 for this supportive comment.

E: All in all, it is a very robust paper. It has been validated to published clinical data, and thereby shows robust and reliable results

We thank Reviewer 2 for comments D and E. We have ensured appropriate tests have been used in consultation with statisticians and for transparency, have declared sample sizes in plots where statistical inferences are drawn. Where appropriate we have corrected p-values for multiple comparisons to give a false discovery rate and have stated this clearly in the text and figure legends. The approaches/methods we have developed and described in the paper have allowed us to generate a very high quality and robust dataset, that is highly concordant with ascribed classifications of published clinical data.

F: I would also like the authors to address

1. Previous papers regarding SGE in other genes, have been shown to have falsely classified some variants due to additional genetic alterations incorporated in the cells, which then have generated false results for these specific variants. I believe this has been tried to be addressed in this paper as a library A and B, with three replications of the results have been generated and outliers removed. However, when discrepancies have been identified not related to PPE, I would like the authors to add more information about the variants

We thank the reviewer for making this point as it was not signposted in the main text previously. They are correct in their belief that, where possible, we chose to independently edit the genome at exactly the same position with the same nucleotide change using independent HDR template libraries and sgRNAs (Library A and Library B). The principal reason for this was to increase the robustness of the analysis so that many more independent observations of each edited variant could be included when we computed the final functional scores (2 libraries in triplicate = 6 independent observations). In addition, this approach also allowed us to create “scarless” maps that examined all coding region amino acid positions inclusive of those where fixed synonymous substitutions were incorporated to prevent Cas9-mediated recutting (PPEs - PAM/Protospacer protection edits). Functional scores for variants in codons containing a PPE were computed from the alternative library to avoid possible effects in *cis* of the PPE variant (i.e., if a variant occurred in a codon containing a sgRNA A PPE, the score for this variant was calculated from library B data alone and *vice versa*). Such variants can be found in the ‘sge_bap1_dataset’ (Extended Data 1) by examining the ‘pam_codon’ field. In the few cases where data for both libraries could not be used, PPE containing codons were removed (n=140 variants) from ‘sge_bap1_dataset’ (Extended Data 2), these variants are listed in the ‘sge_bap1_expanded_dataset’ (‘pam_flag’ field = ‘Y’).

As highlighted by Reviewer 2, the use of library A and library B has also enabled us to treat the separately calculated continuous Log₂-Fold Change values as independent datasets, calculating separate functional scores, FDR *q*-values and functional classifications for each variant. When we do this, we observe a very high correlation in scores (Pearson’s Correlation Co-efficient of $R=0.95$ at $p<0.0001$, Fig.4a).

When we independently classify variants using library A or library B data alone, we observe that 90% of variants are concordantly classified as ‘depleted’, ‘unchanged’ or ‘enriched’ (13,106/14,624).

A ‘concordance’ column has been added to the final dataset (Extended Data 1) so that researchers and clinicians may consult this aspect of the data for a variant of interest. However, the whole is greater than the sum of the parts, in that independent observations for the same variant that have a higher calculated standard error contribute less to the combined functional score, giving a score that is weighted, increasing robustness quantitatively in relation to experimental error. In other words, we believe the combined (final) functional score and (final) functional classification to be more robust than the independent library A and library B metric for a variant when considered alone, improving the confidence with which the (final) functional score and functional classifications can be used.

As documented in the Online Methods and the Results sections of the manuscript, the nature of our experimental design means that where a SNV leads to a missense, synonymous or stop-gained change, an alternative codon for the same change is also included in the HDR template library. This can be used as a measure of internal experimental concordance for classification at the protein level. Importantly, we find a very high concordance between functional scores for such independent observations (Pearson’s $R=0.91$, $p<0.0001$, Extended Data Fig.5b, and $R=0.89$, $p<0.0001$ for missense

alone). This approach is included in our data in addition to the multiple SNVs at redundant codons that result in the same missense change which are inherently included through the systematic inclusion of all SNVs. We have included this sentence in the text to highlight this analysis:

Line 210: *“Of note, 8,822 unique nucleotide-level changes in our screen result in 4,619 unique missense changes at the protein-level and of these 3,993 could be examined using alternative codon generation, with 16.7% (667/3,993) showing different functional classifications. Thus, not all missense changes have equal effects when encoded by alternative codons, further highlighting the importance and richness of SGE functional assessment at the nucleotide-level.”*

These 667 variants are intriguing and may, for example, represent cryptic splice events or events that change RNA structure. We plan to explore these further in follow-up studies.

In terms of Reviewer 2’s comment: “...additional genetic alterations incorporated in the cells”. Concerns about PPEs have been addressed above. For additional variants incorporated into the genome due to inefficient editing/sequencing, it is important to clarify that the QUANTS pipeline employs exact matching, we use only reads that exactly match intended sequences for each variant, filtering out partial edits and un-intended indels. These unmapped counts have been uploaded to the project GitLab. We are also mindful that any variation private to the cells we use could influence editing outcomes. Importantly we have whole genome sequenced the A5 HAP1 cells, and we found no HAP1 specific SNPs in the target regions assessed, we include these data on the GitLab (and will release as an ENA accession).

2. In addition, the authors have in general compared the data to Clinvar pathogenic variants, and the previously published pathogenic variants regarding the 181 families. But I would like the authors to give more information about the variants that are altered with this classification, both regarding the current supporting clinical data, and if the SGE data has had discrepancies between library A and B, or outliers have been removed from the data. This also refers to the depleted variants currently only observed in Gnomad, where are of the observed patients, and which version of Gnomad (non-cancer) has been used.

Reviewer 2 makes good points which we have clarified in the revised manuscript. Taking each point in turn:

- **“regarding the 181 families...give more information about the variants that are altered with this classification...”**
 - We have expanded our analysis of the variants found in the 181 families. This includes the observation that carriers of depleted variants have an earlier age of onset (Extended Data Fig.7j), and an analysis that shows that the effect size of the SGE functional score for depleted variants is not correlated with cancer type (Extended Data Fig.7k).
- **“give more information...regarding the current supporting clinical data...”**
 - We are mindful to be cautious about providing a clinical reclassification of specific variants in the absence of having a team of clinical experts with access to all of the primary patient and other supporting data (as has been made clear with the exemplar of c.535C>T in Figure.5). We do, however, provide a supplementary dataset of variants found in our screen that were also part of Walpole *et al.*¹ (Supplementary Table 5). This table also includes available clinical information on each case. Variants found in ClinVar and gnomAD are highlighted in the final dataset with the fields ‘is_in_clinvar’ and ‘is_in_gnomAD’ (Y/N), together with accession identifiers. gnomAD metadata is also included, such as allele frequencies in different ancestries. gnomAD variants are Version 3, which is a fixed release and stated in the Online Methods section. ClinVar variant releases are stated in the Results section.

- **“...if the SGE data has had discrepancies between library A and B...”**
 - The ‘concordance’ field is now present in response to the suggestions made above, so that concordance between libraries A and B can be examined for variants of interest. As above, this is not necessarily a measure of classification certainty, as the final classification considers the error in separate measurements and will be more accurate than either separate measurement of LFC. Therefore, the classification made on the combined LFC scores and FDRs gives a more accurate measure of variant function.
 - We see 438 nucleotide-level variants in ClinVar that are depleted in our SGE experiment. We screened 353/438 of these variants with both library A and B finding 82% (289/353) to be concordantly classified between libraries.
 - For Walpole *et al* data, 60/85 nucleotide-level variants in our screen are depleted, 44 are measured separately in library A and B, 93% (41/44) have a concordant classification.
 - 33 variants are depleted in gnomAD, 27/33 are seen in library A and B, 59.3% (16/27) are concordantly classified between libraries.
- **“...have outliers have been removed from the data...”**
 - Outliers have not been removed from data. ClinVar, gnomAD and Walpole *et al* datasets were merged with our SGE data based on HGVS identifier. We state in the introduction what species of nucleotide variants we include, notably: “SNVS.....exon flanking intron and UTR, single nucleotide and codon deletions, together with short indels in ClinVar and gnomAD”.

3. Would you add information regarding splice prediction to all synonymous depleted variants

We are grateful for this request from Reviewer 2, especially as we observe an interesting result. As detailed in the Results section and Extended Data Fig.7a, we see that depleted synonymous and intronic variants have significantly higher SpliceAI scores than unchanged synonymous and intronic variants. We do not see this association for missense variants in non-splice regions.

We have included SpliceAI scores for all variants in the final dataset (Extended Data 1), not only the synonymous depleted variants. This includes multi-nucleotide variants, which are presently not possible to retrieve using the SpliceAI web interface, we achieved this by creating a Sanger Institute based repository of the SpliceAI source code and re-running the model locally on *BAP1* SGE variants, rather than using web-interface (or VEP) pre-computed spliceAI scores. Details are included on the GitLab repository.

4. Line 17: Most inherited variants are not rare, it is only because you have previously filtered for rare variants. Could you please rephrase

The reviewer is completely correct. In the revised manuscript we have tightened up our language and made this clear. Regarding this sentence we have changed “most” to “many”:

Line 22: “Many variants we inherit from our parents, or acquire *de novo* or somatically are rare, limiting the precision with which we can associate them to disease.”

5. Line 34: I assume these variants are from clinvar. Please state if they are from clinvar or other places and state time for the clinvar search

We have amended this sentence:

Line 38: *“Most variants identified in BAP1 that are causally linked to tumour predisposition, are frameshift or truncating variants, yet to date >1000 missense variants have been clinically observed, including 396 reported by multiple investigators, most of which are rare and functionally ambiguous, with >98% classed as VUS (variants with ≥1* review status, ClinVar 20/09/2023)².”*

6. Line 38: please add “surveillance recommendations and risk-reduction strategies”, as very few strategies will prevent cancer.

We agree that this is an important distinction and as suggested we have amended the sentence accordingly.

7. Line 42. Please state if this is only mesotheliomas or if other specific cancer types also have been shown to benefit from these treatments. If not alter to mesotheliomas

As suggested, we have amended to: *“For example, recent evidence suggests that BAP1-deficient mesotheliomas...”*

8. Line 60: Please specify the types of variants examined or state that not all variants have been examined, ie insertions, large indels etc.

As suggested, we have outlined the types of variants examined in our study, in this sentence:

Line 65: *“In this study we use saturation genome editing (SGE)³ to profile 99% of all possible single nucleotide variants in the BAP1 coding sequence (6,501/6,570) with the aim of improving precision medicine. We also exhaustively profile exon flanking intron and UTR sequence, single nucleotide and codon deletions, together with short indels in ClinVar² and gnomAD⁴.”*

We have included an extensively annotated final dataset (‘sge_bap1_dataset’, Extended Data 1) in which investigators and clinicians can search for a variant of interest using several possible identifiers: e.g., “chrom_pos_ref_alt”, “HGVS_c”, “HGVS_p” among others.

To address this comment further we have also created an online BAP1 viewer which can be queried with plots and metrics returned to the user:

<https://bap1-viewer.shinyapps.io/bap1viewer-development/>

9. Line 83 “HDR repair” is a pleonasm (homology directed repair repair). Please remove repair. This phrase has been used multiple places in the manuscript.

We thank Reviewer 2 for clarifying this point and we have rephased this description to ‘HDR template library’ throughout the manuscript.

10. Line 152: Please add if this is a scientific significant change. Comparing to the frequency in the background population is very strong. In addition, please add a comparisons between LOF variants frequency in the background population and constrained missense variants (are they less constrained or as the LOF variants)

This is a statistically significant change. Variants classed as depleted and enriched are observed proportionally less in gnomAD than unchanged variants (Chi-squared; $\chi^2 = 49.1, p < 0.0001$). We have stated this is the text and legend for Fig. 3d. $p = 2.138e-11$ for the Chi-squared test, and $p = 8.314e-15$ for a two-sided Fisher’s Exact test using the same contingency table, both are $p < 0.0001$, so we have reported the Chi-squared statistic alone as <20% of cells are <5 counts (if >20% cells were <5 counts it would be more appropriate to report the Fishers’ exact test result.

The contingency table and Chi-squared test outputs are combined in the table below (R1), note that the residuals between the observed and expected frequency for depleted and enriched variants in gnomAD are negative (highlighted in orange). This indicates that the low observed frequency of these variants contributes to the rejection of the hypothesis that there is no difference between functional classifications and observed frequencies in gnomAD.

classification	χ^2 statistic	p.value	observed freq in gnomAD	observed freq in SGE screen	expected freq in gnomAD	expected freq in SGE	residuals for freq in gnomAD	residuals for freq in SGE
depleted	49.13679	2.14E-11	6	5665	43.50833	5627.4917	-5.6864539	0.5000003
enriched	49.13679	2.14E-11	3	531	4.096887	529.9031	-0.5419198	0.0476501
unchanged	49.13679	2.14E-11	131	11912	92.394783	11950.6052	4.0162642	-0.3531434

Table R 1. Chi-squared test computed in R using ‘stats’ package (version 3.6.2). The contingency table for the test contained the following columns: ‘observed freq in gnomAD’ and ‘observed freq in SGE screen’, which are the variants found in gnomAD and also our SGE screen, and the total variants for each functional classification in the SGE screen, respectively. Expected frequencies computed by the test are shown, with negative residuals for comparisons between functional classes for variants in gnomAD highlighted in orange.

These tests have used variants found in our SGE screen and gnomAD only, and not in ClinVar (n=140). When we assess variants found in our SGE screen that are seen in gnomAD as well as in ClinVar (n=593, see Fig.4d lower panel for classification breakdown), we also see a significant difference between the frequency of functional classifications seen in gnomAD (Chi-squared; $\chi^2 = 228.5, p = 2.408906e-50$).

To compare between LoF and missense frequency as requested we have broken down the observed/expected (o/e) ratios by synonymous, missense or stop-gained (LoF) consequence. The observed frequencies are the number of unique SNVs seen in:

- gnomAD (independent of whether seen in SGE, for comparison)
- SGE screen (all functional classifications) and in gnomAD
- SGE screen unchanged and in gnomAD
- SGE screen depleted and in gnomAD
- SGE screen enriched and in gnomAD

The denominator expected frequencies are predicted by gnomAD: missense = 431.58, pLoF = 33.05, synonymous = 179.05. pLoF variants in gnomAD are constrained, consistent with *BAP1* being an essential gene, with an o/e of 0.12 observed, the LOEUF score (essentially o/e with a confidence interval) is 0.28. Therefore, variants were an o/e value below 0.28 is seen can be considered to be constrained in *BAP1*. As can be seen from Fig. R1 A, synonymous and missense variants that are unchanged are not constrained, whereas those that are depleted and enriched are constrained to significant levels (Chi-squared; $p < 0.0001$). As expected, all LoF variants are constrained regardless of functional classification (no significant difference is seen in o/e between classifications for LoF variants by Fisher's Exact test).

“In addition, please add a comparisons between LOF variants frequency in the background population and constrained missense variants (are they less constrained or as the LOF variants).”

We observe that missense and LoF depleted and enriched variants are both constrained.

Figure Revision 1 A. Bar chart showing observed/expected ratio of variant counts in gnomAD and the *BAP1* SGE screen. gnomAD bars are not based on SGE data. All other bars are observed variants in gnomAD that are also in the SGE dataset, categorized as either all variants (SGE) or by functional classification (unchanged, depleted, enriched). Categories are separated by mutational consequence (synonymous, missense and LoF), with a different expected frequency used for each (derived from gnomAD model). Dashed black line shows the pLoF o/e based on gnomAD data, with the LOEUF upper limit for constraint shown by a dashed red line. Chi-squared test statistic and p-values are shown for comparisons between classifications for each mutational consequence. Fisher's exact test was used for LoF due to low frequencies. **B.** gnomAD SNV variants assayed in SGE that have a stated allele frequency (n at top of the plot). Unchanged variants have a greater number (n=15) of true outliers that are common (AF>0.001%) than either enriched (n=1) or depleted (n=0).

“Comparing to the frequency in the background population is very strong.”

More unchanged variants than enriched or depleted are seen in gnomAD, and whilst the majority of BAP1 variants in gnomAD are rare with an allele frequency (AF) <0.001%, we can see that more (n=15: 6 intron, 3 missense, 4 synonymous, 2 3'UTR) unchanged variants are common (AF>0.001%), than enriched (n=1, AF=0.002%, an intron variant, rs143659795, with Likely benign ClinVar status, variant ID=240070), no depleted variants are common (Fig.1R B). As there are so few observations of depleted and enriched variants in gnomAD we have decided not to include this analysis in the manuscript so that the data is not over interpreted, we are confident that there are fewer depleted and enriched variants in gnomAD than expected, and that Fig.3d (and Fig.4d) is sufficient to highlight this point. There is no significant difference in median allele frequencies between the functional classifications, however as most variants are rare this is not surprising (Table R2).

Comparison	Z	P.unadj	P.adj
depleted - enriched	-1.8309301	0.06711098	0.10066646
depleted - unchanged	-0.6635165	0.50699979	0.50699979
enriched - unchanged	2.03180055	0.04217385	0.12652155

Table R 2. Dunn’s BH FDR values for pairwise comparisons between median allele frequencies for data shown in Fig.1R B. Note that comparisons for a difference in allele frequency between unchanged, enriched and depleted variants is not seen in p values corrected for multiple testing (P.adj, highlighted in orange).

11. Line 171: please specify the number of the specific variants (LOF, missense etc)

We have listed the specific variants used in the ROC analysis in Supplementary Table 4. We have also amended the sentence (below) to allow for re-retrieval from ClinVar if necessary/desired.

Line 189: *“To examine functional scores further we first identified variants with strong clinical/functional data in support of their classification curating 851 benign (‘true negative’) and 199 pathogenic (‘true positive’) variants which had at least one star ($\geq 1^*$) in ClinVar (downloaded 04/09/2023).”*

In response to reviewer comments below we now use variants with $\geq 1^*$ for the ROC analysis. In addition, due to the imbalance in the number of true negative and true positive variants we also calculated the precision recall (PR) area under the curve, which we find to be >0.999 and state this in the manuscript (see Fig. R2 for comparison between ROC AUC and PR AUC).

Figure Revision 2. A: ROC AUC as in Fig.4b, here reproduced in PRROC software to compare with a precision recall curve calculated from the same truth set data. **B.** Precision recall graph with AUC shown. The colour scale on the right gives an indication which classification threshold results in a particular point on the graph, i.e. certain pairs of sensitivity & false positive rate (that is FPR, or 1-specificity) in A, or certain pairs of precision & recall values in B.

12. Line 190: Please add any relevant published or available clinical data regarding these patients/families, which potential can corroborate the change in classification.

In the revised manuscript we have endeavoured to use a more clinically relevant process when documenting the potential for our SGE data to aid in variant re-classification. This has taken the form of an evaluation of SGE assay performance against the ACMG evidence framework (i.e., Brnich *et al.*). Encouragingly we find strong and very strong evidence for a pathogenic/benign classification for depleted and unchanged variants, respectively. We believe that this is a more responsible approach than stating the breakdown of functional classifications for ClinVar clinical significance categories, alone. Notably we have also had our approach reviewed so it aligns with the new *BAP1* practice guidelines – this was done by Helen Hanson (co-author) who chaired the international team who wrote this guide⁵.

13. Line 217: Please add the average age of cancer onset in the different cohorts with STD depletion and non-depletion, to see if there is a younger onset of cancer in the depletion cohort.

We thank the reviewer for this request, we have included the following sentence to address:

Line 333: “Importantly, we find that the average age of cancer onset for SGE-depleted non-synonymous *BAP1* variant carriers and non-carriers in UKBB is similar at 62.54 ($n=24$) and 60.71 ($n=95,185$) years, respectively (60.57 years, $n=9,071$, for SGE-unchanged *BAP1* variant carriers).”

14. Line 226: could you add information if there are differences in IGF_1 levels in depleted variants carriers with cancer compared to without cancer

This is a very important consideration. As requested, we have performed this analysis. We have added the following paragraph in the Results section of the manuscript:

Line 346: "Importantly, we do not see a difference in mean IGF-1 levels between SGE-depleted non-synonymous BAP1 variant carriers with cancer (IGF-1 level=22.19 nmol/L), and those without cancer (IGF-1 level=24.36 nmol/L, $p=0.19$). Likewise, non-carriers with and without cancer have similar mean IGF-1 levels, 20.85 and 21.51 nmol/L, respectively. This highlights that significantly increased IGF-1 levels are specific to individuals with SGE-depleted non-synonymous BAP1 variants rather than a cancer diagnosis and suggests a possible mechanism of BAP1-mediated pathogenicity".

15. Figure 3c lower panel, add significance level

We thank the reviewer for this comment, this refers to Figure 3.d (not 'lower panel' of 'Fig3.c', we have made the letters bolder), we have added the Chi-squared test p-value ($p<0.0001$) to the plot and have stated the χ^2 value (49.2) in the Results text and figure legend.

16. Extended data figure 9: Alter the pedigree to standard. It looks like husband and wife where it presumably are siblings.

We thank Reviewer 2 for noticing this and we have revised the figure accordingly to illustrate more clearly the relationships in this pedigree.

G: yes

H: Very clear and in a clear context. Remarks to abstract is in the above suggestions.

We sincerely thank Reviewer 2 for their comments which we feel have helped improve the clarity and quality of our manuscript.

Reviewer #3:

In this manuscript, the authors present a large variant effect data set, gathered using saturation genome editing, for the cancer-associated gene Bap 1. The main attractions of the work are that the data is of beautiful quality; that functional data for Bap 1 is needed because of the burden of germline and somatic variants as well as the relationship of the gene to multiple diseases; and that the analysis includes data from various human cohorts like UKB that yield some surprising results. In particular the association of Bap1 function with IGF-1 levels in people and tumor cells is fascinating and really showcases the power of combining comprehensive functional data and large human cohorts. However, the work has some serious flaws. Most importantly, I feel that the analysis of the saturation genome editing data that was performed was far from optimal and, in places, didn't make sense. Additionally, much of the analyses following the SGE itself felt cursory and, in places, were factually incorrect.

We thank Reviewer 3 for their appreciation of the quality of our data and the efforts we have made to link what is a clinically important and much needed dataset to the basic biology of *BAP1*-associated malignancies, and other *BAP1*-related conditions. We welcome the reviewers' insights into the analysis approaches we have used, which we have addressed in full below. At a high level we show that we can define variant effects with extremely high precision and accuracy when compared to "truth sets" and we have further refined these approaches with their input.

Major comments:

-The authors generated data from a cell growth assay, measuring variant frequency at numerous time points. But, instead of doing a straightforward growth rate analysis they execute a number of confusing comparisons of specific time points to compute log-fold changes, then they compute something kind of like a growth rate and, finally, resort to inappropriate dimensionality reduction methods. The conclusions they reach aren't wrong, they're just not as good as they could be and quite confusing to follow. The work would be much better if the authors picked one metric for scoring each variant (in my opinion an apparent growth rate computed by log-linear regression, with an appropriate error model) and deviated from it only when necessary (e.g. to explore "growth rate changes," see comments below).

We thank Reviewer 3 for their comments and appreciate their insights into how we may improve and clarify our analysis approach. Firstly, we have discussed our approach with Wolfgang Huber, the creator of DESeq2, to confirm that we have used DESeq2 correctly and have employed the z-test correctly in relation to the assumed distribution of data.

To be as transparent as possible about our approach to analysis we have elaborated on several points which were raised by Reviewer 3. We respond to each point in turn below:

Choice of metric

In the original manuscript we used the LFC between D4 and D21, converted to a z-score (by dividing by the standard error), throughout the manuscript to classify variants and also generated other comparisons such as D4, to Days 7, 10 and 14 to demonstrate that the accuracy of functional scores improves with time. We accept that the previous presentation of the various comparisons was complex and thank the reviewer for suggesting an alternative approach.

To make our analysis and description of the data more straight-forward we have now elected to use a single metric as suggested, the '**continuous LFC**' which we have re-named as '**functional score**' for clarity. This, as the reviewer correctly notes is a growth rate, and is indeed produced with a log-linear regression with an appropriate error model (Negative Binomial a.k.a. Gamma-Poisson distribution for count processes with over-dispersion). This is the analysis approach suggested by the reviewer. We have made this explicit in the text.

The functional score when hypothesis-tested asks:

"Is there a significant linear correlation between time and normalised log read count?"

Use of DESeq2

- We used DESeq2 to calculate a 'continuous LFC' metric, which after some adjustment (see below) we term 'functional score'. This is produced through a log-linear regression, with an error model fitted (negative binomial distribution for count processes with over-dispersion between replicates) followed by an appropriate statistical test (two tailed z-test). We have made this clear in the revised text. We detail below why we have used DESeq2 for our analysis.
- As Reviewer 3 notes (below), our data is not normally distributed. However, as count data (discrete positive numbers) with over-dispersion, they can be well modelled by a negative binomial distribution (a Poisson distribution would not account for over-dispersion). If we wish to fit the data to a log-linear regression as suggested, then we should model the count data using a negative binomial distribution and this can only be achieved by using a generalized linear model (GLM) such as that employed in DESeq2.
- A GLM to fit data as a negative binomial has two key parameters: the mean and dispersion. Variability between replicate raw count data about the mean is routinely modelled using dispersion estimates. However, in experimental designs that have few, but reasonable, numbers of replicates ($n=2$ or 3), dispersion estimates will be highly variable for each variant (over-dispersed), and accurate estimation of dispersion is critical for statistical inferences about variant change over time. If dispersion estimates are used directly, they will be highly noisy, compromising subsequent statistical tests. As reviewer 3 highlights, an appropriate error model needs to be employed on a log-linear regression. DESeq2 satisfies this requirement as it accurately estimates dispersion and applies it to the GLM.
 - DESeq2 includes an empirical Bayesian approach to dispersion estimation, sharing the dispersion estimates across all variants in a target region and shrinks dispersion estimates towards the curve. This reduces false positives in cases where dispersion is underestimated. Conversely, when a variant's dispersion is far above the target region estimate (more than 2 residual standard deviations above the curve), the DESeq2 dispersion error model assumes that this particular variant does not obey the modelling assumptions; in such cases DESeq2 does not shrink the dispersion estimate towards the curve as doing so may result in undesirable false positives. Therefore, DESeq2 allows us to accurately estimate the dispersion of variant counts between replicates making subsequent statistical tests more reliable.
- The two above components (GLM negative binomial fitting and dispersion estimation) are the core elements of count handling models such as DESeq2 and MAGECK. In our experience DESeq2 allows the user to change more parameters (see below), so we used DESeq2. In

addition, DESeq2 has been found to have a lower false positive rate than alternative models⁶ (although this concerns RNASeq and not SGE, so is not our predominant reason for selecting this program/analysis approach).

- Of note, DESeq2 allows data to be fitted as a negative binomial together with accurate estimates of inter-replicate variability (dispersion). At the same time DESeq2 also allows us to calculate LFCs between time-points when timepoints are considered as a discrete, categorical variable, or over all time-points when time is considered as a continuous variable. The latter we call 'continuous LFC' renamed as 'functional score' (after some additional adjustments, below) in the revised text. DESeq2 also calculates a standard error value for each variant's LFC estimate, which together with the 'functional score' (LFC estimate), allows us to categorize the variant into a 'functional classification' following a statistical test (below).
- LFC estimates in which time is considered as a continuous variable are computed as a log-linear regression through DESeq2. This is because the GLM of DESeq2 includes the requisite exponential function to link the probability distribution parameter (the central parameter of negative binomial distribution) and the linear regression term (the linear predictor, change in log read counts over time).
- The link can be understood mathematically as follows:

Assuming that for an oligo i and time t , the observed cell count is given by a negative binomial random variable $N_{it} \sim \text{NB}(u_{it}, \alpha_i)$, where u_{it} is the mean of the distribution and α_i is the dispersion coefficient. Assuming that at two different time stamps t and t_0 the expected count follows log linear growth model, thus:

$$E[N_{it}] = E[N_{it_0}] \exp\left(-\frac{t-t_0}{\Delta_i}\right) \quad \forall \Delta_i > 0$$

Where Δ_i is decay time parameter, $E[N_{it}]$ and $E[N_{it_0}]$ are expected cell counts at time t and t_0 , respectively. Then we can put these assumptions into one-to-one correspondence with DESeq2⁷ (equations '1' and '2' in Love *et al*, 2014, see supplementary page below) such that:

$$K_{ij} \approx N_{it},$$
$$\mu_{ij} \approx \mu_{it} \propto q_{it} = \exp\left(-\frac{t-t_0}{\Delta_i}\right)$$

and

$$\log q_{(it)} = \beta_i t + \beta_{i_0}$$

Where β_i and β_{i_0} are precisions and given by $-\frac{1}{\Delta_i}$ and $\frac{t_0}{\Delta_i}$, respectively.

- The 'functional score' for a variant is therefore the Log₂-Fold change in count abundance per unit time over Days 4 to 21 (inclusive of D7, 10, 14), which is a growth rate computed through log-linear regression. As above, this is the metric suggested by Reviewer 3, with an error model (also suggested by reviewer 3) in the form of fitting to an appropriate probability distribution with dispersion estimation, reducing false positives. In addition, corresponding standard error values for each LFC estimate are produced.

Changes to DESeq2 default parameters and post-DESeq2 adjustments

- As mentioned above, we change some default parameters in DESeq2 to tune the model to be more appropriate for SGE over its common use in RNA-Seq differential gene expression analysis. In the original submission, we estimate the experimental size factors for DESeq2 input using only counts from variants which we do not expect to change greatly over time, namely synonymous and intronic variants. We did this because by default DESeq2 estimates size factors based on all variants/genes, because in differential gene expression analysis (the most common use case for DESeq2) most genes do not change over time. For SGE, however, where many variants in a target region will be expected to change, we define a group of variants for each target region and replicate, the majority of which we do not expect to change in abundance over time. In addition, we now output the un-shrunken LFC values from DESeq2 and use these for our downstream analyses. By default, DESeq2 shrinks LFC estimates closer to zero for low count or highly dispersed variants based on a Bayes process. However, upon review we realise that the priors used for this shrinkage are designed for RNA-Seq and, therefore, we disabled LFC shrinkage in our most recent analysis. The effect of this will be that depleted variants with low counts will have a slightly more negative LFC than in the previously analysis and *vice versa* with enriched variants.
- As in the previous analysis, to produce the final ‘functional score’ we adjust the ‘continuous LFC’ produced by DESeq2. Firstly, we scale the LFC by the median of the synonymous and intronic variant LFCs, to normalize for differences in LFC scale between target regions. Secondly, we weight the library A and library B continuous LFCs by their respective standard errors and then combine them to produce the final ‘functional score’ for each variant – see step by step section below.

Step by step walk-through of the analysis process to summarize our approach

- Counts for unique variants in each target region are obtained through the QUANTS pipeline to get exact matches to intended edits for each timepoint (D4, 7, 10, 14, 21 and replicate R1, 2, 3) with the 15 samples (5 timepoints, 3 replicates) arranged in a dataframe.
- Counts are filtered if a variant has <10 counts over all timepoint-replicates (15 samples).
- VEP consequence annotations are merged and synonymous and intronic variants used to estimate size factors for each of the 15 samples in the target region dataframe.
- DESeq2 is run on the target region with the computed size factors defined, un-shrunken LFCs and Standard Errors for each variant are produced.
- The LFCs are scaled by the median of the synonymous and intronic variant LFCs.
- The median scaled LFCs for the same target region derived from Library A or Library B are now combined.
- A weight factor is produced for each independent variant observation in Library A and B, by taking the reciprocal of the LFC Standard Error squared.
 - ‘weight_a’ = $1/(A_lfcSE)^2$
 - ‘weight_b’ = $1/(B_lfcSE)^2$
- We sum the weights to give ‘sum_of_weight’ and then raise by the exponent -0.5 to produce a new single Standard Error value for the variant (called SE_bind)
- The median scaled LFCs (‘adj_lfc’) are adjusted in proportion to their Standard Error, thus:
 - $weighted_A_LFC = weight_a * A_adj_lfc$
 - $weighted_B_LFC = weight_b * B_adj_lfc$
 - PPE codons are not weighted: library A LFCs are used for library B PPEs, *vice versa*

- And to produce the final weighted LFC as a single value:
 - $\text{sum_of_weighted_LFC} = \text{weighted_A_LFC} + \text{weighted_B_LFC}$
 - $\text{LFC} = \text{sum_of_weighted_LFC} / \text{sum_of_weight}$
- We then repeat this process for regions that overlap between target regions (*i.e.*, 11.1 & 11.2, 12.1 & 12.2, 13.1 & 13.2, 13.2 & 13.3, 17.1 & 17.2).
- We then collate all the target region functional scores, which are in the form of one score for each unique variant.
- We then calculate a z-score for each variant which is functional score/Standard Error (SE_bind).
- We perform a two-tailed z-test on this z-score distribution and correct the *p*-value by the Bonferroni-Hochberg method to produce a FDR for each variant.
- Functional classifications are then derived from the functional score and FDR:
 - Unchanged = $\text{FDR} \geq 0.01$
 - Depleted = $\text{FDR} < 0.01$ & functional score -ve value
 - Enriched = $\text{FDR} < 0.01$ & functional score +ve value

–“....and, finally, resort to inappropriate dimensionality reduction methods.”

We reflected on the comment about dimensionality reduction and agree with the reviewer. As such we have removed these analyses. To make comparisons we now use the simpler approach of ranking variants by functional score magnitude to identify any interesting effect size findings. This approach distinguishes the same putative hypomorphic variant as the previous analysis through a more appropriate (simpler) and intuitive process.

-The authors make statistical assumptions that are unsupported by the data, most especially assuming that various quantities are normally distributed when they are obviously not. There is no reason to do this and it causes problems for the interpretations that follow.

The reviewer is correct that the count data is not normally distributed. In particular, the count data are discrete positive numbers. They can be well modelled by a negative binomial distribution. Our analysis method to generate functional scores does not assume normal distributed data. As detailed above we used DESeq2, a generalized linear model that fits the count data using a negative binomial distribution for the stochastic component, and a linear regression followed by an exponential function for the deterministic component. As above, we have confirmed this to be appropriate in consultation with Wolfgang Huber.

-The manuscript contains some serious errors. The best example of this is the explanation of how Bap1 SGE data could be used to reinterpret germline VUS. As written, the manuscript is wrong, and would mislead readers into thinking the authors did something they did not actually do.

The data we have generated is of extremely high quality and exhaustively validated. Using the American College of Medical Genetics Guidelines (ACMG)⁸ such datasets can be used to assist variant reclassification, together with clinical information, and indeed SGE data of selected domains of *BRCA1* from Greg Findlay (Findlay *et al.* Nature 2018³) are being used for such a purpose. Shawn Fayer’s recent paper (Fayer *et al.* AJHG 2021⁹) showed several hundred entries on ClinVar in which these data have been used to refine *BRCA1* variant classification¹⁰. Of note, for some conditions, such as Fanconi Anemia and pediatric metabolic disorders, functional tests have always been used to assist variant (re-) classification. Importantly, in providing the information in Table 1 (the odds ratios of a variant we score as being disruptive or non-disruptive based on “truth sets”) we now provide information in the established framework for clinicians to use our data, together with other pieces of clinical evidence,

to classify variants but we do make clear that ultimately these decisions are for the clinician. Of note, two of the authors on the revised manuscript are consultant clinical geneticists (Prof. Clare Turnbull and Dr. Helen Hanson), who have co-authored the CanVIG-UK/ACMG guidelines and the BAP1 clinical guidelines, respectively. Thus, our approach is the state-of-the-art.

-The work does not deliver on many of the promises made in the abstract. The best example of this is the IGF-1 finding, which is billed in the abstract as “Our analyses demonstrated that disruptive germline BAP1 variants are significantly associated with higher circulating levels of the mitogen IGF-1, suggesting a possible pathological mechanism and therapeutic target.” This is true - they did find an association and it’s very cool. But, they don’t explore either mechanism or the possibility of a therapeutic. This finding at least needs quite a bit more exposition. The authors never explain what IGF-1 is or why it might relate to cancer or make a therapeutic target. Ideally, they would do more follow up experimentation and analysis but at least a solid discussion backing up the assertion in the abstract is needed.

We thank Reviewer 3 for noting that this observation is very cool – we agree! After discussing this point with the Editor, we feel that detailed mechanistic studies are for the next paper and beyond the scope of this manuscript, which is already an enormous body of work (pushing right at the limit of the word and figure count). We have, however, further examined the IGF-1/BAP1 relationship using UKBB data (as requested by Reviewer 2) and enhanced the discussion of IGF-1 in our revised manuscript. We also cite additional key papers on the role of IGF-1 in cancer and development and make note that our observation may suggest a potential prophylactic or therapeutic role for agents that modify the IGF-1 axis, several of which are already available¹¹ in patients carrying germline pathogenic *BAP1* alleles.

-The manuscript contains a lot of undefined jargon which will cause problems for generalist readers.

We thank Reviewer 3 for this comment and have asked several non-SGE colleagues (generalists) to read the paper and identify any jargon that is not defined which we have clarified.

Detailed comments

Line 72 - The authors screen many sgRNAs and pick some. Since they are presenting improvements for the SGE method they should explain why they picked the sgRNAs they did. It’s not clear from looking at Fig 2a - some chosen sgRNAs deplete very quickly (e.g. have a low frequency even at D7) whereas others don’t. In other cases, what seem like clearly superior choices are ignored (e.g. in exon 3 the chosen sgRNA has less depletion from D7->28 than other not chosen sgRNAs). I know they point to reference 12, but they should briefly explain here, too.

We thank for the reviewer for looking at the plot in detail, we have altered the methods to outline our selection process for sgRNAs:

Line 101: *“We elected to use cell fitness as a biological readout of BAP1 function, first rigorously re-confirming BAP1 essentiality (Extended Data Fig.2a-c) and SGE efficacy (Extended Data Fig.2d) in HAP1. To aid selection of appropriate sgRNAs for experimentation, we next performed a targeted CRISPR/Cas9 screen with 193 sgRNAs tiled across all 17 BAP1 exons (Fig.2a). sgRNAs for SGE were selected based principally on design parameters (as previously described¹²), with depletion kinetics also considered (see Methods).”*

And in methods section in question:

Line 726: “sgRNA selection and cloning

“All sgRNAs with a 20nt spacer across the BAP1 gene were obtained through the CRISPR function within Geneious™, with off-targets scored against the GRCh38 genome. sgRNAs for SGE were chosen based on a set of criteria, as previously reported¹², including: synonymous changes possible in PAM or protospacer codons to give PAM/protospacer protection edits (PPEs), the sgRNA target site positioned in CDS and distal to splice junctions in target region, no predicted off-targets in coding sequence and >2 mismatches in any non-CDS off-target. Also sgRNA A and sgRNA B for the same target region were chosen to be non-overlapping where possible and PPEs selected to avoid codons where ClinVar or gnomAD variants have been reported, where possible. sgRNA selection for SGE was also evaluated through depletion dynamics in the targeted CRISPR screen of BAP1 (see Methods: ‘Essentiality phenotyping - targeted CRISPR/Cas9 screen’), with those demonstrating gradual depletion over-time (~ 25% reduction in cell fitness between each of the first 3 time points) preferentially selected; we hypothesise that such sgRNAs exhibit cleavage events associated with loci-specific death, whereas general genotoxicity might be expected to result in immediate, strong depletion.”

A sgRNA that depletes more in the CRISPR screen is not necessarily a preferential choice for SGE; it may still cut the genome but not result in frame-shifting NHEJ for instance. The most important aspect of the design process are the considerations listed in the VaLiAnT reference¹² (shown below), and then we consider the depletion kinetics to ensure the sgRNA is likely effective at cutting the genome.

Supplementary Table 1. sgRNA site and PAM/protospacer protection nucleotide selection criteria

sgRNA selection criteria		PAM/protospacer protection criteria	
Attribute	Reason	Attribute	Reason
sgRNA binding site at appropriate locus.	Fits design parameters.	Only mutate 3 rd base of a codon at PAM site or in protospacer.	Codon redundancy allows for synonymous edits.
sgRNA binding site within targeton CDS (if targeting exon).	Frameshifting indels created by NHEJ will deplete (if CDS mutation is deleterious) increasing relative representation of HDR edits in screen.	Always change pyrimidines to pyrimidines (C/T) and purines to purines (A/G).	Missense changes caused by SNV saturation at PAM/protospacer protected codons will replicate missense changes achieved as if codon were wild-type (with some exceptions*).
Synonymous edit possible at PAM and/or protospacer.	Prevents Cas9 from targeting HDR library template, which depletes representation of edited loci.	Wherever possible select a pyrimidine base to change.	No exception to above missense representation.
No 0 (exact), 1 or 2 mismatch to CDS off-target.	Avoids off-target effects and preserves on-target activity.	Wherever possible change terminal G of PAM over more protospacer proximal G.	Avoids NAG PAM sites which spCas9 still cuts with limited efficacy.
No 4 (or more) consecutive thymine residues (TTTT).	Prevents premature transcriptional termination of sgRNAs expressed from RNA polymerase III promoters (such as U6::3).	Antisense sgRNAs are preferred (CCN PAM rather than NGG).	An increased number of pyrimidine changes are possible.
Avoid sgRNAs that have G/GG adjacent to PAM NGG (leads to two consecutive GG at PAM after protection edits).	Despite PAM editing to protect from Cas9, cutting still occurs if NGG exists adjacent to wild-type PAM.	No protection edits in the unique codons, ATG[M] and TGG[W] or ATA[I].	Purine protection edits at the 3 rd position of these codons do not lead to synonymous changes.

*in addition to a purine change at 3rd base, if: 1st base=T or 2nd base=G, then W/STOP missing in SNV SGE library; 1st base=A or 2nd base=T, then M/I missing in SNV SGE library; 1st+2nd base=TT or 1st+2nd base=AG, then both W/STOP and M/I will be missing in SNV SGE library.

Line 104/Figure 2e - The authors compute a z-score which is inappropriate for data that is, clearly, not normally distributed. It seems to me that a linear rescaling of the data would achieve much the same result without putting the data on a scale (z-score) that has a meaning (e.g. each unit is 1 SD) for normally distributed data but doesn't have that meaning here because the data are not normally distributed.

We have discussed this point above and have altered the manuscript accordingly. Reviewer 3 is making an insightful point since read counts used to calculate z-scores for depleting variants will be lower and potentially show more dispersion between replicates (and hence a higher error). That said we note that using LFCs (as suggested by the reviewer) instead of z-scores does not change the plots or the results of our analysis. Nonetheless, we have altered the text and figures to be based on LFC alone throughout.

Line 104 - The authors use a z-test to identify significantly depleted variants. The core assumption of the z-test is that the data are normally distributed. Do we know that the replicate scores are actually normal? Would not a t-test (at least) be more appropriate since n is small and the assumption that score distributions are normal seems fraught? Maybe even better would be to use a nonparametric (e.g. Wilcoxon rank-sum test, etc) test.

As above we discussed this with Wolfgang Huber (who wrote/developed DESeq2). It is not necessary for count data replicates to be normally distributed for the calculation of the z-statistic (also known as Wald statistic) to be appropriate, as it is a summary statistic, and the central limit theorem can be applied. The z-statistic (Wald-statistic) calculated using DESeq2 LFC estimates and standard errors accounts for dispersion based on the negative binomial distribution of count replicates. If we were to employ a native form of a log-linear regression model (without an assumed negative binomial distribution and dispersion estimate) then the t-test would be appropriate. However, for a more advanced approach using a generalized linear model we do not need to estimate the population variance/standard deviation, as we know the standard error and the sample size is large ($n > 30$), therefore the z-test (Wald-test) is appropriate.

We thank Reviewer 3 for this comment. We spotted that the FDR calculations in the previous analysis could be improved upon, as we made all our variants unique by HGVS annotation rather than by the actual sequence edited into the genome (which we term 'mseq' in code and datasets), the former would have had duplicate rows in some annotation instances. This has an extremely minimal effect on the analysis and statistics but we have updated our analysis to address this, only processing completely unique sequences from the beginning to the end of the analysis process giving 18,108 completely unique sequences separately edited into the genome and the corresponding FDRs based on this number. As FDR calculations depend on the number of unique observations, the most recent analysis and FDR calculation will be slightly more accurate.

Line 104 - Related to the above comment, it is difficult to tell exactly what tests were performed, here and in the methods. Did the authors really compute a test for every variant at each time point relative to D4? If so, why? It seems to me that they are creating a big multiple testing problem that, after correction, will result in many important variant depletions being nonsignificant. Why do they need to use an all-by-all statistical design (e.g. all variants x all timepoints)? Wouldn't just one timepoint suffice?

We did not compute separate tests for every variant at each time point. We previously only computed the FDR for one timepoint, the D4 D21 comparison. As we have now adopted a single metric, the functional score, we only use the functional score FDR. We use a highly stringent significance level of 1%, however we also correct for multiple testing by adjusting the p -value

produced from the z-test for the number of tests performed using the Benjamini-Hochberg procedure to produce an FDR q -value, which is the only test used for functional classification. We have made this clear in the online methods section: "Calculation of variant abundance and functional scores", in particular:

Line 915: "To avoid confusion all 'functional classifications' were made using the 'functional score' and its corresponding FDR, in summary; the 'continuous LFCs' produced from DESeq2—which are LFCs per unit time over all timepoints, separately calculated for Library A and B—were adjusted by median scaling and Library A and B values combined by weighted average to produce the 'functional score'. The functional score FDR for each variant was calculated by computing a z-score and then performing a z-test, which was then adjusted for multiple testing using the Benjamini-Hochberg procedure. The functional score and the functional score FDR were then used to produce functional classifications."

Code and detailed description/justification for the functional score calculation process can be found here: https://github.com/team113sanger/Waters_BAP1_SGE

Line 104 - The authors collected beautiful time series data. Why not use the time series data to compute an apparent growth rate for each variant and then test for differences in growth rate? Many MAVE-type experiments have done exactly this, and it would ease the interpretation and statistical issues. The authors themselves state "scores between later timepoints were found to be linearly related" suggesting this approach would work well.

We are encouraged by this comment, and thank the reviewer for their appreciation of our time series approach to SGE, which goes beyond the standard 2 timepoint experimental design comparison employed in other studies. We agree and have changed our primary metric. See above for the interpretation of the 'continuous LFC', renamed 'functional score' throughout.

Line 148 - The authors assert that in silico predictors "were found to be highly sensitive, but not specific compared" to SGE. However, the cited extended figure does not contain any quantitative assertions about specificity and sensitivity of either SGE or the predictors (instead it just has plots where, because of overplotted points, it is impossible to appreciate sensitivity and specificity). The authors should make a quantitative comparison (e.g. by computing sensitivity and specificity) and summarize it in the main text.

We agree and have modified our approach to *in silico* comparisons. One of the major benefits of SGE is obtaining experimentally validated functional scores for missense variants *en masse*. Building a classifier with stop-gained and synonymous variants using *in silico* methods will be quite accurate in terms of distinguishing pathogenic and benign variants. For this reason, it would be misleading to directly compare ROC AUC for *in silico* classifiers as so few missense variants have been classified as truly 'likely pathogenic' or 'benign'. We have instead asked how well *in silico* tools report SGE functional classifications for missense variants, as a measure of comparability. We cannot make the claim that SGE is objectively more sensitive and specific than the *in silico* classifiers tested, as a quantitative metric cannot currently be computed to test this exact hypothesis, however we do see a relative difference in classification between SGE and *in silico* classification of missense on a per-variant level which suggests that the tools are less discerning in classifying variants as 'pathogenic', with proportionally more 'benign' classifications with SGE. See Extended Data Fig 7 (c and d-g), and modified main text:

Line 216: *“As very few BAP1 missense variants have been ascribed to be pathogenic or benign, a direct comparison of sensitivity and specificity using an AUC summary metric between in silico tools and SGE functional scores for missense variants alone is not possible. However, when we compare experimental data with in silico tools, we find that EVE, PolyPhen and CADD predict SGE classifications of non-splice region missense variants with 77-79% accuracy (Extended data Fig. 7c). With per-variant examination, it is notable that EVE, PolyPhen and CADD¹³ classify proportionally more missense variants as pathogenic/probably damaging/likely pathogenic, respectively, suggesting SGE may have a relatively higher specificity (Extended data Fig. 7d-g).”*

Line 150 - The authors assert that there are important differences between the frequency of functional classes of variants in gnomad. They should do a statistical test to support this assertion (e.g. chi-squared, etc). More importantly, figure 3d is shown as a bar chart but that hides the fact that the assertions are based on very small numbers (e.g. just two enriched variants are in gnomad). Perhaps each bar could be labeled with n=X?

We have performed the appropriate test and have added the numbers to the figure. Also please see responses to Reviewer 2 concerning gnomAD frequencies.

Line 167 - Editorial comment: It’s great that the authors did a good experiment, but the type of inter-replicate correlations they see are pretty standard for this type of experiment these days. I’m not sure that it warrants three main figure panels.

We thank Reviewer 3 for confirming the quality of our experiment, we agree and have reduced this point in the main text and figure (Figure 4a) and have moved the plots in question to extended data Figure 4 a & b.

Line 171 - What is a “PPE containing codon?” Nature Genetics is a generalist journal and this kind of jargon should be avoided. Moreover, I don’t know what a PPE codon is and why it is reasonable to remove them. The authors should explain.

We thank Reviewer 3 for this point, PPEs (PAM/protospacer protection edits) are fixed synonymous changes that are necessary to obtain high editing rates using spCas9 based cutting. We have defined the meaning in the text and subsequently use this abbreviation for brevity.

Line 90: *“These libraries contain two different synonymous PAM/protospacer protection edits (PPEs) which are refractory to sgRNA/Cas9 cutting, preventing cleavage of incorporated tracts”*

Line 171 - The authors used ClinVar data but don’t (as far as I could discover) explain how it was filtered. ClinVar is notoriously noisy, and the authors should remove variants from submitters from <1* at a minimum. They should also explain, in the methods, when ClinVar was accessed for their analysis.

We completely agree with Reviewer 3 and have changed all plots and analyses to be based on at least 1* status, with the exception of the “Evaluation of BAP1 SGE assay performance against ACMG evidence framework” section as it is common practice to take all ClinVar data in this analysis as performed elsewhere.

Line 175 - Ah, so the authors did calculate a growth rate (or something very like it)! The methods are confusing, but I think their “continuous LFC” is basically a growth rate computed from all the time points. Unsurprisingly, it performs better than any single LFC.

See above for the interpretation of the ‘continuous LFC’, renamed ‘functional score’ throughout.

Line 177 - The authors say that a LFC is “more intuitive” than the continuous/growth rate score which performed worse. But, it’s only more intuitive because the authors explain the continuous score in a confusing way - I think that growth rates are extremely intuitive for a cell growth assay!

We agree and have altered all analyses, classifications, and plots in response to the Reviewer 3’s points. We have also clarified the descriptions of our analyses to make the explanations less confusing.

Line 182 - The VUS distribution appears not to match the all missense distribution, with more VUS appearing non-damaging. Is that true by a statistical test? If so, it might be interesting to mention.

This is likely because Fig.4e in the original paper (now the corresponding figure is Fig.4c) is a histogram with variants binned into coarse intervals of 75 variants. The density plot in Fig.2e showing all missense, is a density plot without fine binning. This has the effect of appearing as though missense have a broader distribution than VUS. When both VUS and missense are plotted as density plots (Fig. R3 A) they are seen to have comparable distributions. On investigation it does appear that VUS are in aggregate less damaging than missense variants, as VUS have a statistically higher median functional score (Fig.R3 B) and proportionally more ‘unchanged’ classifications than missense variants (Fig.R3 C). As the sample sizes are very large, small differences may result in a significant difference in Fig.R3 B and C, we feel that this result should not be over-interpreted so have included here but have not mentioned in the text.

Figure Revision 3 A. Density plot showing VUS (purple) and missense (green) density distributions with tick marks for each variant and equal bandwidth of settings of 0.00866 (as in Fig.2e). The distributions around a functional score of 0 are similar. Both have a spectrum of depletion. Missense has many more observations, so this spectrum of change is more apparent. **B.** A boxplot showing that VUS variants have a significantly higher mean functional score than missense ($p < 0.0001$, two-sided Mann-Whitney-Wilcoxon Test), as the variants in the spectrum of depletion are mostly outliers to the mean this likely reflects that VUS are on average more likely to be classified as ‘unchanged’. **C.** Proportion of functional classification for VUS and missense, which is significantly different as measured by Chi-squared test ($\chi^2 = 19.77$, $p < 0.0001$).

Line 186 - The authors assert “Therefore ‘depleted’ variants can be classified as ‘Pathogenic/Likely pathogenic’ and ‘unchanged’ variants as ‘Benign/Likely benign’,” implying in Figure 4f that all missense VUS can be reclassified as pathogenic or benign on the basis of the data presented in the manuscript. This is egregiously wrong. VUS cannot be reclassified only on the basis of functional data alone. Instead, each must be reclassified using all available data (e.g. patient phenotype, pedigree, variant frequency, etc, etc). The authors own citation 19 are the (now old) rules for this procedure, which they did not follow. Citation 20 does actually do VUS reclassification using functional data and following the proper, updated rules. The authors should either do this analysis correctly or remove figure 4f and accompanying text. The data they collected is high quality and, if they were to use the appropriate rules would probably lead to the reclassification of many VUS. But, they didn’t do that here and shouldn’t imply that they did. The same comment applies to splice variants. A few sentences later the authors use “might be reclassified” phrasing, but this is totally inadequate especially when the figure gives the impression they did something they didn’t actually do

We thank Reviewer 3 for this important point. We have clarified our language throughout, being mindful that ‘pathogenic’ has a very definite meaning in clinical nomenclature. We have endeavoured to make our functional data accessible to clinicians in support of other evidence for classification and have therefore evaluated the quality of our data using standardised ACMG rules, see section: “Evaluation of BAP1 SGE assay performance against ACMG evidence framework”. In addition we have removed Figure 4f.

Line 202 - The authors “restricted our analyses to primarily European genetic ancestry (due to power)”. This is an insufficiently detailed explanation of the choice to exclude individuals from other ancestries. Why could those individuals not be included (e.g. is there some cancer- or BAP1-specific biology or genetics that means all individuals couldn’t be analyzed)? Did the authors actually check other ancestries to verify that there were insufficient numbers for the analysis? They should include a table of the UKB dataset by ancestry and explain in more detail why pooled analyses and analyses of other ancestries is not possible.

We thank Reviewer 3 for this point. We have rerun our analysis in UKBB without restricting samples to be European (which we originally did for reasons of statistical power). Importantly, we found more carriers and computed an even stronger p -values for nearly all gene burden masks. Please see Supplementary Table 7 for a comparison between variant and phenotype masks in European together with all ancestries.

Line 208 - “Editorial comment: we generated cancer-type phenotypic variables and rare variant burden test 209 masks” is another example of jargon that many readers will not follow. The authors should explain these types of procedures in simple terms that the generalist geneticist can understand.

Burden masks have been used in Nature Genetics previously (Rajagopol *et al.*, 2023¹⁴) and have been defined there in parenthesis as ‘variant sets’, we have chosen to follow this pattern, and have changed this sentence accordingly:

Line 312: “To evaluate the association of these variants to overall cancer risk, we generated cancer-type phenotypic variables and rare variant burden test masks (variant sets) (Fig.5a, Extended Data Fig.8a, Supplementary Table 7).”

Line 213 - The authors used CADD here, but EVE earlier. Why the switch? They should at least explain. I worry that CADD was cherry picked because it happened to not be very good in this analysis, whereas other predictors did better. The authors should therefore ideally conduct a more robust analysis using a small panel of predictors (e.g. EVE, REVEL, etc).

This is an important point, we certainly did not selectively present data/analysis. We only performed an analysis with CADD previously as this was used as the standard in the MRC Institute informatics pipeline, however we have now updated this analysis to include CADD, EVE and REVEL. There are five new masks: EVE \geq 0.5, EVE \geq 0.7, EVE \geq 0.75, REVEL \geq 0.5 and REVEL \geq 0.7. For all of these masks, we did not observe a significant association with cancers for European only or for All ancestries analysis across any cancer phenotype. We do when variants are classified by SGE (Fig5.a and Extended Data Figure 8a, Supplementary Table 7).

Line 220 - The authors state that SGE-depleted missense variant carriers do not have a significantly increased cancer burden and note that "This is likely due to low power and effect size of some variants." OK. It might be nice to actually show a spectrum of SGE scores for these 40 variants, with points colored by cancer/not cancer, so the reader could appreciate whether there was a gradient with effect size. The truncating variants could be included too, since they were scored in SGE.

We thank Reviewer 3 for this suggestion. We have amended this sentence to make clear that we mean PheWAS effect size rather than SGE functional score effect size, see response to comment below. However, the suggestion to examine SGE functional score effect size in relation to a cancer diagnosis for UKBB variants is interesting, so we have performed this analysis. We find that depleted missense variants associated with cancer have a significantly higher SGE functional score than high confidence protein truncating variants (HC PTVs) associated with cancer (Fig.R4). Depleted missense variants not associated with cancer have a higher median SGE functional score than depleted missense variants associated with cancer, although this difference is not significant.

As there are very few carriers, comparing PheWAS effect sizes for individual variants and performing statistical tests would lead to unreliable conclusions. This is in fact why we chose to run gene burden tests (masks/variant sets) as the effect sizes for individual variants are unlikely to yield informative associations. We observe that SGE depleted HC PTVs have a PheWAS effect size of 0.893 and SGE depleted missense variants have a PheWAS effect size of 0.411 (~ 2-fold difference). We see a similar magnitude of difference (~3-fold) in SGE functional score between cancer associated missense and cancer associated HC PTVs (median SGE depleted missense = -0.044, and median HC PTV = -0.143).

Taken together, all SGE depleted HC PTVs found in UKBB associate with cancer and have a large PheWAS effect size and low SGE functional score. SGE depleted missense variants have a smaller PheWAS effect size than HC PTVs, and those missense variants that associate with cancer have a higher SGE functional score than HC PTVs that associate with cancer. As very few variants and carriers are found in UKBB, it may be the case that with more observations of missense variants, SGE depleted missense variants may significantly associated with cancer. As there are few observations, and the link between PheWAS and functional score effect size is speculative, we have chosen not to include this analysis in the manuscript.

Figure Revision 4 Boxplot to compare SGE functional score for SGE depleted missense and PTV variants found in UKBB with and without a cancer diagnosis. 26 SGE depleted missense variants were found in 43 carriers in UKBB. 20/26 missense variants are associated with a cancer diagnosis in at least one carrier. 6/26 depleted missense variants are not associated with a cancer diagnosis. These missense variants (blue) have a higher median functional score than cancer associated depleted missense variants (red), but this difference is not significantly different (ns, $p=0.11$, two-sided Mann-Whitney-Wilcoxon Test). Eight SGE depleted high confidence protein-truncating variants (PTVs) are found in the UKBB in 39 carriers. All associate with cancer, and have a significantly lower functional score compared with cancer associated missense variants ($p<0.0001$, two-sided Mann-Whitney-Wilcoxon Test).

Line 222 - See comment above, but I didn't follow what "roughly double the effect" meant here - is this in the PheWAS or the SGE?

We have amended this sentence for clarity:

Line 326: "This is likely due to low power and PheWAS effect size of some variants, with few ($n=43$) missense BAP1 carriers observed in UKBB, and the observation that SGE-depleted high-confidence (HC) protein truncating variants (PTVs) have roughly double the effect compared with SGE-depleted missense variants (0.898 and 0.411, respectively), when all cancers combined are assessed. Consistent with this, SGE-depleted missense variants significantly associate with solid cancers (i.e., excluding blood), with blood cancers generally not linked to BAP1 mutation/loss¹⁵ (Extended Data Fig.8a)."

Also please see comment above, concerning a comparison between PheWAS and SGE effect sizes.

Line 227 - The authors identify a significant association with IGF-1 but don't say whether the p-value was corrected for multiple testing across all variables. They should explain.

We have run our analysis for 24 traits. The adjusted p -value threshold is $0.05/24=2.08e-03$. In our manuscript, the p -values we reported for IGF1 are $1.17e-03$ for SGE-depleted non-synonymous BAP1 variant and $9.77e-04$ for SGE-depleted non-synonymous BAP1 variant plus PTV carriers, both are below the threshold, so the significance for both observations survives multiple testing.

We have also used the Bonferroni-Hochberg correction of p -values, with q -values for the IGF-1 association of 0.02809172 and 0.02344595 found for SGE depleted non-synonymous variants and HC

PTV variants, respectively. Both are below $p=0.05$ and therefore survive multiple testing. Traits examined together with effect sizes, p -values are q -values are shown in Supplementary Table 8.

Line 250 - The authors describe a case report, stating that the “variant had been classified as a VUS with no functional data available, SGE functional classification will now likely impact clinical management of this kindred.” Much more information is required here. On the basis of what data was the variant classified as VUS? Was the variant actually reclassified by a clinician? If so, on the basis of what data (other than the SGE data). See comment above.

We have amended this section, with this sentence:

Line 363: *“This variant had been classified in the clinic as a VUS, but together with our SGE data has been re-classified as likely pathogenic (ACMG, class IV), a result that will contribute to the clinical management of this kindred.”*

In addition to updating the pedigree with more digestible descriptions as suggested by Reviewer 3, we have also added histological data to give increased context and to make clear that this is a specific, clinically managed kindred and not a meta-data analysis of a ClinVar accession for example.

Line 259 - What is a “short variant”?

The short variant category in the foundation medicine study includes SNVs and indels which are validated for 1-40bp in length¹⁶. We have now reduced this section so it is no longer in the text choosing to reference the foundation medicine paper to reduce word count, but we have responded to this question here for completeness.

Line 270 - I have major reservations about this analysis. Must the authors really use UMAP to conduct a growth rate analysis? Looking at Figures 6a, b and c make it pretty clear that the UMAP is just “discovering” growth rate. In fact the authors state this “As change across time is the principal dimension regressed from changes between multiple timepoints using UMAP, the ‘continuous LFC’ metric—which is an orthogonally-calculated change across time—was used to compare LFCs between clusters (Fig.6b).” However, the “continuous LFC” metric, which is just a complicated and not-as-good-as-regression method for calculating a growth rate, is in no way orthogonal to the UMAP, because the data fed into both analyses is identical. If the authors want to use a UMAP, they need to defend why it is superior to simply presenting “continuous LFC” or, better, a growth rate derived from the variant frequency data.

We have removed this analysis and have replaced it with a ranking approach, as above. The continuous LFC metric is in fact a regression method that is described in detail above and in the Methods.

Line 300 - The authors make “rate of change” arguments. I don’t think they are actually trying to claim that the rate of depletion of a variant changes over time - I’m guessing most variants have log-linear depletion (as expected). But, if they are actually saying that the rate at which a variant depletes or enriches changes over time, then they need a more rigorous analysis to prove it (e.g. deviation from log linearity).

We have removed the UMAP section, so this point is no longer included. Reviewer 3 is correct; we did not wish to make rate of change arguments. Our functional score is change per unit time, so is a growth rate. The log-linear approach assumes monotonic exponential decay of depleted variants over all timepoints, so to investigate rate change of depletion at any point along the curve

irrespective of the whole would require a different metric (such as LFC between two categorical timepoints, which we have used in one analysis, D4 vs D10 in Extended Data Fig. 7h).

Line 325 - This entire section boils down to “cancer and developmental Bap1 variants have a similar spectrum of SGE variant effects.” It’s great that the authors SGE data line up with previous low-scale functional measurements. But, the take-home for this section and indeed the paper is that a simple growth assay could not distinguish the pleiotropic effects of variants in Bap1 (e.g. between cancer and developmental effects). That take home is later mentioned in the discussion, “It is interesting that variants with similar depletion kinetics are associated with both cancer and developmental disorders,” but in other places the manuscript suggests otherwise.

We agree that it is great that our data are highly correlated with orthogonal functional effects screens, which in the absence of missense truth sets, helps readers to appreciate the reliability of the SGE functional scores and classifications, together with other metrics of quality, such as the ROC AUC, PR AUC and the ACMG evaluation we describe. We have altered our text to simply say that variants behave similarly in both cancer and Kury-Isidor syndrome. We also expand upon the link between IGF-1, cancer and developmental phenotypes in the discussion.

Figure 2a - The legend is too pithy to really understand what this panel is showing. Presumably every plot is an exon and every line is a different sgRNA?

We have updated the panel to show that each plot is indeed an Exon. The key shows that each line is a sgRNA. We have expanded our explanation of the sgRNA evaluation selection process in the methods section, as described above.

Figure 2b - Forgive me if I’m missing something, but it seems like a Cas9+, LIG4+ condition is required in this panel. Also, the legend should show replicates if they exist and also include number of cells sorted (e.g. >3,000 or whatever).

Key experimental lines are *LIG4-* and Cas9+ as polyclonal and as a clonal A5 line, the latter outperforming the polyclonal line in terms of Cas9 activity (arrow in Fig.2b).

We do not use *LIG4+* cells experimentally. It has previously been reported by Findlay et al. 2018, that *LIG4+* HAP1 cells expressing Cas9 have lower editing rates than *LIG4-* HAP1 cells containing the Cas9 construct. We included two control lines, which are both negative for Cas9, neither of which cut the control construct, as expected. We included *LIG-* and *LIG4+* (Cas9-) to show that the *LIG4* background does not influence the reporter assay. In this context, a *LIG4+* Cas9+ line is not necessary to include.

We have made clear how many cells were screened in the legend and in the methods section. This particular experiment did not include replicates, however the FACS protocol has been used multiple times when a new cell bank is thawed and expanded, with consistent results seen for % Cas9 activity in the A5 clonal line.

Figure 2e - Editorial comment: the tick marks are pretty useless because they are so dense and actually obscure the tail of the missense distribution, making it look like all missense are synonymous-like. I suggest doing something different.

We have moved the tick marks to below the limit of the y-axis so that they do not overlap the density plots and no longer obscure any data. We feel this nicely summarizes how the data is distributed for different mutational consequences, providing more granularity than a violin plot or similar.

Figure 3e - Editorial comment: most heatmaps of this type have a blue/red color scheme. Why break this convention?

We have changed heat maps (and other figures) throughout to show negative as blue and positive as red. We have further modified heat maps for clarity by using the discrete 'functional classifications' rather than the continuous 'functional scores', which we believe to more clearly represent the data.

Figure 5d - This panel would benefit from plain text annotation or a graphical legend as opposed to 3 letter abbreviations.

We thank Reviewer 3 for this request, we have updated the pedigree to be more immediately interpretable.

We sincerely thank Reviewer 3 for their insightful comments and suggestions. This review has made us thoroughly revise our analyses making us even more confident that our data is indeed of extremely high quality and robustness. We have addressed all of Reviewer 3's points in full and the manuscript is much improved as a result of these revisions.

Response to Reviewers References:

1. Walpole, S. *et al.* Comprehensive Study of the Clinical Phenotype of Germline BAP1 Variant-Carrying Families Worldwide. *JNCI: Journal of the National Cancer Institute* **110**, 1328–1341 (2018).
2. Landrum, M. J. *et al.* ClinVar: improving access to variant interpretations and supporting evidence. *Nucleic Acids Res* **46**, D1062–d1067 (2018).
3. Findlay, G. M. *et al.* Accurate classification of BRCA1 variants with saturation genome editing. *Nature* **562**, 217–222 (2018).
4. Chen, S. *et al.* A genome-wide mutational constraint map quantified from variation in 76,156 human genomes. *bioRxiv* 2022.03.20.485034 (2022)
doi:10.1101/2022.03.20.485034.
5. Laloo, F. *et al.* Clinical practice guidelines for the diagnosis and surveillance of BAP1 tumour predisposition syndrome. *European Journal of Human Genetics* **31**, 1261–1269 (2023).
6. Sonesson, C. & Delorenzi, M. A comparison of methods for differential expression analysis of RNA-seq data. *BMC Bioinformatics* **14**, 91 (2013).
7. Love, M. I., Huber, W. & Anders, S. Moderated estimation of fold change and dispersion for RNA-seq data with DESeq2. *Genome Biol* **15**, 550 (2014).
8. Brnich, S. E. *et al.* Recommendations for application of the functional evidence PS3/BS3 criterion using the ACMG/AMP sequence variant interpretation framework. *Genome Med* **12**, 3 (2019).
9. Fayer, S. *et al.* Closing the gap: Systematic integration of multiplexed functional data resolves variants of uncertain significance in BRCA1, TP53, and PTEN. *The American Journal of Human Genetics* **108**, 2248–2258 (2021).
10. Kim, H.-K. *et al.* Impact of proactive high-throughput functional assay data on BRCA1 variant interpretation in 3684 patients with breast or ovarian cancer. *J Hum Genet* **65**, 209–220 (2020).
11. Osher, E. & Macaulay, V. M. Therapeutic Targeting of the IGF Axis. *Cells* **8**, (2019).
12. Barbon, L. *et al.* Variant Library Annotation Tool (VaLiAnT): an oligonucleotide library design and annotation tool for saturation genome editing and other deep mutational scanning experiments. *Bioinformatics* **38**, 892–899 (2022).
13. Rentzsch, P., Witten, D., Cooper, G. M., Shendure, J. & Kircher, M. CADD: predicting the deleteriousness of variants throughout the human genome. *Nucleic Acids Res* **47**, D886–D894 (2019).
14. Rajagopal, V. M. *et al.* Rare coding variants in CHRNA2 reduce the likelihood of smoking. *Nat Genet* **55**, 1138–1148 (2023).
15. Asada, S. *et al.* Mutant ASXL1 cooperates with BAP1 to promote myeloid leukaemogenesis. *Nat Commun* **9**, 2733 (2018).
16. Frampton, G. M. *et al.* Development and validation of a clinical cancer genomic profiling test based on massively parallel DNA sequencing. *Nat Biotechnol* **31**, 1023–1031 (2013).

Response to Reviewers Supplementary Page (Love et al, 2014):

of a wide range of techniques that require homoskedastic input data, including machine-learning or ordination techniques such as principal-component analysis and clustering.

DESeq2 hence offers to practitioners a wide set of features with state-of-the-art inferential power. Its use cases are not limited to RNA-Seq data or other transcriptomics assays; rather, many kinds of high-throughput count data can be used. Other areas for which *DESeq* or *DESeq2* have been used include ChIP-Seq assays (e.g., [39]; see also the *DiffBind* package [40, 41]), barcode-based assays (e.g., [42]), metagenomics data (e.g., [43]), ribosome profiling [44] and CRISPR/Cas-library assays [45]. Finally, the *DESeq2* package is well integrated in the Bioconductor infrastructure [10] and comes with extensive documentation, including a vignette that demonstrates a complete analysis step by step and discusses advanced use cases.

Methods

A summary of the notation used in the following section is provided in Supplemental Table S1.

Model and normalization

The read count K_{ij} for gene i in sample j is described with a generalized linear model (GLM) of the Negative Binomial family with logarithmic link:

$$K_{ij} \sim NB(\text{mean} = \mu_{ij}, \text{dispersion} = \alpha_i) \quad (1)$$

$$\begin{aligned} \mu_{ij} &= s_{ij} q_{ij} \\ \log q_{ij} &= \sum_r x_{jr} \beta_{ir}. \end{aligned} \quad (2)$$

For notational simplicity, the equations here use the natural logarithm as the link function, though the *DESeq2* software reports estimated model coefficients and their estimated standard errors on the \log_2 scale.

By default, the normalization constants s_{ij} are considered constant within a sample, $s_{ij} = s_j$, and are estimated with the median-of-ratios method previously described and used in *DESeq* [4] and *DEXSeq* [29]:

$$s_j = \text{median}_{i: K_i^R \neq 0} \frac{K_{ij}}{K_i^R} \quad \text{with} \quad K_i^R = \left(\prod_{j=1}^m K_{ij} \right)^{1/m}.$$

Alternatively, the user can supply normalization constants s_{ij} calculated using other methods (e.g., using *cqn* [12] or *EDASeq* [13]), which may differ from gene to gene.

Expanded design matrices

For consistency with our software’s documentation, in the following text we will use the terminology of the *R* statistical language. In linear modeling, a categorical variable or *factor* can take on two or more values or *levels*. In standard design matrices, one of the values is chosen as a reference value or *base level* and absorbed into the intercept. In standard GLMs, the choice of base level does not influence the values of contrasts (LFCs). This, however, is no longer the case in our approach using ridge-regression-like shrinkage on the coefficients (described below), when factors with more than two levels are present in the design matrix, because the base level will not undergo shrinkage while the other levels do.

To recover the desirable symmetry between all levels, *DESeq2* uses *expanded design matrices* which include an indicator variable for *each* level of each factor, in addition to an intercept column (i.e., none of the levels is absorbed into the

Decision Letter, first revision:

19th Nov 2023

Dear Dr. Adams,

Thank you for submitting your revised manuscript "Comprehensive saturation genome editing of BAP1 to functionally classify somatic and germline variants" (NG-A62524R). It has now been seen by the original referees and their comments are below. The reviewers find that the paper has improved in revision, and therefore we'll be happy in principle to publish it in Nature Genetics, pending minor revisions to satisfy the referees' final requests and to comply with our editorial and formatting guidelines.

Congratulations!

My best wishes,
Chiara

Chiara Anania, PhD
Associate Editor
Nature Genetics
<https://orcid.org/0000-0003-1549-4157>

Reviewer #2 (Remarks to the Author):

All the previously comments have been sufficiently addressed and I have no other concerns regarding the paper.

A: Preemptive evaluation of all almost all SNPs in BAP1 gene. Novel correlation to IGF-1 concentration
b. Original as it has not been performed with BAP1 previously, and clinical extremely significant, as this is needed in a clinical setting.

C. Valid approach, high quality data

D. relevant statistics

E. Robust, valid and reliable

F. none

G appropriate

H. Clear

Reviewer #3 (Remarks to the Author):

In their revision, the authors have addressed all of my major and most minor points. I particularly appreciate the work they put into rationalizing their fitness data, and addressing my concerns about the validity of their statistical procedures. Although it should not be a requirement for publication, I suggest that they add some of the detail from their response document to the methods of their manuscript. I think that the rationale for using DESeq2 and related statistical analyses would be particularly helpful to add.

I have no further substantive reservations regarding this manuscript.

Author Rebuttal, first revision:

None

Final Decision Letter:

14th May 2024

Dear Dr. Adams,

I am delighted to say that your manuscript "Saturation genome editing of BAP1 functionally classifies somatic and germline variants" has been accepted for publication in an upcoming issue of Nature Genetics.

Your paper will be published online after we receive your corrections and will appear in print in the next available issue. You can find out your date of online publication by contacting the Nature Press Office (press@nature.com) after sending your e-proof corrections.

Acceptance is conditional on the data in the manuscript not being published elsewhere, or announced in the print or electronic media, until the embargo/publication date. These restrictions are not

intended to deter you from presenting your data at academic meetings and conferences, but any enquiries from the media about papers not yet scheduled for publication should be referred to us.

Please note that *Nature Genetics* is a Transformative Journal (TJ). Authors may publish their research with us through the traditional subscription access route or make their paper immediately open access through payment of an article-processing charge (APC). Authors will not be required to make a final decision about access to their article until it has been accepted. Find out more about Transformative Journals

Authors may need to take specific actions to achieve compliance with funder and institutional open access mandates. If your research is supported by a funder that requires immediate open access (e.g. according to Plan S principles) then you should select the gold OA route, and we will direct you to the compliant route where possible. For authors selecting the subscription publication route, the journal's standard licensing terms will need to be accepted, including <https://www.nature.com/nature-portfolio/editorial-policies/self-archiving-and-license-to-publish>. Those licensing terms will supersede any other terms that the author or any third party may assert apply to any version of the manuscript.

If you have not already done so, we invite you to upload the step-by-step protocols used in this manuscript to the Protocols Exchange, part of our on-line web resource, natureprotocols.com. If you complete the upload by the time you receive your manuscript proofs, we can insert links in your article that lead directly to the protocol details. Your protocol will be made freely available upon publication of your paper. By participating in natureprotocols.com, you are enabling researchers to more readily reproduce or adapt the methodology you use. [Natureprotocols.com](http://natureprotocols.com) is fully searchable, providing your protocols and paper with increased utility and visibility. Please submit your protocol to <https://protocolexchange.researchsquare.com/>. After entering your nature.com username and

password you will need to enter your manuscript number (NG-A62524R1). Further information can be found at <https://www.nature.com/nature-portfolio/editorial-policies/reporting-standards#protocols>

Sincerely,
Chiara

Chiara Anania, PhD
Associate Editor
Nature Genetics
<https://orcid.org/0000-0003-1549-4157>